# The influence of nonlinear resonance on human cortical oscillations

Ying Wang [1], Min Li [2], Ronaldo García Reyes [1,3], Maria L. Bringas-Vega [1,3], Ludovico Minati [1,4], Michael Breakspear [5] ✉ & Pedro A. Valdes-Sosa [1,3] ✉

Whether macroscale brain signals reflect linear or nonlinear organization remains poorly characterized. This distinction matters for modeling neural dynamics and interpreting oscillatory biomarkers of cognition and disease. Spectral analysis reveals aperiodic broadband and rhythmic narrowband components but does not capture nonlinear resonance, such as quadratic phase coupling among oscillations, which requires higher-order spectral analysis. We introduce BiSpectral EEG Component Analysis (BiSCA), combining spectral and bispectral analysis to separate aperiodic (Xi) from rhythmic (Rho) components, localize nonlinear signatures, and distinguish nonlinearity from non-Gaussianity; simulations confirm this separation. Applying BiSCA to two large datasets (1771 intracranial channels; 960 scalp EEG subjects), we detect significant nonlinear or non-Gaussian structure in 81.6% of scalp EEG and 67.9% of iEEG channels; forward modeling indicates the higher scalp prevalence reflects volume-conduction spread of focal nonlinear sources. In spatially focal iEEG, aperiodic Xi shows no detectable quadratic nonlinearity or non-Gaussianity, whereas Rho components, including Alpha and Mu, carry the dominant cortical quadratic coupling. Despite higher occipital Alpha power, the strongest nonlinear signatures arise from parietal Mu. Nonlinear resonance is thus expressed primarily through oscillatory rather than aperiodic dynamics.

Can mesoscale brain dynamics, as reflected in resting-state EEG (rsEEG) and fMRI (rsfMRI), be described entirely as linear, Gaussian stochastic processes? Recent studies, such as Nozari et al. [1] and Raffaelli et al. [2], argue that linear time-series models suffice to describe these signals. The authors of these studies suggest that the intricate nonlinearities observed at the microscale—by neurons, synapses, and neural masses—are effectively "averaged out" at larger scales. If this observation were true, the physics of the observation process would imply that mesoscale signals lack observable nonlinear signatures, limiting our ability to infer microscale dynamics from macroscopic measurements. Conversely, empirical refutation of this claim would imply that neuronal signals possess correlations at all scales, sufficient to avoid the diffusive "washing out" of macroscopic averaging and thus retain nonlinear properties at large scales[3].

The high temporal resolution of the scalp or intracranial resting-state EEG (rsEEG) (Fig. 1a) makes it particularly suitable for investigating dynamic neural processes. Therefore, reports asserting linearity or Gaussianity are somewhat surprising since substantial evidence indicates that nonlinear dynamics are observable in rsEEG signals[4,5]. Nonlinear dynamics

in neural systems, as evidenced by phenomena in the frequency domain such as nonlinear resonance, by which we denote quadratic frequency mixing that generates combination frequencies matching natural modes of the system, yielding amplified oscillations[6,7], cross-frequency coupling (CFC), and phase-amplitude coupling (PAC), are critical for understanding the complex oscillatory interactions observed in resting-state EEG (rsEEG) and task-related EEG signals. One of these common features of nonlinear neural oscillations is their non-sinusoidal waveform shape, which is abundant in human electrophysiological recordings and reflects the complex biophysical dynamics of underlying neural generators[8], as modeled by early neural mass models[9–12]. While the Fourier transform faithfully captures non-sinusoidal waveforms through harmonic decomposition, the conventional power spectrum discards the inter-harmonic phase relationships. Two signals with identical power spectra can exhibit fundamentally different phase structures; the bispectrum preserves this phase coupling information that the power spectrum discards. In time-domain terms, the power spectrum is determined by a single lag, whereas the bispectrum depends on two lags $c_3(\tau_1, \tau_2) = \mathrm{E}[x(t)x(t + \tau_1)x(t + \tau_2)]$ and therefore

[1]China-Cuba Belt and Road Joint Laboratory on Neurotechnology and Brain-Apparatus Communication, University of Electronic Science and Technology of China, Chengdu, China. [2]Hangzhou Dianzi University, Hangzhou, Zhejiang, China. [3]Cuban Neuroscience Center, Havana, Cuba. [4]Center for Mind/Brain Sciences (CIMeC), University of Trento, Trento, Italy. [5]School of Science, College of Engineering, Science and the Environment, University of Newcastle, Newcastle, New South Wales, Australia. ✉e-mail: mjbreaks@gmail.com; pedro.valdes@neuroinformatics-collaboratory.org

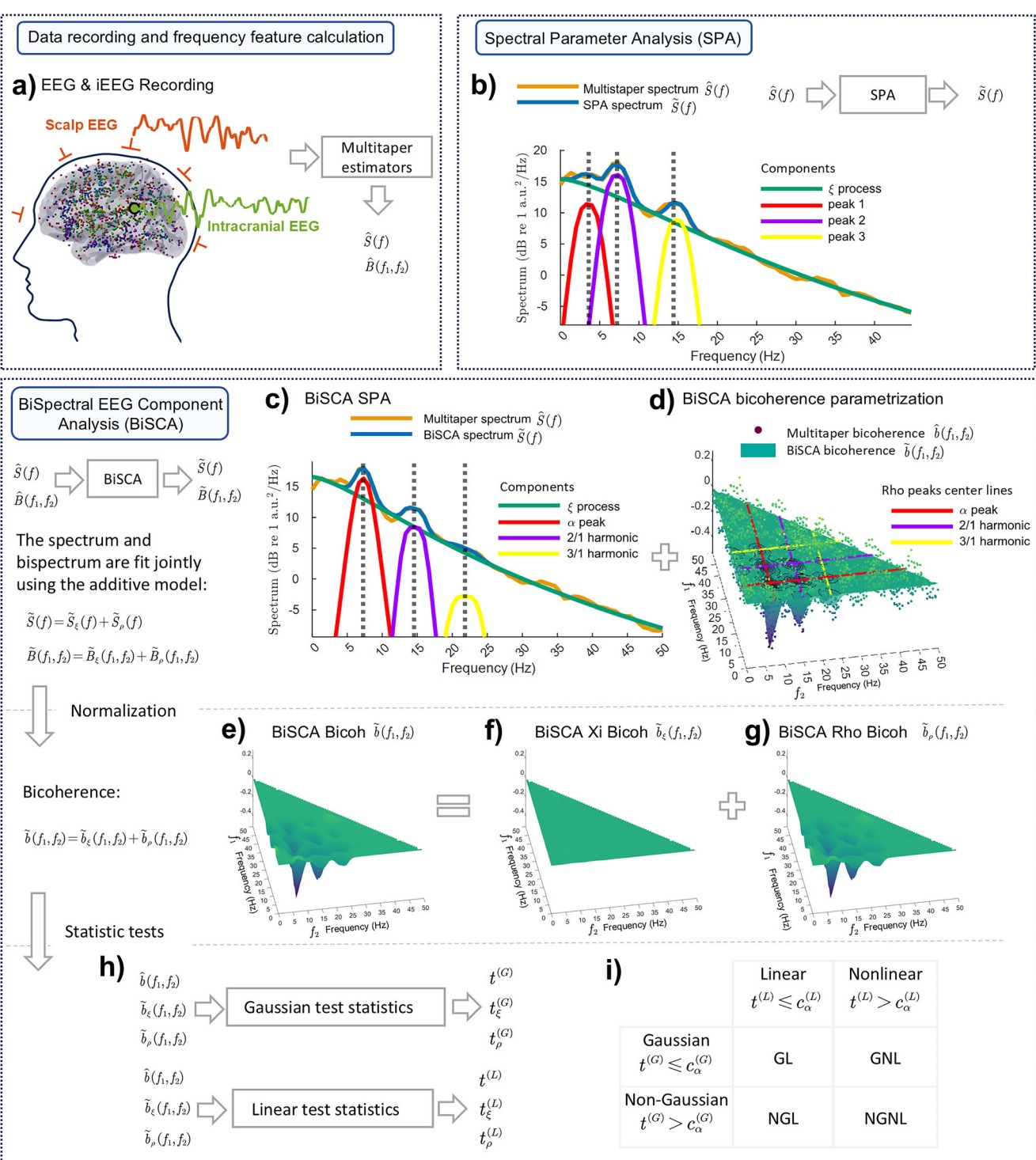

**Fig. 1 | Overview of the BiSpectral Components Analysis (BiSCA) framework.**
**a** Data recording (iEEG and scalp EEG) and calculation of Multitaper spectral and bispectral estimators. **b** Spectral Parameter Analysis (SPA) decomposing the power spectrum into individual components. **c** BiSCA model fitting the power spectrum, capturing harmonic relationships. **d** BiSCA model fitting the bicoherence to parametrize nonlinear interactions. The peak labeled "alpha" at bifrequency (10, 10) Hz represents the quadratic phase coupling alpha + alpha -> 2alpha, i.e., the interaction that generates the second harmonic at 20 Hz. The point $(f_1, f_2)$ denotes coupling that generates $f_1 + f_2$; thus the 'alpha peak' at $(\alpha, \alpha)$ represents $\alpha + \alpha \to 2\alpha$

coupling. Colored lines indicate where each harmonic acts as a coupling source (**e**–**g**) Additive decomposition of the BiSCA bicoherence into (**e**) total bicoherence, (**f**) the Xi-component, and (**g**) the Rho-component. **h** Derivation of statistics for Gaussianity and linearity tests from bicoherence measures. **i** Table for classifying a signal as Gaussian/Non-Gaussian and Linear/Nonlinear based on statistical thresholds. Panels displaying $Re[b(f_1, f_2)]$ are signed and can be negative. Magnitude bicoherence $|b(f_1, f_2)|$ is nonnegative and bounded in [0,1]. A flat pattern in $|b|$ does not imply zero raw bispectrum $B(f_1, f_2)$.

**Table 1 | Percentages of EEG and iEEG recordings that are Linear, Nonlinear, Gaussian or non-Gaussian**

|  | EEG | | | iEEG | | |
|---|---|---|---|---|---|---|
|  | Linear | Nonlinear | Total | Linear | Nonlinear | Total |
| Gaussian | 14.7 | 39.3 | 54 | 25.7 | 17.1 | 42.7 |
| Non-Gaussian | 3.6 | 42.3 | 46 | 6.4 | 50.9 | 57.3 |
| Total | 18.4 | 81.6 | 100 | 32.1 | 67.9 | 100 |

captures higher-order temporal structure beyond second-order statistics. We illustrate this in Fig. S2 where simulated time series with identical power spectra (same AR model) but different innovations or system nonlinearities produce distinct bicoherence signatures. These carry important physiological information, and can distinguish between different behavioral states[8,13–15]. Such complex waveforms inevitably generate harmonic components in the frequency spectrum[16–24]. This asymmetric nonlinearity or quadratic phase coupling can be described by bicoherence[9,25]. This emphasis on Bicoherence is bolstered by recent theoretical and empirical work that shows that other popular measures, such as phase-amplitude coupling, are actually equivalent to it. This measure thus provides a unified framework for understanding nonlinear resonance in neural systems[26]. Furthermore, large-scale brain network models have shown that cross-frequency coupling emerges naturally from the nonlinear dynamics of interconnected neural populations, supporting the view that nonlinear resonance is a fundamental organizing principle of brain activity[27]. It is important to distinguish two sources of higher-order structure: nonlinearity of the system (frequency-dependent bicoherence) and non-Gaussianity of the input (constant bicoherence across frequencies). Critically, if macroscopic signals were limited to low-order statistical properties (e.g., mean, variance), none of these nonlinear resonance phenomena would have been observed. Moreover, widely and successfully used techniques like Independent Component Analysis (ICA) — which relies on higher-order moments—would fail to separate signal sources.

The debate upon the linearity or nonlinearity of the EEG tends to be muddied by the fact that a critical concept is ignored, namely, the distinction between linearity, that is, the nature of a system's temporal evolution, and Gaussianity, that is, the statistical distribution of its variables. Even when a system is linear, for example, a linear autoregressive model, when driven by non-Gaussian inputs (e.g., skewed or heavy-tailed noise), it can yield non-Gaussian outputs. On the other hand, a nonlinear system can also produce approximately Gaussian output due to the summation of multiple independent processes, where the Central Limit Theorem applies. Non-Gaussianity can also stem from the intrinsic dynamics of the system[28], especially in the presence of harmonic coupling or multiplicative feedback. Consequently, neural dynamics can embody any combination of linearity/nonlinearity and Gaussianity/non-Gaussianity. These are two orthogonal properties: system linearity/nonlinearity is determined by the presence of higher-order Volterra kernels $(H_2, H_3, ...)$[29–31], while input Gaussianity/non-Gaussianity is determined by the third cumulant $\gamma_3$ of the i.i.d. innovation. Crucially, these produce distinct bicoherence signatures (Table 1): frequency-dependent bicoherence reflects system nonlinearity, whereas constant bicoherence across frequencies indicates non-Gaussian input[32]. Simple inspection of a power spectrum cannot disambiguate these generative mechanisms, as demonstrated in Supplementary Materials S.3. Distinguishing the generative mechanisms requires moving beyond distributional analysis or spectral tests and directly testing for nonlinear interactions. Prime examples of nonlinear behavior to search for are nonlinear resonance phenomena like phase coupling and harmonic generation.

It is also important to recognize that the EEG signal is likely a mixture of different types of processes and that Gaussianity or linearity (or their absence) may vary across these processes. By dissecting the contributions of various subprocesses—such as oscillatory rhythms, aperiodic activity, and noise—it becomes possible to identify which components adhere to linear

dynamics and Gaussian statistics and which deviate due to nonlinear interactions or non-Gaussian inputs.

A successful area of research for decomposing the EEG into such processes is based on the EEG second-order cumulant $S(f)$ or spectrum for frequency $f$. For strictly stationary processes without long-term memory[32], the distribution of optimized estimators $\hat{S}(f)$ (for example, with Multi-taper) completely characterizes linear time-invariant signals driven by noise. The separation of an EEG $\hat{S}(f)$ into distinct components defined by simple functional forms, each determined by only a few parameters, is known as **Spectral Parameter Analysis** (**SPA**) (Fig. 1b & Supplementary S.2). An early example of SPA is Pascual-Marqui et al.[33], who proposed the Xi-Alpha (ξα) model. They showed that EEG spectra have two dominant components: an aperiodic Xi (ξ) component and multiple peaks characterizing rhythmic activity (which we will call generically Rho components or ρ), the best known being the Alpha (α) component (the Peak 1 in Fig. 1b). The FOOOF (Fitting Oscillations & One Over F) model is a recent example of SPA[34]. Not only do these components have distinct behavioral correlates[35–37], they may also derive from distinctive physiological processes[38–40].

However, SPA, which relies on second-order moments (the power spectrum), only provides a sufficient characterization for linear Gaussian systems. For such systems, all higher-order cumulants are zero. Consequently, SPA is thus blind to nonlinear dynamics. For this type of activity, higher-order spectral tools, such as the bispectrum $B(f_1, f_2)$ and its normalized version, the bicoherence $b(f_1, f_2)$, between frequencies $f_1$ and $f_2$, are required to detect phenomena such as harmonic generation, cross-frequency phase coupling, or other nonlinear interactions. Nonlinear analysis of time series is therefore needed. The Wiener-Khinchin theorem implies that the power spectrum, as the Fourier transform of the autocorrelation function, is a purely second-order statistic and therefore captures only linear correlations. By contrast, the bispectrum explicitly quantifies inter-frequency phase coupling. Another limitation of state-of-the-art SPA is that it does not explicitly model harmonic relationships among second-order spectral peaks, although such relationships are ubiquitous in EEG recordings.

We introduce **BiSCA** (**Bi**Spectral EEG **C**omponent **A**nalysis), a generalized model of EEG spectral components designed to detect and quantify nonlinear resonance. It thus addresses the limitations of the current practice of relying solely on isolated second-order spectral peaks. Instead, BiSCA provides a combined model specifically designed to identify nonlinear resonance phenomena in neural signals, also modeling harmonically related second-order peaks and their corresponding bispectral counterparts. Furthermore, BiSCA parameter estimation is based on maximum likelihood, enabling direct statistical testing of linearity and Gaussianity for each component (Fig. 1e–g). Following in-silico validation, we demonstrate how the BiSCA model reveals substantial nonlinearities and non-Gaussian characteristics in EEG data using 1771 resting-state intracranial EEG (iEEG) recordings from 106 patients.

## Results
### Overview of BiSpectral Components Analysis (BiSCA)
The BiSpectral Components Analysis (BiSCA) framework is designed to uncover nonlinearity and non-Gaussianity in resting-state EEG signals. Figure 1 provides a schematic overview of this methodology. The analysis pipeline begins with electrophysiological data, including intracranial (iEEG) and scalp EEG recordings (Fig. 1a). From these time series, we compute initial data summaries using Multitaper estimators to obtain the power spectrum, denoted as $\hat{S}(f)$, and the bispectrum, $\hat{B}(f_1, f_2)$.

A conventional approach, Spectral Parameter Analysis (SPA), decomposes the power spectrum $\widetilde{S}(f)$ on either a natural or logarithmic scale into a linear superposition of an aperiodic background component and several oscillatory peaks (Fig. 1b). However, SPA does not account for higher-order statistics, a limitation that prevents it from modeling the harmonic relationships between these peaks.

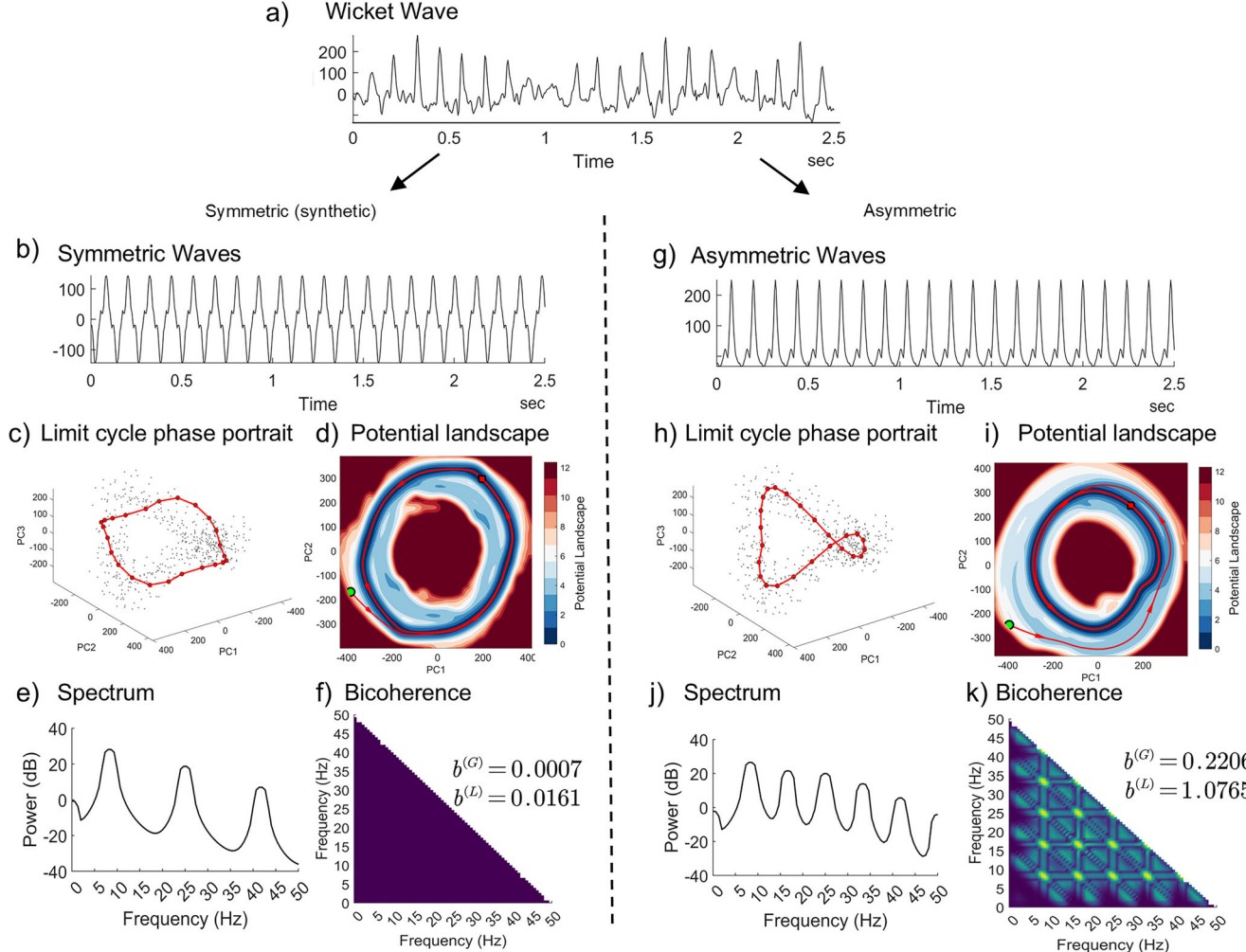

**Fig. 2 | Comparison of asymmetric and symmetric nonlinear systems to illustrate quadratic nonlinearity.** The simulations are derived from an NNAR model fitted to an iEEG Wicket Wave segment (**a**). Left Panel (**b**–**f**): Synthetic symmetric system. (**b**) Symmetric waveform; (**c**) symmetric 3D limit cycle phase portrait (red line) with pruned model states (gray dots) used for the model fit; (**d**) symmetric 2D potential landscape; (**e**) corresponding power spectrum exhibiting only odd-order harmonics; and (**f**) negligible bicoherence, with $b^{(L)} = 0.0161$, below the significance threshold

$c_\alpha^{(L)}$ estimated in Methods. Right Panel (**g**–**k**): Asymmetric system. **g** Asymmetric waveform; (**h**) asymmetric 3D limit cycle; (**i**) asymmetric 2D potential well; (**j**) power spectrum containing both even and odd harmonics; and (**k**) significant bicoherence peaks indicating quadratic phase coupling. PC1, PC2, and PC3 denote the first three principal components from PCA applied to Takens' delay-embedded state vectors (Hankel-matrix time-delay embedding with lag order p).

The BiSCA model extends this approach by jointly fitting both the power spectrum and the bispectrum (Fig. 1c, d). This joint modeling explicitly parameterizes the nonlinear resonances that manifest as harmonic relationships in the spectrum and corresponding peaks in the bispectrum. The model additively separates the signal into components, such as the aperiodic Xi ($\xi$) process and the oscillatory Rho ($\rho$) process. Figure 1c shows the BiSCA model fits to the power spectrum, identifying a fundamental alpha peak and its harmonics. Figure 1d illustrates the corresponding fit to the bicoherence $\hat{b}(f_1, f_2)$ (the normalized bispectrum $\widetilde{B}(f_1, f_2)$), where harmonic relationships are visible as distinct patterns. A key feature of BiSCA is the ability to decompose the total bicoherence into contributions from individual components (Fig. 1e–g). For instance, the total bicoherence (Fig. 1e) can be separated into the bicoherence arising from the Xi-component (Fig. 1f) and the Rho-component (Fig. 1g), isolating the source of nonlinearity.

Finally, to formally test for the presence of such phase-coupled interactions, statistical tests are applied based on the bicoherence estimates to assess Gaussianity and linearity (Fig. 1h). These tests are conducted jointly, are sensitive to quadratic nonlinearity, and distinguish whether deviations

from a Linear Gaussian process stem from non-Gaussian inputs or from the system's nonlinear dynamics. As summarized in Fig. 1i, this parallel assessment classifies a signal as Gaussian Linear (GL), Non-Gaussian Linear (NGL), Gaussian Nonlinear (GNL), or Non-Gaussian Nonlinear (NGNL), the latter three categories (NGL, GNL, and NGNL) all manifest as non-Gaussian signals at the observational level.

## Simulation illustrating asymmetric waveforms and quadratic nonlinearity

To build a clear intuition for the bicoherence-based tests central to the BiSCA framework, we use a targeted simulation to illustrate what they specifically detect: quadratic (or even-order) nonlinearities, which manifest as waveform asymmetry. This comparison highlights the specific features our test detects, rather than merely differentiating a linear-Gaussian process from a nonlinear one.

The analysis begins with an example of the "wicket waves"[41] from a 2.5-second iEEG segment Fig. 2a, waveforms known for their strong asymmetry. A Nonparametric Nonlinear Autoregressive (NNAR) model was

fitted to this data and pruned to define its core dynamics. This model was then used to generate two types of long-running simulations.

The right panel of Fig. 2g–k shows a simulation preserving the original asymmetry of the wicket waves. For this system, the estimated 2D potential landscape of the dynamics reveals a distinctly asymmetric basin of attraction (Fig. 2i). The shape of this potential well geometrically constrains the system's trajectory thus forcing the system into the distorted, non-uniform limit cycle shown in the 3D phase portrait (Fig. 2h). This dynamical structure generates the asymmetric morphology of the time-domain waveform (Fig. 2g), reflected in the spectrum (Fig. 2j). Note that the spectrum contains both even and odd harmonics of the fundamental frequency (around 8.7 Hz). This harmonic structure is reflected by quadratic phase coupling among the harmonics, signaled by as high-valued peaks in the bicoherence plot Fig. 2k, quantified by a bicoherence magnitude of $b^{(L)} = 1.0765$.

Conversely, the left panel (Fig. 2b–f) shows a synthetic simulation where the waveform is artificially symmetrized by enforcing odd symmetry in the model dynamics ($F(-\boldsymbol{x}) = -F(\boldsymbol{x})$, see details in supplementary). This process generates a perfectly symmetric, albeit still non-sinusoidal, waveform (Fig. 2b). The corresponding state-space representations are also symmetric. The geometry of the dynamics corresponds to a concentric 2D potential well (Fig. 2d) forcing the trajectory to trace a symmetric 3D limit cycle (Fig. 2c), which is plotted along with the pruned states (coreset, gray dots) used by the model. Critically, the spectrum of this symmetric system (Fig. 2e) contains only odd-order harmonics. Since bicoherence quantifies quadratic (even-order) interactions, the absence of even harmonics results in a statistically negligible bicoherence value (Fig. 2f), with a magnitude of $b^{(L)} = 0.0161$. This value is the measured maximum bicoherence, not a significance threshold; the corresponding threshold derived via Gumbel extreme-value calibration (Methods Section "Test for Gaussianity and linearity through Higher-Order Spectrum") is far above the observed value. This directly addresses the observation that a non-sinusoidal waveform can have minimal bicoherence if its nonlinearity is purely symmetric (odd-order). This lack of sensitivity to even harmonic coupling will not affect our conclusions since analyzing higher-order statistics would be more cumbersome and only increase the rejection of linear behavior.

These two simulations illustrate how the geometric asymmetry in the state-space representations of the estimated potential landscape and corresponding limit cycle show up in the bifrequency plane as quadratic phase coupling quantifiable by bicoherence. Such dynamic behavior is inaccessible to a linear system, which is fundamentally constrained to a symmetric, parabolic potential well, producing only an elliptical trajectory and exhibiting negligible bicoherence. This example illustrates how our test effectively identifies quadratic nonlinearities and distinguishes them from other forms of nonlinear dynamics.

## Nonlinear and non-Gaussian cortical dynamics in EEG/iEEG revealed by Bicoherence

To assess the prevalence of nonlinear and non-Gaussian dynamics in the EEG and iEEG datasets, we applied statistical tests for linearity and Gaussianity. These tests are based on Multitaper bicoherence estimates $\hat{b}(f_1, f_2)$, employing a modified Hinich test[42] The thresholds for the tests were adjusted for multiple comparisons using the Benjamini-Hochberg FDR procedure ($q = 0.001$). Each recording was classified by the joint outcome of two parallel tests—one for linearity and one for Gaussianity. These two results allow us to classify each recording into one of four categories: Gaussian Linear (GL), Non-Gaussian Linear (NGL), Gaussian Nonlinear (GNL), or Non-Gaussian Nonlinear (NGNL).

The results, summarized in Table 1, indicate widespread deviations of the data from linear Gaussian behavior. A minority of recordings were classified as GL: only 14.7% for EEG and 25.7% for iEEG. Conversely, a substantial majority—85.2% (81.6% + 3.6%) of EEG and 74.3% (67.9% + 6.4%) of iEEG recordings, respectively—were identified as either nonlinear or non-Gaussian. signals that were strictly linear but non-

Gaussian (NGL) were uncommon, accounting for only 3.6% of EEG and 6.4% of iEEG recordings.

The spatial distribution of these classifications across electrodes are revealed by topographic maps. These maps exhibit distinctive patterns (Fig. 3a, c). For EEG, linear signals (both Gaussian and non-Gaussian) are most frequently observed at central electrodes (Fig. 3a). In contrast, iEEG shows a predominance of GL signals in the frontal region, while NGL signals are more frequent in the occipital area. GNL activity is strongly expressed in the frontal areas for EEG but is less apparent in iEEG. Nonlinear Non-Gaussian (NGNL) dynamics in EEG show an anterior-to-posterior gradient, peaking at occipital electrodes, whereas in iEEG, they are concentrated centrally with a right-hemispheric asymmetry in the occipital lobes.

We use the notation $B$ for the bispectrum and $b$ for the bicoherence (see Section "Multitaper Estimator for the Bispectrum" for formal definitions). An alternative perspective is offered by plotting the proportion of test result types over the triangular bifrequency domain $(f_1, f_2)$ and $f_1 + f_2 \leqslant 50 \text{Hz}$ (corresponding to the bicoherence principal non-redundant domain. For visualization, we display the closure-constrained region[32,43,44] so all closure-valid bins remain visible; we intentionally do not additionally collapse by permutation symmetry, so that all valid frequency pairs are displayed in a single panel. The principal domain, hereafter bifrequency panel) (Fig. 3b, d). Note that here, though we deal with statistics for each $(f_1, f_2)$ pair, the threshold for deciding tests is for the family of tests over the entire triangular plane, corrected with FDR. Also, by construction, the Gaussianity threshold $c_\alpha^{(G)}$ and the linearity threshold $c_\alpha^{(L)}$ are related as $c_\alpha^{(L)} > c_\alpha^{(G)}$, for individual tests, resulting in a zero proportion of nonlinear but Gaussian results (lower left of Fig. 3b, d). A diagonal ridge of Non-Gaussian Linear test results suggests time domain asymmetries of waveforms for all frequencies. Note, however, the off-diagonal bands of interference coupling for the Alpha frequency harmonics suggest frequency doubling and intermodulation effects. Note the NGNL plot for EEG reveals a localized region of high proportion near (10 Hz, 10 Hz) associated with Nonlinearity and Non-Gaussianity. This result indicates evidence of intra-band nonlinear coupling in the alpha rhythm. The bifrequency distribution for iEEG (Fig. 3d) shows a qualitatively similar structure to the EEG results. However, these patterns in the iEEG data are more diffusely distributed over the frequency triangle compared to scalp EEG data.

It is instructive to view the topographies of the peak frequencies at which the spectrum is maximum, $f_{S_{\max}} = \text{argmax}_f(S(f))$, and the frequency at which the bicoherence is maximal, determined using the maximum of the main diagonal bicoherence $f_{b_{\max}} = \text{argmax}_{f'}(b(f', f'))$, indicates that where the frequency-doubling occurs (Fig. 3e, f). The latter is restricted to the bispectral values on the main diagonal of the principal domain. The occipital and parietal areas possess these local maxima in the alpha band, while the frontal region is in the theta band. The peak frequency of the bicoherence is higher than the peaks found by the spectrum. In terms of the nonlinear resonance framework introduced in the Introduction, frequency doubling ($f + f = 2f$, diagonal ridge) and intermodulation ($f_1 + f_2$ where $f_1 \mathrel{!}= f_2$, off-diagonal) are both visible in the bispectral maps.

## Nonlinear resonances manifest in Rho peaks, while the Xi process remains linear and Gaussian at the bispectral (quadratic) level

To understand how each process contributes to the signal's overall statistical properties, we analyzed them separately. The aperiodic Xi process consistently behaved as a linear, Gaussian process with respect to quadratic phase coupling: its Gaussianity statistic ($b_\xi^{(G)}$) remained well below the significance threshold (Fig. 4a), and its nonlinearity statistic ($b_\xi^{(L)}$) rarely exceeded the threshold (Fig. 4b).

In stark contrast, the periodic Rho processes were the primary source of quadratic nonlinearity. While being largely Gaussian (with $b_\rho^{(G)}$ below threshold), their nonlinearity statistic ($b_\rho^{(L)}$) was significantly higher than the

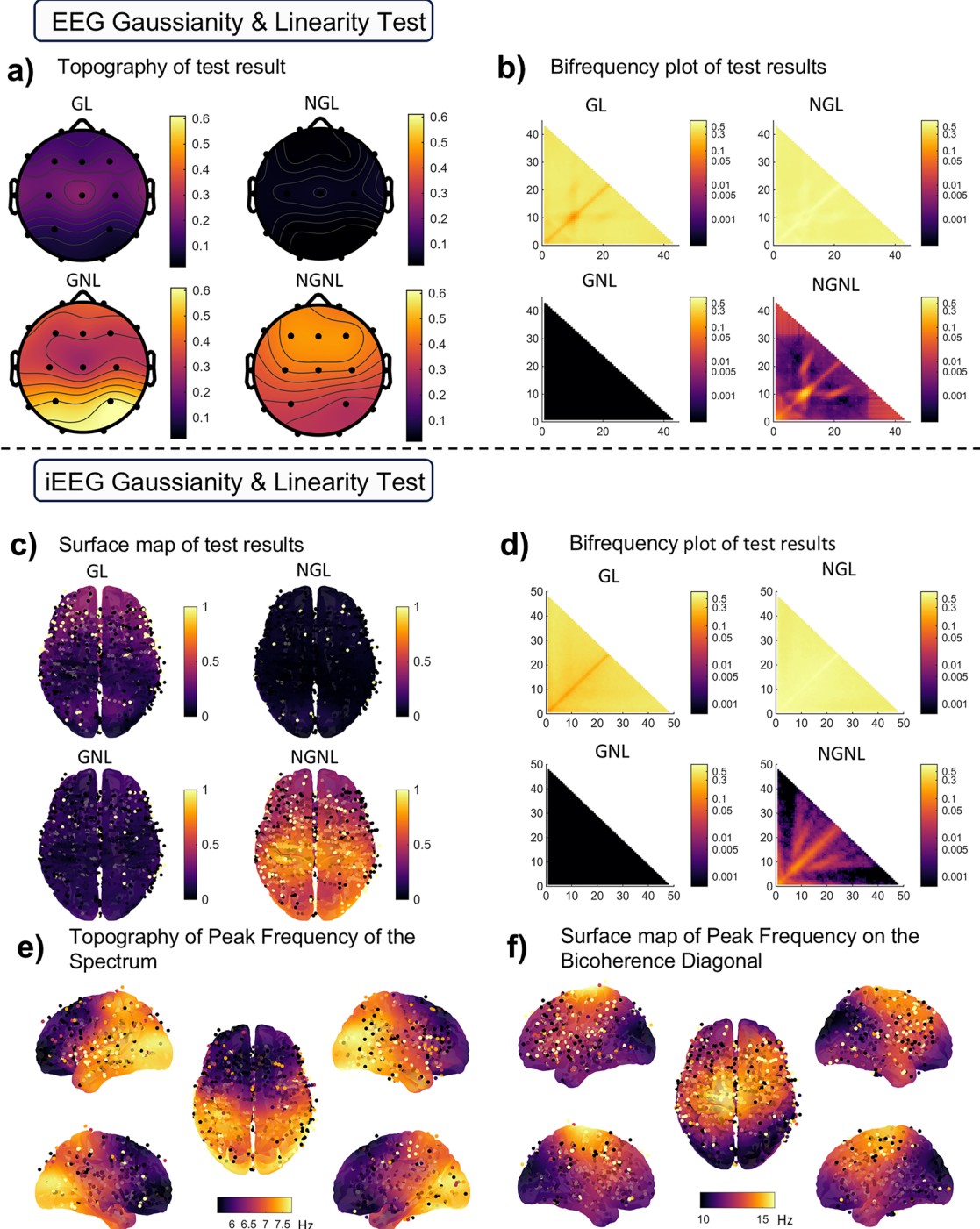

**Fig. 3 | Results of Gaussianity and linearity tests applied to iEEG and scalp EEG recordings. a** The topography map of the proportion of test classes of EEG data; (**b**) the proportion of test class of EEG along the bifrequency (Hz-Hz) panel; (**c**) the surface plot interpolated by the statistical test result (0: accept, 1: reject) of iEEG channels; (**d**) the proportion of test class of iEEG along the bifrequency panel; (**e**, **f**) the spatial distribution of the frequency for the maximum spectrum peak and bicoherence peak after removing the Xi trend.

threshold, mirroring the behavior of the full signal. These results clearly indicate that the quadratic nonlinearity detected in the composite iEEG signal is driven by its rhythmic components, not its aperiodic background.

We further examined how the statistical properties of the full signal are driven by its components. The scatterplots reveal a strong correlation between the nonlinearity of the full signal and that of the Rho processes (Fig. 4f), but a substantially weaker relationship exists for the Xi process ($r = 0.40$) compared to the Rho process ($r = 0.90$) (Fig. 4e). Similarly, the full

signal's deviation from Gaussianity is also tied to the Rho components (Fig. 4c, d). An interesting question is the extent to which tests for Linearity and Gaussianity of the original signal are influenced by those of each spectral component. The scatterplots of the tests for both linearity and Gaussianity between the full signal and its components shed light on this issue. A clear relation exists between the two types of tests for the full signal and the Rho processes (Fig. 4f). This is not the case for the Xi process (Fig. 4e). Therefore,

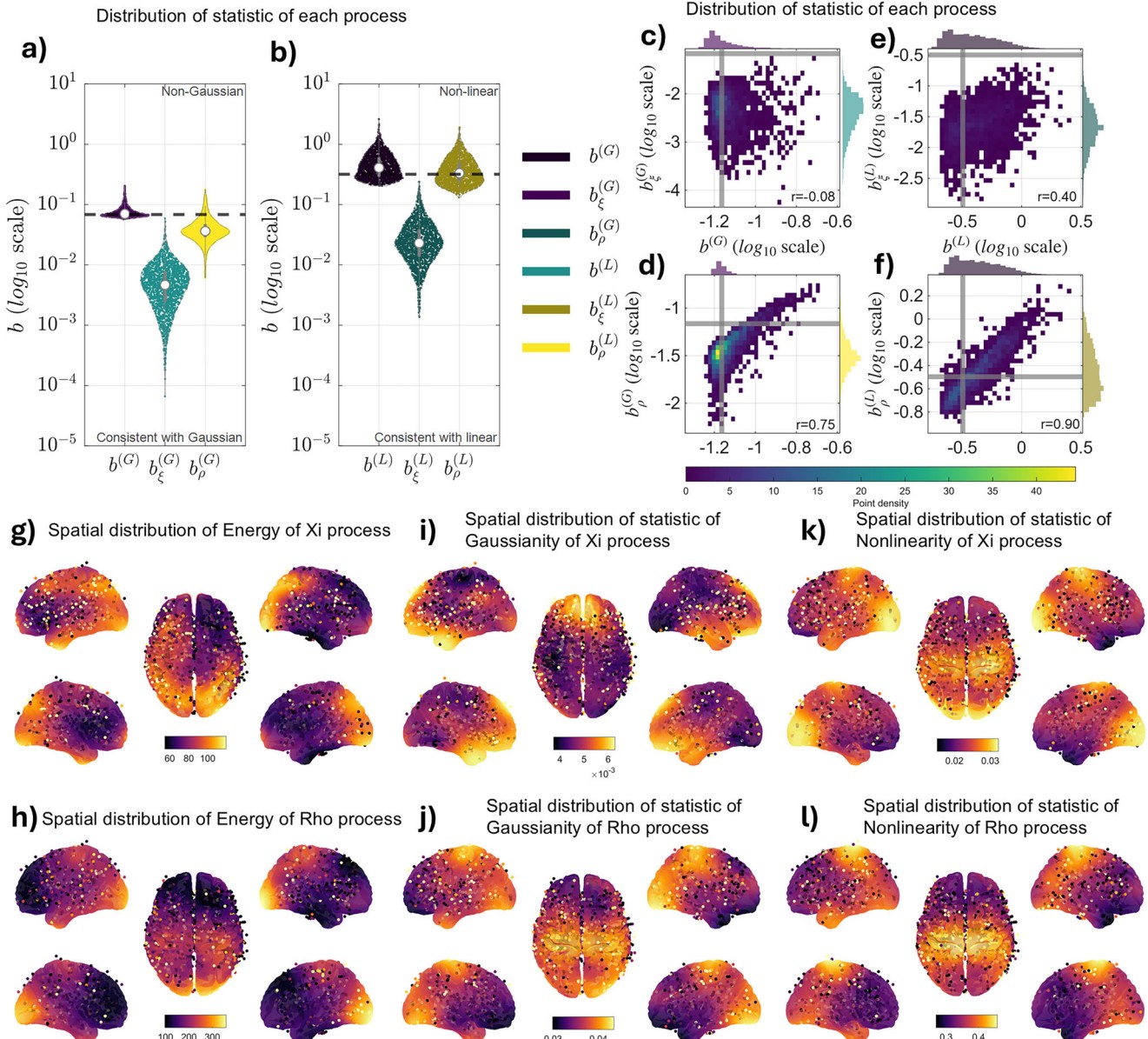

**Fig. 4 | Statistical features of iEEG signals decomposed into Xi trends and periodical Rho components. a, b** Violin plots showing the distribution of the (**a**) Gaussianity statistic $b^{(G)}$ and (**b**) nonlinearity (bicoherence) statistic $b^{(L)}$ for the full signal, the aperiodical Xi process (ξ), and the periodical Rho processes (ρ). The dashed line indicates the threshold for statistical significance. Values above this line indicate statistically significant rejection of the null hypothesis of Gaussianity (**a**) or linearity (**b**). **c–f** Scatter plots showing the relationship between the full signal and its components. Plots (**c, d**) show the correlation between the Gaussianity statistics of the full signal and the (**c**) Xi processes and the (**d**) Rho process. Plots (**e, f**) show the correlation between the nonlinearity statistics of the full signal and the (**e**) Xi

processes and the (**f**) Rho process. Pearson r values: Panel (**c**) (Xi Gaussianity vs Full), $r = -0.08$ (negligible); Panel (**d**) (Rho Gaussianity vs Full), $r = 0.75$ (strong); In panels (**c–f**), colors represent point density estimated from 2D histograms; corresponding colorbars are shown for each panel. Panel (**e**) (Xi Linearity vs Full), $r = 0.40$ (moderate); Panel (**f**) (Rho Linearity vs Full), $r = 0.90$ (very strong). **g–l** Spatial distribution maps showing key statistics for the aperiodic Xi and Rho processes. **g–h** Spatial distribution of energy. **i–j** Spatial distribution of the Gaussianity statistic $b^{(G)}$. **k, l** Spatial distribution of the nonlinearity statistic $b^{(L)}$. Red indicates higher values, while blue indicates lower values. We kept independent scales of (**g–l**) to maintain visual contrast and avoid compression of ξ-panel's scale.

the primary factors contributing to the rejection of Gaussianity and linearity in the iEEG appear to be nonlinear resonances related to the Rho processes.

Another crucial aspect of each spectral component is the spatial distribution of their statistical features across the brain (Fig. 4g–l). The energy (sum of the power spectrum over all frequencies) of the aperiodical Xi process is broadly distributed, with notable concentrations in the occipital lobe, temporal lobe, and left parietal lobe (Fig. 4g). The energy of the periodical Rho processes shows a more focal distribution, with high energy localized explicitly in the occipital lobe, corresponding to the Alpha rhythm, and an anterior-to-posterior decrease toward the prefrontal cortex with some energy in the parietal lobes corresponding to the Mu rhythm (Fig. 4h).

These energies have been harmonized across recordings to account for differences in EEG amplifiers and site conditions with the Incomplete Observed Linear Mixture Model (IOLMM) model[45].

Overall, the Gaussianity statistics are relatively low across space for both processes. Non-Gaussianity tests for the Xi process are higher in the frontal region, while the Rho processes have a higher value in the occipital and parietal regions. When observing the maps for the nonlinearity statistic the two processes have distinct spatial distributions. The Xi process is barely considered nonlinear across the entire brain (Fig. 4k) at quadratic level. Conversely, the Rho processes show notable nonlinearity in regions that also

exhibit high energy, particularly in the parietal lobes (Fig. 4l), and to a lesser degree in the occipital areas.

## Discussion

By analyzing a dataset of 1,771 intracranial EEG (iEEG) and 960 subjects of scalp EEG, we reveal that only a small minority of recordings can be described as both linear and Gaussian at the bispectral (quadratic) level. Over 81% of scalp EEG and 67% of intracranial EEG recordings deviate from this ideal, exhibiting significant nonlinearity and/or non-Gaussianity. This challenges the notion that mesoscale brain dynamics are predominantly linear and Gaussian[1,2] at the quadratic level. By decomposing the iEEG signals with BiSCA, we find a clear separation between the aperiodic background activity and rhythmic oscillatory components. With respect to quadratic phase coupling, the broadband Xi process (aperiodic component) behaves largely as a linear, Gaussian process, whereas the narrowband Rho components (oscillatory peaks, e.g., the alpha rhythm and Mu rhythm) are the primary sources of quadratic nonlinearity in the EEG. In other words, the resonant oscillations in the brain carry nearly all of the detected quadratic nonlinear interactions (as evidenced by significant bicoherence), while the aperiodic component shows minimal quadratic nonlinear coupling. While previous SPA methods have successfully characterized the power-spectral features of these components, they are blind to phase relationships and therefore cannot assess their nonlinearity.

We observed clear signs of nonlinear resonance phenomena in the data. In the bifrequency domain analysis of the bicoherence, rhythmic peaks (especially the 10 Hz alpha rhythm) exhibited harmonic coupling – for example, frequency-doubling and intermodulation effects were visible as diagonal and off-diagonal bicoherence. These results confirm that well-known cross-frequency coupling phenomena (e.g., harmonics of alpha) are present even in resting-state activity and are quantitatively detectable with our approach. With our BiSCA model we can parameterize the coupling strength between oscillatory peaks. The more diffuse bifrequency distribution in iEEG reflects greater spatial specificity: iEEG electrodes detect local nonlinear sources at distinct frequencies. Conversely, volume conduction causes each EEG channel to mix multiple sources, concentrating significance at fewer bifrequency locations.

### Linear vs. nonlinear modeling debate

The question of whether large-scale brain signals exhibit quadratic nonlinearity instead of purely linear behavior is directly addressed by our bicoherence results, which detect the presence of quadratic phase coupling at the macroscopic signal level. As mentioned before, recent studies reporting evidence that macroscopic resting-state dynamics are adequately captured by linear models suggest that microscopic nonlinearities are "averaged out" at the EEG/fMRI level. In contrast, our bicoherence tests revealed that, as shown in Table 1 and Fig. 3, 81.6% of EEG channels and 67.9% of iEEG channels exhibited significant higher-order statistical structure. Nonlinear interactions among neural population oscillations seem to persist despite both spatial and temporal averaging. In fact, our finding that a greater percentage of scalp EEG channels appeared nonlinear than iEEG channels suggests that, rather than decreasing nonlinearity as might be expected[17,46,47], our Forward-Model Simulation (Section S.3.3) provides numerical evidence that volume conduction can spread quadratic nonlinear features across the scalp, contributing to a wide array of channels (94.7% significant EEG vs. 30.9% iEEG in our forward-model simulation). For this reason, our subsequent analyses of regional nonlinearity patterns focus on iEEG. Prior work[8–15] showed that nonlinear phenomena like cross-frequency coupling and harmonic generation can occur in neural mass models and actual EEG/LFP recordings. Our findings also suggest that large-scale measures retain higher-order structure reflecting nonlinear resonance in neural circuits. This finding suggests that to capture brain dynamics fully, nonlinear terms are needed. The debate should thus move beyond the binary question of "linear vs. nonlinear models" and consider how specific nonlinear features (like those we observed in Rho oscillations) can be integrated into interpretable models of brain activity. It would

therefore be premature to assert the superiority of linear models[1] without optimally tuning and evaluating nonlinear models on an equal footing.

### Interpretation of oscillatory vs. aperiodic components

The biophysical separation[48] of components in BiSCA follows the assumption that two processes are additive and independent, deriving from different layers of the cortex[38,49–52]. A recent study provides further neurophysiological evidence for biophysical separation, further demonstrating with biophysical modeling and experimental evidence that the Xi process is a genuine signal generated by arrhythmic network activity, rather than merely an artifact of overlaid brain rhythms[38].

The analytical separation of components in BiSCA is theoretically grounded in two principles. First, for statistically independent; under this assumption, cumulant additivity yields $S_x = S_\xi + S_\rho$ and $B_x = B_\xi + B_\rho$, the cumulant spectrum (including the bispectrum) is additive[32]. Second, and more broadly, even for single nonlinear systems described by Volterra functional series[29,30], the bispectrum can be expressed as a sum of additive frequency components. Each component is analytically defined by the system's linear and quadratic transfer functions, allowing for the formal description and identification of nonlinear interactions.

A central finding of our study is that the aperiodic (Xi, ξ) component of the EEG signal—which is widely considered to exhibit a 1/f-like power law—is predominantly a linear and Gaussian stochastic process at the bispectral (quadratic) level. At first glance, this observation may appear to challenge established opinions that the aperiodic component has a 1/f spectrum. If true, this would tie the Xi process to critical, avalanche-like processes. While "1/f" is widely used, empirical exponents range from 0 to 4 and vary with frequency band —even parametric frameworks require a knee parameter for low-frequency deviations[34], and nonparametric decomposition reveals frequency-dependent aperiodic exponents and irregular spectral shapes[53]. The precise functional form thus remains an open model-selection question. On the contrary, in our findings, the Xi process showed negligible bicoherence and near-Gaussian amplitude statistics across the cortex consistent with fitted bispectral amplitudes $h_{B,\xi} \approx 0$ (see Methods Eq. 15). The predominance of GL classification in frontal iEEG reflects weaker alpha-band power in frontal cortex, rather than an absence of complex dynamics. Conversely, the NGL classification in occipital cortex arises from the alpha rhythm's marginal non-Gaussianity paired with its relatively symmetric waveform, which does not produce sufficient quadratic phase coupling. This suggests that the aperiodic background can be modeled as a superposition of many independent or weakly interacting sources (consistent with a linear Gaussian process). A nonparametric decomposition[53] avoids imposing a fixed parametric law on the aperiodic component, in line with our treatment of Xi. The bispectral linearity of the Xi component can thus be understood as an emergent property, likely arising from the superposition of numerous weakly correlated neural sources where nonlinearities are averaged out at the macroscopic scale. This empirical finding is supported by recent theoretical models such as that of Kramer & Chu [39], who demonstrated that a general, noise-driven dynamical system, when linearized around a stable equilibrium, naturally generates spectra with 1/f-like characteristics. Early research by this group (Valdes et al. 1990)[54,55] on the spatial structure of the Xi process used frequency domain tests and found that the aperiodic cross-spectrum matched a brain eigenmode pattern characteristic of an isotropic random process. It was proposed that such a process may result from short-range cortical neural interactions with exponential decay. This provides a framework for understanding the Xi process throughout the cortex. This mechanism is conceptually identical to a basic linear filter in control theory, whose characteristic Bode plot slopes produce such power-law spectra, offering a parsimonious explanation. It provides a compelling theoretical foundation for why a macroscopically linear stochastic process can produce the observed aperiodic background without invoking complex nonlinear mechanisms. It is also crucial to consider the specificity of our methodology; the bicoherence is a powerful tool for detecting quadratic nonlinearity, and our results show a definitive lack of such second-order interactions in the Xi component. While we cannot formally exclude other, more complex forms

of nonlinearity, our work robustly refines the debate on neural linearity. Whether the Xi process is predominantly linear and Gaussian beyond quadratic phase coupling remains a hypothesis to be confirmed with higher-order statistical tests (e.g., the trispectrum).

In contrast, the Rho components (distinct spectral peaks corresponding to oscillatory neural generators) display robust nonlinear signatures. For the alpha rhythm (around 10 Hz) and other peaks, we found significant bicoherence indicating phase-locked interactions, such as harmonics (e.g., 20 Hz, 30 Hz components related to alpha) and possible subharmonics or cross-frequency coupling with other bands. The fact that the full EEG signal's nonlinearity closely tracked that of the Rho component (and not Xi) implies that oscillatory networks are the location of nonlinear resonant dynamics. The resonances detected seem to correspond mainly to thalamocortical loops in posterior cortices for alpha, and sensorimotor cortices for mu. This is also consistent with corticothalamic neural field models, which generate bursts of nonlinear alpha oscillations through noise-driven exploration of a subcritical Hopf bifurcation[56]. The Valdes et al. (1990)[54,55] paper found that spatial isotropy was not present at the frequencies corresponding to the spectral peaks, supporting the conclusion that nonlinear resonances are more locally distributed on the cortex. The close correspondence between full-signal and Rho bicoherence confirms that the oscillatory component drives the nonlinear structure of the composite signal.

**Dissociation between power and nonlinearity**. An interesting observation is the mismatch between the cortical distributions of oscillatory power and degree of nonlinearity. This spatial dissociation suggests that a high-power spectrum of oscillation does not relate systematically to strong nonlinearity. Consider the posterior alpha rhythm, with high power but weaker nonlinearity than the mu rhythm. There is evidence that in usual recordings of the rhythm, there is a mixture of intermittent, high-amplitude bursts of nonlinear activity interspersed with periods of low-amplitude, quasi-linear dynamics close to a stable fixed point[57]. Such bistable dynamics can emerge from models near a subcritical Hopf bifurcation[56,58], would yield high spectral power due to the amplitude of the bursts, but the bicoherence, when averaged over a long time window, would be diluted by the intervening linear periods. In contrast, the less powerful parietal mu rhythm may operate in a more persistently nonlinear state, resulting in a disproportionately stronger bicoherence signature. This observation is consistent with a prior study[59], which reported that there are power peaks that coexist in the Alpha and Beta bands in the parietal lobe, while only peaks in the Alpha band appear in the occipital lobe. Our BiSCA method allows us to specifically pinpoint functionally distinct sources of nonlinearity. A similar EEG source-reconstruction study[60] also describes this dissociation between the spatial patterns of power and bicoherence. It seems that different neural circuits and networks have different dynamical regimes which are projected onto the statistical signatures at the EEG. Some thalamic circuits operate closer to linearity and have high power, while others operate more nonlinearly but have lower power.

**Methodological limitations**
The scalp EEG dataset we used includes recordings from 960 healthy subjects, whereas the iEEG dataset comprises 1771 channels from 106 patients. The nature of these datasets should be considered when interpreting the results. The iEEG recordings come from presurgical epilepsy patients, which might have influenced the prevalence or distribution of nonlinear dynamics (e.g., due to cortical irritability or medication effects). However, we observed consistent patterns between iEEG and healthy scalp EEG. In addition, these patterns are also consistent with a prior study in healthy subjects[60] suggesting that our conclusions are unlikely to be driven primarily by pathological factors.

Another consideration is volume conduction and referencing in scalp EEG, which can smear true source interactions. We mitigated this by focusing many analyses on iEEG, where signals are more focal, and by using a conservative statistical threshold (FDR $q = 0.001$). Still, some ambiguity in the interpretation of scalp recordings cannot be entirely ruled out. Our concern here is not multivariate mixing between sensors, but rather the univariate superposition of multiple sources at a single sensor. If multiple independent sources with different frequencies are superposed at a single sensor, interpretation of the observed bicoherence becomes more difficult, because the measured signal may reflect overlapping contributions from multiple generators rather than a single source in isolation. Source localization combined with BiSCA could help isolate individual generators and clarify whether the observed nonlinearity is intrinsic to a specific generator or reflects source overlap. Laplacian re-referencing and source-space bispectral imaging therefore represent important future directions for extending bispectral analysis to scalp EEG.

The current analysis focuses on the eyes-closed resting state. Comparing eyes-open and eyes-closed conditions would provide additional insight into state-dependent nonlinearity. In addition, the current BiSCA parametric model focuses on discrete harmonic coupling. Extending the model to wideband cross-frequency coupling patterns, such as theta-gamma phase-amplitude coupling, would require continuous Volterra kernel estimation and represents an important direction for future work.

**Implications**
**Nonlinear resonance as an organizing principle**. We found that the dominant EEG exhibits nonlinearity. This finding agrees with copious theoretical work that models these oscillations with nonlinear models. Explicitly linking the spectral "peaks" to bispectral evidence of coupling, we highlight the importance of nonlinear resonance in its more general formulation. We envisage that our methods may contribute to deal with disparate phenomena like EEG cross-frequency coupling, nested oscillations (e.g., phase–amplitude coupling), and event-related harmonic responses within a unified resonance-based theory. It also supports methods like ICA or nonlinear signal decompositions, which concentrate on the nonlinear and non-Gaussian aspects of the EEG.

**Refining macroscale models**. In practical terms, our findings encourage the development of new macroscale brain models that incorporate nonlinearity with the prior knowledge provided in this paper in a controlled way[61–68]. Such models might treat the aperiodic background as an additive Gaussian noise process (as linear models do) while nonlinear coupling would be reserved for oscillatory subsystems. This hybrid perspective – essentially what our data-driven BiSCA components represent – could improve how we simulate or predict EEG dynamics. For example, large-scale neural mass or neural field models might include nonlinear feedback only in certain frequency-generating circuits (like alpha or mu generators), which could reproduce the selective nonlinearity we observed. Formal validation of the statistical independence between Xi and Rho components--for example, via mutual information applied to laminar recordings, where distinct oscillatory generators occupy different cortical layers[49–51] and aperiodic determinants vary independently from spectral peaks[38] is planned for future work. BiSCA or methods like it might provide data-driven statistics that neural models should match.

**Extend the BiSCA to non-stationary EEG/iEEG.** BiSCA analyzes the system's quadratic nonlinearity (bispectrum/bicoherence) based on the assumption of stationarity. This is a condition that the providers of the data tried to ensure. Nevertheless, brain activity is widely considered to exhibit non-stationarity. It is clear that phenomena such as multistability indicate dynamic changes in the nonlinear properties[56,57,69] in EEG data, which shows that a linear system with a single global attractor is an insufficient model. The BiSCA approach is a tool that assumes a stationarity complementary perspective. We must extend BiSCA to identify persistent quadratic interactions from those that change dynamically[70].

## Materials and methods

### Materials

We analyzed resting-state EEG data from 960 healthy participants drawn from the globally diverse HarMNqEEG dataset[71]. Participants were selected from normative or control groups, excluding those with neurological/psychiatric conditions. EEG recordings consisted of eyes-closed resting-state data. For all subjects, 19 channels were included from the 10–20 system after average referencing. Samples recorded for less than 1 minute, such as the Medicid-4-Cuba-2003, were excluded. Preprocessing involved converting signals to an average reference, segmenting data into 300 time points (1.5 seconds), and overlapping segments to have 157 segments per channel to maintain consistency with the iEEG data length and segmentation. Segments were retained only if artifact-free, with no interictal discharges or slow-wave anomalies.

For Resting-state Intracranial EEG (iEEG), we used the dataset of eyes-closed resting-state wakefulness recordings, sEEG/ECoG recordings from the Montreal Neurological Institute (MNI) open iEEG atlas[59]. These iEEGs were recorded from 106 patients with refractory focal epilepsy (1771 bipolar channels; we reconciled channel counts as 1785 (published atlas) to 1772 (public release file) to 1,771 (final analysis subset after excluding one single-channel subject)), ensuring that tracings retained were from non-lesional, non-seizure onset zones, free of interictal discharges, slow-wave anomalies, or post-seizure/stimulation effects. Preprocessing involved bandpass filtering (0.5–80 Hz), down-sampling to a common sampling rate of Fs = 200 Hz, and removing zero values. The study focused on 60-second eyes-closed recordings, in order to be consistent with the EEG data set.

### Method

**Multitaper estimator for the spectrum**. Each segment of the signal was subjected to a Discrete Fourier transform $X(f) = \frac{1}{N_t} \sum_{n=0}^{N_t-1} x_k(n)e^{-j2\pi fn}$, where $N_t$ is the number of time points for each epoch. To balance the frequency resolution and computational resources, in our study, the window size was 300 time points (1.5 seconds) segments. To estimate the power spectrum, $S(f) = \mathrm{E}\left[X(f)X^*(f)\right]$ we used overlapped segments by 75% and also applied the Multitaper method, which creates windowed versions of the original time signal $x_k(n) = x(n)v_k(n)$ [72,73] in order to reduce frequency leakage bias and increase the consistency of the spectral estimate, $k$ is the index of taper. For this purpose, we used the Sine tapers $v_k$ [74] to diminish amplitude bias further. The resulting estimation procedure is then:

$$X_{k,e}(f) = \frac{1}{N_t} \sum_{n=0}^{N_t-1} x_{k,e}(n)e^{-j2\pi fn} \tag{1}$$

$$\hat{S}_k(f) = X_{k,e}(f)X_{k,e}^*(f) \tag{2}$$

$$\hat{S}(f) = \frac{1}{U_2} \sum_{k=0}^{K-1} \hat{S}_k(f) \tag{3}$$

where $U_2 = K/N_t$. In this paper, we use $NW = 1.5$, i.e., $K = 2NW - 1 = 2$ for the recordings. $NW$ controls the estimator's trade-off between frequency resolution and variance.

**Multitaper estimator for the bispectrum**. We apply the theory developed by Brillinger for nth order cumulant spectra[43,75]. The usual power spectrum is the 2nd-order cumulant spectrum we just presented above. In this paper, we also deal with the bispectrum, which is the 3rd-order cumulant spectrum, a function of the two frequencies (bifrequency) $f_1$ and $f_2$

$$B(f_1,f_2) = \mathrm{E}[X(f_1)X(f_2)X^*(f_1+f_2)] \tag{4}$$

As mentioned above, the bispectrum has the favorable properties of (i) being zero for Gaussian processes, (ii) having constant (but non-zero) bicoherence across frequencies for linear systems with non-Gaussian input, and (iii) exhibiting frequency-specific peaks that reflect quadratic nonlinear interactions between bifrequency pairs $f_1$ and $f_2$ and their sum $f_1 + f_2$.

The Multitaper estimator for the bispectrum and bicoherence[76,77]

$$\hat{B}_{k_1 k_2 k_3}(f_1,f_2) = X_{k_1,e}(f_1)X_{k_2,e}(f_2)X_{k_3,e}^*(f_1+f_2) \tag{5}$$

$$\hat{B}(f_1,f_2) = \frac{1}{U_3} \sum_{k_1 k_2 k_3=0}^{K-1} P_{k_1 k_2 k_3}\hat{B}_{k_1 k_2 k_3}(f_1,f_2) \tag{6}$$

To preserve the correct scale, the 'bienergy' $P_{k_1 k_2 k_3} = \sum_{t=0}^{N_t-1} v_{k_1}(t)v_{k_2}(t)v_{k_3}(t)$ and $U_3 = \frac{1}{N_t} \sum_{k_1,k_2,k_3}^{K} P_{k_1 k_2 k_3}^2$.

We estimate the bicoherence $b(f_1,f_2)$ using the following normalization of the bispectrum[78], which benefits the statistical tests defined in the next section:

$$b(f_1,f_2) = \frac{B(f_1,f_2)}{\sqrt{S(f_1)S(f_2)S(f_1+f_2)}} \tag{7}$$

**Test for Gaussianity and linearity through higher-order spectrum**. To characterize the underlying dynamics of the time-series data, we employed higher-order spectral analysis to test for linearity and Gaussianity. Linear and nonlinear systems are fundamentally distinguished by their properties of response to input frequencies: a linear system processes frequencies independently, whereas a nonlinear system generates new harmonic and intermodulation components through frequency interactions. We distinguish between the bispectrum $B$ (unnormalized, complex), the normalized bispectrum $b$ (complex), and the bicoherence $|b|$ (magnitude, real, bounded in [0,1]). Bicoherence is utilized to detect and quantify this quadratic phase coupling. Following the Brillinger and Hinich works, we assume the input (innovation) is a zero-mean i.i.d. process. Under this assumption, a linear system produces constant bicoherence, while frequency-dependent peaks indicate nonlinear phase coupling.

The theoretical foundation for this method is predicated on the behavior of the bicoherence for a linear time-invariant (LTI) system. For an LTI system driven by a non-Gaussian input, the magnitude of the bicoherence, $|b(f_1,f_2)|$, is constant across all frequency pairs and is determined solely by the input signal's third-order ($\gamma_3^x$) and second-order ($\gamma_2^x$) cumulants[44]:

$$|b(f_1,f_2)| = \frac{|\gamma_3^x|}{(\gamma_2^x)^{3/2}}$$

This property yields a clear framework for hypothesis testing, used in Table 1:

1. A **Gaussian Linear (GL) process** is the null case. It has a third-order cumulant of zero, resulting in a bicoherence that is identically zero across all frequencies.
2. A **Non-Gaussian Linear (NGL) process** is driven by a non-Gaussian input passing through a linear system. This results in a **diffuse, widespread elevation** of bicoherence across the entire frequency plane.
3. A **Gaussian Nonlinear (GNL) process** involves a Gaussian input passing through a nonlinear system. The nonlinearity generates phase coupling at specific frequency interactions, producing **sharp, localized peaks** in the bicoherence plot against a near-zero background.
4. A **Non-Gaussian Nonlinear (NGNL) process** represents the combined case where a non-Gaussian input drives a nonlinear system. Its bicoherence exhibits both localized peaks superimposed on a diffuse, non-zero background.

To achieve a more nuanced understanding of underlying dynamics and explicitly separate system nonlinearity from input non-Gaussianity, this refined framework explicitly separates all four cases (GL, NGL, GNL, NGNL,

NGNL; see Supplementary Section S.6 for the complete mathematical derivations including the colored-noise cascade extension). The test for intrinsic non-Gaussian input assesses the diffuse, central tendency of bicoherence, whereas the test for system nonlinearity targets sparse, high-magnitude peaks indicative of quadratic phase coupling. It is important to note that the term "non-Gaussian" within this framework refers to any deviation from Gaussianity in the observed time series that is not solely a result of system-induced phase coupling. This includes both non-Gaussian properties of the system's innovations (inputs) and the presence of independent additive non-Gaussian noise, both of which manifest as a diffuse, non-zero bicoherence.

The statistical implementation of these tests relies on the asymptotic distribution of the bispectrum estimator $\hat{B}(f_1, f_2)$. For a zero-mean stationary process, the bispectrum asymptotically follows the complex Gaussian distribution, its expected value and variance are given[75]

$$
\begin{aligned}
\mathrm{E}\big[\hat{B}(f_1, f_2)\big] &= B(f_1, f_2) \\
\mathrm{Var}\big[\hat{B}(f_1, f_2)\big] &= S(f_1)S(f_2)S(f_1 + f_2)
\end{aligned}
\tag{8}
$$

Statistical tests for (quadratic) Gaussianity and linearity can be effectively formulated using bicoherence, which measures quadratic phase coupling. The traditional test proposed by ref. 42 relies on the sum of squared bicoherence values, comparing the result to a chi-squared distribution. However, this sum-based approach is overly sensitive to extreme values, which can arise from strong, localized nonlinear interactions like narrow-band phase coupling. This sensitivity can cause the test to reject the null hypothesis of Gaussianity even when the process only exhibits isolated nonlinearity, leading to false positives. The sum is easily dominated by a few large values, misrepresenting the overall distribution as non-Gaussian.

For testing linearity specifically, traditional methods like the interquartile range (IQR) test are often inadequate. The IQR focuses on the central 50% of the bicoherence distribution, making it a poor detector of narrow-band quadratic phase coupling. Such interactions manifest as sparse, extreme values in the bicoherence map, which have little impact on the IQR. Consequently, the test lacks the power to identify significant but isolated nonlinear peaks.

To address these limitations, we propose two robust tests that enhance the Hinich framework by targeting distinct aspects of deviation from the null hypothesis. The first is a median-based test to assess central tendency for Gaussianity, and the second is a maximum-based test to detect nonlinear interactions. Both tests leverage bicoherence's ability to measure phase coupling, with their statistical properties validated through the theoretical derivations below.

**Median-Based Test for Gaussianity.** To provide a robust assessment of Gaussianity that is less affected by outliers, we introduce a test focused on the median of the squared bicoherence values. We first define a normalized statistic, $b^{(G)}$, which represents the median bicoherence magnitude and serves as a measure of effect size for asymmetricity:

$$
b^{(G)} = \sqrt{\operatorname*{median}_{f_1, f_2}\{|b(f_1, f_2)|^2\}}
$$

The formal test statistic, $t^{(G)}$, is a scaled version of this quantity:

$$
t^{(G)} = 2N_s^{\mathrm{eff}} \cdot \big(b^{(G)}\big)^2 = 2N_s^{\mathrm{eff}} \cdot \operatorname*{median}_{f_1, f_2}\{|b(f_1, f_2)|^2\}
$$

where $N_s^{\mathrm{eff}}$ is the effective number of segments (see Section "Estimation of the Effective Number of Segments" below for estimation procedure), and the median is computed over all bifrequency pairs $(f_1, f_2)$. Under the null hypothesis of Gaussianity and linearity, this statistic follows an asymptotic normal distribution. The test evaluates:

$$
P\big\{t^{(G)} > c_\alpha^{(G)}\big\} \approx \alpha
\tag{9}
$$

where $c_\alpha^{(G)}$ is the critical value from the asymptotic normal distribution at the significance level $\alpha$. This test excels at detecting non-Gaussianity.

This test is derived from the asymptotical normal distribution of bicoherence. Under the null hypothesis of Gaussianity, for each independent bifrequency pair, the scaled squared bicoherence $x = 2N_s^{\mathrm{eff}}|b(f_1, f_2)|^2$ follows a chi-squared distribution with two degrees of freedom $x \sim \chi^2(2)$. The test statistic $t^{(G)}$ is the sample median of $P$ such independent variables. For a large number of bifrequency pairs $P$, the sample median is asymptotically normal. The population median of a $\chi^2(2)$ distribution is $m = 2\ln2 \approx 1.386$. This is the expected value of our test statistic, $E[t^{(G)}] \approx 2\ln2$. The asymptotic variance of a sample median from a sample of size $P$ is given by $\frac{1}{4P[f(m)]^2}$, where $f(m)$ is the value of the probability density function (PDF) at the population median $m$. The PDF of a $\chi^2(2)$ distribution is $f(x) = \frac{1}{2}e^{-x/2}$. At the median $m = 2\ln2$, the PDF value is $f(2\ln2) = \frac{1}{2}e^{-\ln2} = 1/4$. Therefore, the variance is:

$$
\mathrm{Var}(t^{(G)}) = \frac{1}{4P(1/4)^2} = \frac{4}{P}
$$

Thus, the test statistics follow the asymptotic distribution:

$$
t^{(G)} \sim \mathcal{N}\left(2\ln2, \sqrt{\frac{4}{P}}\right)
$$

**Maximum-Based Test for Nonlinearity.** To identify nonlinear interactions that may be missed by the median test, we introduce a test based on the maximum squared bicoherence value. The normalized statistic, $b^{(L)}$, directly captures the magnitude of the strongest quadratic phase coupling:

$$
b^{(L)} = \max_{f_1, f_2}\{|b(f_1, f_2)|\}
$$

The corresponding test statistic, $t^{(L)}$, is its scaled squared value:

$$
t^{(L)} = 2N_s^{\mathrm{eff}} \cdot \big(b^{(L)}\big)^2 = 2N_s^{\mathrm{eff}} \cdot \max_{f_1, f_2}\{|b(f_1, f_2)|^2\}
$$

To distinguish localized linearity from background non-Gaussianity, we estimate a non-centrality parameter $\hat{\lambda}_0$. The distribution of $t^{(L)}$ is modeled using extreme value theory, specifically a Gumbel distribution. The test evaluates:

$$
P\big\{t^{(L)} > c_\alpha^{(L)}\big\} \approx \alpha
\tag{10}
$$

where $c_\alpha^{(L)}$ is the critical value from the Gumbel distribution at the significance level $\alpha$. This test is designed to pinpoint significant localized nonlinearity.

To account for baseline non-Gaussianity, we estimate a non-centrality parameter $\lambda_0$:

$$
\hat{\lambda}_0 = \max\left(\operatorname*{median}_{f_1, f_2}\{2N_s^{\mathrm{eff}}|b(f_1, f_2)|^2\} - 2\ln2,\ 0\right)
$$

This reflects that the median of a non-central $\chi^2(2, \lambda_0)$ distribution is approximately $2\ln2 + \lambda_0$ for small $\lambda_0$. Using extreme value theory (EVT), the maximum of $P$ independent $\chi^2(2, \lambda_0)$ variables is approximated by a Gumbel distribution. The normalizing constants are: $b_P = F^{-1}(1 - 1/P)$, the value where the cumulative distribution function (CDF) $F$ of $\chi^2(2, \lambda_0)$ reaches $1 - 1/P$. And $a_P = \frac{1}{Pf(b_P)}$, where $f$ is the PDF of $\chi^2(2, \lambda_0)$. Thus, the nonlinearity test statistic is distributed as:

$$
t^{(L)} \sim \mathrm{Gumbel}(b_P, a_P)
$$

**Estimation of the Effective Number of Segments.** The theoretical distributions described above assume that bicoherence values are derived from independent data segments. However, modern spectral estimation techniques like the multitaper method we used here and the use of overlapping segments introduce dependencies. This means the nominal number of segments, $N_s$, is larger than the effective number of independent segments, $N_s^{\text{eff}}$. Using the nominal $N_s$ in the test statistics will lead to incorrect scaling, thresholds, and $p$-values.

To ensure accuracy, we estimate $N_s^{\text{eff}}$ with the following procedure:

1. Fit an AR model: Fit an autoregressive (AR) model to the observed time series $x(t)$ using the Burg method. Select the optimal model order using the Corrected Akaike Information Criterion (AICc; across the dataset, the selected oscillatory-component count is typically K = 3–6).
2. Generate surrogate data: Generate $M$ surrogate time series (e.g., $M = 100$) using the fitted AR model with Gaussian white noise as innovations. Each series should have the same length $N$ as the original data.
3. Compute bicoherence: For each of the $M$ surrogate series, compute the bicoherence $b(f_1, f_2)$ using the exact same parameters (e.g., Multitaper settings, window length, overlap) as used for the original data.
4. Calculate the mean squared bicoherence: For each surrogate series, calculate the mean of the squared bicoherence magnitudes over all $P$ valid bifrequency pairs:

$$\text{mean}(|b(f_1, f_2)|^2) = \frac{1}{P} \sum_{f_1, f_2} |b(f_1, f_2)|^2$$

5. Estimate $N_s^{\text{eff}}$: Average the result from step 4 across all $M$ simulations. Under the null hypothesis, $E[|b(f_1, f_2)|^2] \approx 1/N_s^{\text{eff}}$ Therefore, the effective number of segments can be estimated as:

$$N_s^{\text{eff}} = \frac{1}{\text{mean}_M \left[ \text{mean}(|b(f_1, f_2)|^2) \right]}$$

This estimated $N_s^{\text{eff}}$ should replace $N_s$ in all calculations for the $t^{(G)}$ and test statistics and their corresponding distributions, thereby accounting for dependencies and ensuring the validity of the statistical tests.

**BiSpectral EEG Component Analysis (BiSCA).** After obtaining the empirical and higher-order spectra, this paper proposed a BiSCA model specifically designed to detect nonlinear resonance by jointly parametrizing the spectrum and bispectrum. This combined model approach is essential for identifying nonlinear resonance phenomena that cannot be detected by traditional spectral analysis alone. Here, the Xi trend and Rho peaks (including Alpha and all other oscillatory peaks) are modeled with two types of independent processes, allowing them to be additive on the natural scale. Additionally, the analysis includes both the spectrum and the bispectrum to understand the components comprehensively.

$$S\left(f; \theta_{2,\xi}, \theta_{2,\rho}\right) = S_\xi(f; \theta_{2,\xi}) + S_\rho\left(f; \theta_{2,\rho}\right) \quad (11)$$

$$B(f_1, f_2; \theta_3) = B_\xi(f_1, f_2; \theta_{3,\xi}) + B_\rho\left(f_1, f_2; \theta_{3,\rho}\right) \quad (12)$$

where different $\theta$ are the parameters for different components in each order.

In this paper, we put all peaks in the group of Rho processes for several reasons. (1) To be consistent with the conventional Xi-Alpha model[33], we model the trend as the Xi process and the oscillatory rhythms as the Alpha process. (2) The models of oscillation peaks have similar structures. (3) We focus on univariate analysis, which views the neural masses as a single integrated oscillator. For the spectra, each component consists of a sum of squares of kernel functions. Since here we use the transfer function to connect different orders of spectra, the sum of the squares of the frequency

transfer function of each component approximates the modeled theoretical power spectra here

$$S_\xi\left(f; \sigma_\xi, \nu_\xi, d_\xi\right) = h_{S,\xi} t\left(f; 0, \sigma_\xi, \nu_\xi, d_\xi\right)^2$$
$$S_\rho\left(f; \sigma_\rho, \nu_\rho, d_\rho\right) = \sum_{k=1}^{K} h_{S,\rho,k} t\left(f; \mu_{\rho,k}, \sigma_\rho, \nu_\rho, d_\rho\right)^2 \quad (13)$$

The kernel function $t(\cdot)$ models the oscillatory rhythm peaks in the spectrum domain. The $h$ is the scale parameter for each peak, and the peak number $K$ represents the number of harmonics to be modeled. We offered two strategies for determining the center frequency of peaks: (1) the $\{\mu_{\rho,k}\}$ is $\{\cdots, \frac{1}{3}\mu_\alpha, \frac{1}{2}\mu_\alpha, \mu_\alpha, 2\mu_\alpha, 3\mu_\alpha, \cdots\}$, in practice, the order $k$ is chosen with the Akaike Information Criterion (AIC) see detail in Supplementary S.4.2. The search grid spans subharmonic and harmonic orders, yielding candidate models with 1 to 11 peaks; AIC typically selects 3 to 6 peaks. This circumstance models the alpha rhythm with the simplest condition that the resting-state EEG is the output of the system excited with Gaussian white innovation, and the 1st-order transfer function has a single tone at Alpha; the rest of the peaks in other bands are the harmonics of the fundamental oscillation peak. (2) fit each of the $\{\mu_{\alpha,k}\}$ through the spectra and then adjust with the joint fitting with the bispectrum. Option (1) offers frequency peak organization following the conventional broadband analysis; option (2) will fit the components better but is more complicated. This paper demonstrates the result with option (2), which is more general and can use the parameter for bicoherence to interpolate the phase relationship between the peaks. The goodness of both option fit is shown in the supplementary.

The student t kernel function has been used in previous studies of spectrum fitting[33,79,80]. Here, we modified the student t function to model the amplitude of the 1st-order frequency transfer function, which also extends the adjustable heavy-tail version of the Lorentzian (Cauchy-Lorentz distribution function).

$$t(f; \mu, \sigma, \nu, d) = \left(1 + \left|\frac{f - \mu}{\sigma}\right|^d\right)^{-\nu} \quad (14)$$

Peak centers and amplitudes are peak-specific. Shape parameters are shared across all oscillatory peaks, reflecting a common kernel profile and reducing free parameters. The parameter $\sigma$ controls the width of the kernel function. The parameter $d$ controls the flat top of the kernel since the Multitaper method is smooth in the frequency domain and distorts the shape function; in practice, the value is larger than 2. The parameter $\nu$ controls the kurtosis or the heavy tail of the peak. Special case: $d = 2$ it turns to the student t function; $\nu = 1, d = 2$ it turns to the Lorentzian kernel.

To simplify the modeling process, we did not directly use the theoretical composition form of transfer function kernels to each order of the spectrum[29–31,81,82]. Instead, we focus on capturing the primary relationships of position and order by a pair-indexed parametrization of the theoretical bispectrum on the closure manifold. Under the harmonic constraint, the closure condition reduces the triple index to a sparse set of admissible pairs[32,43,44]. The bispectrum is parameterized $M_3(f_1, f_2)$ as follows:

$$
\begin{aligned}
\text{Re}[M_3(f_1, f_2)] =\ & \text{Re}(h_{B,\xi}) t_2\left(f_1, f_2; \mu_\xi, \mu_\xi, \sigma_\xi, \nu_\xi, d_\xi\right) && '\xi' \\
& + \sum_{m=1}^{K} \sum_{n=1}^{m} \text{Re}(h_{B,\rho,m,n}) t_2\left(f_1, f_2; \mu_{\rho,m}, \mu_{\rho,n}, \sigma_\rho, \nu_\rho, d_\rho\right) && '\rho' \\
\text{Im}[M_3(f_1, f_2)] =\ & \text{Im}(h_{B,\xi}) t_2\left(f_1, f_2; \mu_\xi, \mu_\xi, \sigma_\xi, \nu_\xi, d_\xi\right) \\
& + \sum_{m=1}^{K} \sum_{n=1}^{m} \text{Im}(h_{B,\rho,m,n}) t_2\left(f_1, f_2; \mu_{\rho,m}, \mu_{\rho,n}, \sigma_\rho, \nu_\rho, d_\rho\right)
\end{aligned}
$$

$$(15)$$

This formula captures the essential features of the bispectral peaks without resorting to complex transfer function compositions. A complete step-by-step derivation showing how this pair-indexed parametrization

arises from the general triple-indexed bispectral expansion is provided in Supplementary Section S.5.2, including worked examples for K = 2 and K = 6, reconciliation of parameter counts, and an independent empirical verification using the full three-dimensional bispectral field. The Xi and Rho processes are assumed statistically independent, which is a widely adopted assumption in the periodic/aperiodic decomposition literature. Under independence, cumulants of all orders decompose additively: the power spectrum and bispectrum each equal the sum of the respective Xi and Rho contributions. The BiSCA models the real and imaginary parts of the bispectrum separately since the phase information of the bispectrum is necessary for the discrimination of systems[83]. Each peak pair (m,n) has an independent complex amplitude, allowing the model to capture heterogeneous coupling strengths across the bispectrum. In particular, the aperiodic bispectral amplitude $h_{B,\xi}$ is a free complex parameter: if the Xi process were non-Gaussian linear (driven by noise with nonzero third-order cumulant $\gamma_3 \neq 0$), the fitted $h_{B,\xi}$ would absorb the resulting flat bispectral contribution. The Gaussian linear (GL) interpretation—$h_{B,\xi} \approx 0$—is therefore a data-driven result of the fit, not a structural constraint of the model (see Supplementary Section S.6.4). The kernel function for the bispectrum is

$$t_2(f_1, f_2; \mu_1, \mu_2, \sigma, \nu, d) = t(f_1; \mu_1, \sigma, \nu, d) t(f_2; \mu_2, \sigma, \nu, d)$$
$$t(f_1 + f_2; \mu_1 + \mu_2, \sigma, \nu, d)$$

The likelihood of the model follows the asymptotically Gaussian distribution of the log-scale spectrum[84] and natural scale bispectrum[85].

To ensure that the Xi process exhibits a wide spectrum, we add a likelihood term for the Xi process to encourage fits that adhere to the overall trend. The joint likelihood for the full model is:

$$
\begin{aligned}
L(\theta_2, \theta_3 | \hat{S}, \hat{B}) =& L(\theta_2 | \widetilde{S}) + L(\theta_3 | \widetilde{B}) \\
=& -\frac{1}{N_S} \sum_f \left( \frac{\log(\hat{S}(f)) - \log(\widetilde{S}(f;\theta_2))}{\widehat{Var}[\log(\hat{S}(f))]} \right)^2 \\
& -\frac{1}{N_B} \sum_{f_1} \sum_{f_2} \frac{|\hat{B}(f_1, f_2) - \widetilde{B}(f_1, f_2; \theta_3)|^2}{\widetilde{S}(f_1; \theta_2)\widetilde{S}(f_2; \theta_2)\widetilde{S}(f_1 + f_2; \theta_2)}
\end{aligned}
$$

where the $\hat{S}(f)$ and $\hat{B}(f_1, f_2)$ is the Multitaper power spectrum and bispectrum estimate by formula (3) and (6), $\widetilde{S}(f;\theta_2)$ and $\widetilde{B}(f_1, f_2; \theta_3)$ is the modeled theoretical power spectrum and bispectrum.

This likelihood-based method makes it advantageous to use statistical methods to do the model comparison.

We estimate the model parameters using the Levenberg-Marquardt (LM) optimization algorithm. The LM method is a standard technique for solving nonlinear least squares problems, effectively balancing the Gauss-Newton algorithm and gradient descent. It is particularly suitable when the objective function is the sum of squares of loss functions, as in our joint likelihood estimation, which is also a heteroscedastic curve/surface fitting problem. Employing the LM method, we iteratively update the parameter estimates to minimize the discrepancy between the empirical and theoretical spectrum and bispectrum, ensuring convergence to an optimal solution. The LM algorithm is not a global optimization method. Therefore, we fit the spectra parameters first and then share the parameters with the bispectrum as an initial parameter values (starting points) warmup for the joint fit.

We can then obtain the bicoherence for each component

$$\widetilde{b}_\xi(f_1, f_2) = \frac{\widetilde{B}_\xi(f_1, f_2)}{\sqrt{S(f_1)S(f_2)S(f_1 + f_2)}}$$

$$\widetilde{b}_\rho(f_1, f_2) = \frac{\widetilde{B}_\rho(f_1, f_2)}{\sqrt{S(f_1)S(f_2)S(f_1 + f_2)}}$$

$$\hat{b} = \widetilde{b}_\xi + \widetilde{b}_\rho + \epsilon_b$$

We use full variance to normalize the component bicoherence to avoid instability and ensure the additive way to model the bicoherence. Where $\hat{b}(f_1, f_2)$ denotes the empirical bicoherence of the full signal, and $\epsilon_b(f_1, f_2)$ is the fitting residual that captures any bicoherence structure not accounted for by the two-component decomposition.

Then we recall the test (9) and (10) for testing the non-Gaussianity and linearity of each component.

## Reporting summary

Further information on research design is available in the Nature Portfolio Reporting Summary linked to this article.

## Data availability

The scalp EEG and intracranial EEG (iEEG) data analyzed in this study were obtained from the work of Li et al.[71] (https://www.synapse.org/Synapse:syn26712693/wiki/) and Frauscher et al.[59] (https://ieegatlas.loris.ca/), respectively. Ethical approval was not required for this specific study as it exclusively analyzed pre-existing, publicly available, and anonymized datasets (the HarMNqEEG dataset and the Montreal Neurological Institute (MNI) open iEEG atlas). The collection of the original data received ethical approval from the respective local authorities, as detailed in the original publications[59,71].

## Code availability

The analysis code and derived data used in this study are publicly available on GitHub at https://github.com/rigelfalcon/BiSCA and Zenodo https://doi.org/10.5281/zenodo.19643376[86].

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

## Acknowledgements

We thank Dr. Nicolás von Ellenrieder for generously sharing the MNI iEEG Atlas dataset and for his expert guidance on its use. This work was supported by the National Key R&D Program of China, (2024YFE0215100), the CNS Program of the University of Electronic Science and Technology of China (UESTC) (Grant No. Y03023206100204). M.L. was supported by Hangzhou Dianzi University Seed Fund Project (KYS055623037) and Zhejiang Provincial Higher Education Institutions' Basic Operations Project (GK239909299001-025). L.M. gratefully acknowledges personal support from the Hundred Talents Program of UESTC, the Outstanding Young Talents Program (Overseas) of the National Natural Science Foundation of China, and talent programs of Sichuan Province and Chengdu Municipality. M.B. acknowledges funding from the National Health and Medical Research Council (NHMRC, Grant No. APP2008612).

## Author contributions

Y.W., M.L., and P.A.V.-S. conceived and designed the study. Y.W., M.L., and R.G.R. developed the software and performed the analyses; Y.W., M.L.B.-V., and P.A.V.-S. curated data and provided resources. Y.W., M.L., R.G.R., L.M., M.B., and P.A.V.-S. wrote the manuscript with input from all authors. M.L.B.-V. and P.A.V.-S. supervised the project, and L.M., M.B., M.L.B.-V. and P.A.V.-S. acquired funding.

## Competing interests

The authors declare no competing interests.
