## [Transparent Peer Review file · Communications Biology]

The influence of nonlinear resonance on human cortical oscillations

Corresponding Author: Professor Pedro Valdes-Sosa

This manuscript has been previously submitted at another journal. This document only contains information relating to versions considered at Communications Biology.

Version 0:

Reviewer comments:

Reviewer #1

(Remarks to the Author)

The authors of "The influence of nonlinear resonance on human cortical oscillations" present an interesting piece of research, analyzing EEG recordings of 960 individuals and iEEG signals of 106 epilepsy patients during resting state, with the aim to distinguish linear Gaussian, nonlinear Gaussian, linear non-Gaussian and nonlinear non-Gaussian dynamics. To my opinion, this is an important contribution, but not acceptable in the present form and needs a major revision before publishing in Communications Biology. The main criticism I have is that results are not reproducible because information is missing or the explanation of technical details of the method used by the authors is deficient. In general, the text is difficult to read, especially for readers who do not have a sound mathematical education. If the text is made more reader-friendly, this paper will have a much greater echo. I try to detail at least some of my observations in the sequel.

1) I recommend including a section with a detailed description of the data sets. In the abstract the authors mention 1771 intracranial channels, on line 125 they declare 1771 iEEG recordings. What is true? How many extracranial recordings have been used? Please clarify details about recordings frequency, window lengths used by the authors, channel selection, etc.

2) The authors talk frequently about nonlinear resonance detected via bicoherence. It would be helpful if they explain what they mean by that term. What is resonating and in which manner? The term is quite clear for a linear resonance, meaning that the magnitude of the response solely depends on the frequency but not on the strength of an external stimulus. But the concept is quite unclear for a nonlinear resonance.

3) Their method is sensitive to quadratic nonlinearity, a particular feature, but is not capable to detect any kind of nonlinear autocorrelations, like e.g. mutual information, which in principle can detect (auto)correlations of any order. Thus, they cannot claim that the so-called aperiodic background does not contain nonlinear features or is not driven by nonlinear processes. They can only claim that they are not able to detect any signature of nonlinearity with the method used.

4) On line 64 the authors claim that "Conventional spectral analysis ignores non-sinusoidal structure". What is meant by that? Fourier analysis works very well even for signals with non-sinusoidal wave forms.

5) What is meant by "multiple lags" on line 116? And probably you mean "Nonlinear analysis of time series" instead of "Analysis of nonlinear time series".

6) The fact that sequence of Fourier amplitudes does not contain any information about determinism or nonlinear behavior is also evident via the Wiener-Khinchin theorem, which states that the Fourier transform of the linear autocorrelation function is the power spectral density of the signal. Hence, signatures of determinism are encoded in the sequence of Fourier phases (which are usually ignored) or in a possible correlation between Fourier phases and amplitudes. This fact constitutes also the theoretical background for the method of Fourier transform surrogates. Maybe this comment is helpful for the authors.

- 7) Figure 1: orange and brown colors are hard to distinguish.
- 8) The table with mathematical notations provided in the supplementary material is very helpful, but unfortunately not complete. Many notations are not listed in this table, like e.g. $b^{\wedge}(L)$, $b^{\wedge}(G)$, $c_{\alpha}^{\wedge}(L)$, Please complete table S1.
- 9) Figure 2: PC1, PC2 and PC3 denote principal components?
- 10) Line 192: you state that the value 0.161 is a statistically negligible bicoherence value. What is the threshold for statistical relevant values? Which tet statistics has been used?
- 11) Table 1 on page 10: If the argument is true that nonlinear features are averaged out on larger scales, I find it surprising that the percentage of data segments with nonlinear signatures in EEG recordings (81.6%) is notably higher than in iEEG signals (67.9%). I think this finding is worth discussing in more detail. Furthermore, I find it surprising that iEEG shows a predominance of GL signals in the frontal regions, which is responsible for higher cognitive information processing. Does this finding imply that this activity is controlled by simpler linear mechanism? And what is the physiological interpretation that NGL iEEG signals are more frequent in the occipital area?
- 12) Figure 3 on page 11: color scales are quite disparate for the different panels, but it is not commented on by the authors. Possibly it might be helpful to present results for the frontal, central and occipital regions in form of a histogram, maybe drawn on a logarithmic scale to make small values visible and to present the results using the same scales. Same could be useful for the results shown in Figure 4 panels G) to L).
- 13) Line 251: what do the authors mean by "frequency doubling and intermodulation effects"?
- 14) Isn't it counterintuitive that for the iEEG bifrequency distribution is more diffuse? I would expect a more accentuated distribution. Please comment on that.
- 15) Figure 4: I guess labels of the x-axis in panel B) should be $b^{\wedge}(L)$. Violine plots for the full signal are above those for the Rho processes. Given that for the aperiodic part almost no significant nonlinear signatures are found, isn't that a counterintuitive result? The mean value for the Rho processes is just upon the significance value, while that of the full signal lies clearly above. How does that come about? What means different colors in panels C) to F), Please provide a color scale. Panel I) and L) are both denoted as I).
- 16) Figure 4: Linearity statistics of the Xi processes are somewhat correlated with those of the full signals. What is the quantitative value of this correlation? Is it above what one would expect for random correlations? Please provide also the values of the Pearson coefficients for panels C) to F). What are the corresponding results for the EEG?
- 17) Line 317/318: As already mentioned above. The fact that the authors do not detect signatures of nonlinearity does not imply that signals do not contain information about nonlinearity. The method they are using is sensitive for a particular nonlinear feature. Though, they should slow down when they are writing "The broadband Xi process (aperiodic 1/f component) behaves largely as a linear, Gaussian process, whereas the narrowband Rho components (oscillatory peaks, e.g. the alpha rhythm and Mu rhythm) are the primary sources of nonlinearity in the EEG". None of the two declarations of this sentence is necessarily true. At least, this study does not provide evidence for them. Note, similar declarations are throughout the whole text and should be corrected.
- 18) Similarly in line 333/334: "The question of whether large-scale brain signals exhibit nonlinear instead of purely linear behavior is directly answered by our results." This statement is simply not true, because their method does not search for determinism and nonlinearity in general!
- 19) Line 336/337: "...our bicoherence tests revealed that the most EEG channels exhibited higher-order statistical structure". The manuscript does not contain any statistics about the number of data channels showing nonlinear behavior.
- 20) Line 340/341: "...volume conduction primarily spreads this activity across the scalp..." is just a hypothesis or a proposal for a possible mechanism, but no evidence for this is provided in the manuscript. Authors should be more careful with their declarations.
- 21) Lines 362 to 370: As outlined above mentioning the Wiener Khinchin theorem. The power spectral density, which determines the $f^{\wedge}(-\beta)$ characteristics, has nothing to do with determinism, nonlinearity or chaos. The sequence of Fourier amplitudes carry exclusively information about linear autocorrelations. Thus, the observation that the aperiodic part does not contain quadratic nonlinearity does certainly not challenge the established opinion that the aperiodic processes have a 1/f spectrum. That is just a matter of fact. It is proven and there is a whole bunch of literature sustaining it. Finally – again - the fact that the authors do not find nonlinear signatures with there method, does not imply that they are not present.
- 22) Line 473/474: "The study focused on 60-second eyes, in order to be consistent with the EEG data set." I think it should be "The study focused on 60-second CLOSED eyes, in order to be consistent with the EEG data set." There are more typos throughout the text.
- 23) Page: U_2,U_3,NW,k_i etc. are not explained.

24) Page 45 (supplementary): what is the MNI atlas, what is meant by Fs?

Figure S3: 1) insets are not readable because of the small size of the letters. 2) one should put labels a), b) c) etc. at each panel, 3) the figure holds only for iEEG, how does it look for extracranial EEG?, 4) blue line for chi-square is almost not visible, please use other color.

Reviewer #2

(Remarks to the Author)

Please see the attached PDF for the review.

Reviewer #3

(Remarks to the Author)

In this paper the authors propose tests for nonlinearity and non-Gaussianity based on the analysis of bicoherence calculated from a huge number of empirical data sets. Both the approach and the results are interesting to the scientific community. Most of all, I found interesting that the flat parts of the bicoherence spectra are not just a bias of a truly linear process with Gaussian input but also appear to reflect genuine cross-frequency coupling. I also appreciate the very reasonable approach to estimate the effective number of segments. The paper is well written and clear with a few exceptions. I still have one major and a few minor objections which should be addressed by the authors.

Major

The authors distinguish nonlinearity from non-Gaussianity by the spectral properties of bicoherence. Roughly speaking: a largely flat and significantly non-vanishing bocherence spectrum corresponds to a linear system with a non-Gaussian input, and sharp peaks correspond to nonlinear dynamics with Gaussian input. First of all, I wonder whether sharp peaks could not also arise from non-Gaussian input in combination with nonlinear dynamics. More importantly, I wonder why one cannot explain any univariate data as a linear dynamical system with non-Gaussian input by considering the signal itself as input. Formally spaeking: let $x(t)$ be the data, then we could write $x(t)=y(t)$ with $y(t)=x(t)$. This is a trivial linear AR-model with all AR-matrices set to zero and with non-Gaussian input $y(t)$. Apparently, the authors have a more restrictive understanding of what properties the non-Gaussian input may have. From personal experience I agree that bicoherence occurs as sharp peaks in EEG superimposed on a largely flat background. But in LFP data one can also observe Theta-Gamma phase-amplitude coupling resulting in basically a line in the bicoherence plot, i.e. a sharp peak in the low frequency and broad band in the high frequency. How would the authors interpret this in terms of non-Gaussianity and nonlinearity?

Minor

1. I. 116 "phase coupling" is also within freequency.

2. Fig.1 D) What is denoted as "alpha peak" is already a coupling between (approximately) 10 Hz and 20 Hz, i.e. a coupling with the 2/1 harmonic.

3. Fig.4. The meaning of statistics is unclear at this point. Are the low or high values the statistically significant ones?

4. I.404: "recordings of thrytum" I guess that should read "recordings of the rhythm".

5. I.413; "nonlinearity. a similar ...". The "a" should be capital.

6. I.428: "Still, some spurious mixing in scalp recordings cannot entirely ruled out. To resolve these issues, source localization combined with BiSCA will help to confirm that the identified nonlinear interactions correspond to true neurophysiological coupling rather than mixing artifacts". This confused me. All the analysis done here is univariate and coupling is found between signals at different frequencies and not between signals at different sensor locations. Mixing artifacts are problematic for the interpretation of the latter.

7. I.452: "Nevertheless, there is no question that brain activity is nonstationary." I understand that this is a widespread view and the whole question is not so relevant for this paper. Therefore, I won't ask for changing anything. Just for the record: I totally disagree with this and I would appreciate if the authors just drop the formulation "there is no question".

8. Section 4.2.1 should be polished a little bit. In line 488, the NW is not part of any sentence. Also, when $NW=1.5$, then $K=2NW-1=2$ and not 4.

9. I. 496 "the bispectrum has the favorable properties of zeroing Gaussian processes, being constant for linear systems with non-Gaussian input, ..." This relates to my major objection. Why should it be zero for linear systems with non-Gaussian input?

10. I.569: It would be helpful if the authors write that the estimation of the effective number of segments is explained below.

11, I.701: "warmup" I guess this is the starting value vor the LM algorithm, right?

Version 1:

Reviewer comments:

Reviewer #1

(Remarks to the Author)

The authors took all my objections into account and answered all my questions satisfactorily. I therefore recommend the publication of this interesting and important work.

Reviewer #2

(Remarks to the Author)

I appreciate the Authors' very comprehensive response to my initial review. I note that they have not taken any issue with my "detailed re-derivation", other than clarifying that the shape parameters are shared across all K "rho" processes.

However, I am still concerned that their reasoning is deficient for how the triple sum of my equation (12) reduces to the double sum of their equation (15). I have read in detail their response and have consulted the references (Brillinger & Rosenblatt 1967b, pp 193-195; Nikias & Petropulu 1993, section 2.3 and equation 2.75; Marzocca etl 2008, equation 24; Schetzen 1980, equation 7.3-8) they have provided to justify this reduction from a triple sum to a double sum. As far as I could tell none of these references explicitly address such a reduction, with many seemingly unrelated to this issue. Indeed, a simple calculation of the triple sum for $K=2$ reveals 8 independent components, which as far as I can tell admits of no symmetries to reduce them to the $K(K+1)/2= 3$ terms as implied by the double sum of their equation (15). Further they claim that for $K=6$ the $K^3=216$ reduces to 15 triads (or "9 unique pairs" – which makes no sense in the calculation of the bispectrum), which is at odds with the number of terms implied by their equation (15) of 21.

Now, it is entirely possible that I have made a mistake in my derivations or I have misunderstood some essential point in their response. However, as most readers are not going to go to my efforts to re-derive your method, but may choose to independently implement it, it behoves the authors to present a clear and detailed step-by-step derivation of their novel method in situ in the supplementary material. Given their detailed and lengthy rebuttal it is clear that the Author's are more than capable of doing this.

Until this issue of the explicit derivation of their BiSCA method is clarified and definitively resolved I feel that this otherwise interesting and potentially important paper would be a confusing addition to the literature.

Reviewer #3

(Remarks to the Author)

The was an impressive response. It turned out that my major concern was rather a misunderstanding and could be addressed easily.

Response to Reviewer #1

The influence of nonlinear resonance on human cortical oscillations

Communications Biology

Contents

1 Detailed Response to Reviewer #1	3
1.1 R1-1: Dataset Description	3
1.2 R1-2: “Nonlinear Resonance” Terminology	6
1.3 R1-3: Method Only Detects Quadratic Nonlinearity	6
1.4 R1-4: “Conventional spectral analysis ignores non-sinusoidal structure”	7
1.5 R1-5: “Multiple Lags” and Terminology	7
1.6 R1-6: Wiener-Khinchin Theorem	8
1.7 R1-7: Fig. 1 Colors	8
1.8 R1-8: Supplementary notation tables incomplete	10
1.9 R1-9: Fig. 2 PC1, PC2, PC3	10
1.10 R1-10: Statistical Significance Threshold	10
1.11 R1-11: EEG vs iEEG Nonlinearity Percentage	11
1.12 R1-12: Fig. 3 Color Scales	12
1.13 R1-13: Frequency Doubling and Intermodulation	13
1.14 R1-14: iEEG Bifrequency Distribution More Diffuse	13
1.15 R1-15: Fig. 4 Issues	17
1.16 R1-16: Correlation Quantification	17
1.17 R1-17: Overclaiming (part 1)	18
1.18 R1-18: Overclaiming (part 2)	19
1.19 R1-19: Overclaiming (part 3)	19
1.20 R1-20: Overclaiming (part 4)	20
1.21 R1-21: 1/f Spectrum and Xi Linearity	20
1.22 R1-22: Typo	21
1.23 R1-23: Symbols Unexplained	21
1.24 R1-24: Supplementary Issues	22
2 Supplementary Section S.3.3: Forward-Model Simulation — Volume Conduction and Nonlinearity Prevalence	24
2.1 S.3.3.1 Waveform-Shape Mechanism	24
2.2 S.3.3.2 Spatial-Mixing Mechanism (Volume Conduction)	24
2.3 S.3.3.3 Regional GL/NGL/GNL/NGNL Distribution	24
2.4 S.3.3.4 Simulation Design	25
2.5 S.3.3.5 Source Model	25

2.6	S.3.3.6 Forward Model	27
2.7	S.3.3.7 Signal Parameters	27
2.8	S.3.3.8 Bicoherence Analysis	28
2.9	S.3.3.9 Results	29
3	Supplementary Section S.7: Complementary Approaches to the Linear-Brain Question (R1-18)	30

Manuscript: The influence of nonlinear resonance on human cortical oscillations

Journal: Communications Biology

1 Detailed Response to Reviewer #1

1.1 R1-1: Dataset Description

Reviewer’s Comment: I recommend including a section with a detailed description of the data sets. In the abstract the authors mention 1771 intracranial channels, on line 125 they declare 1771 iEEG recordings. What is true? How many extracranial recordings have been used? Please clarify details about recordings frequency, window lengths used by the authors, channel selection, etc.

Changes Made: Methods Section 4.1 and Supplementary Tables S1-S3: - Replaced ambiguous wording (“iEEG recordings”) with “iEEG bipolar channels” throughout the manuscript where counts are reported, to distinguish channels from per-subject recording sessions. - Added an explicit channel-count reconciliation in the revised supplementary text to explain the sequence 1785 (published atlas) -> 1772 (public release file) -> 1771 (analysis subset after excluding one single-channel subject). - Added the full dataset/parameter package in Supplementary Tables S1-S3 and kept only a short pointer sentence in Methods Section 4.1.

Revised supplementary content added under Tables S1-S3:

Stage	Channels	Subjects	Source/Reason
Frauscher et al. (2018) publication	1,785	106	Original atlas (SEEG: 1,520; ECoG: 265)
Downloaded dataset	1,772	106	Minor version difference in public release
Our analysis	1,771	105	Excluded 1 subject with single channel

Table S1: iEEG Dataset Description (Frauscher et al., 2018)

Parameter	Description
Dataset name	MNI Open iEEG Atlas
Reference	Frauscher, B., et al. (2018). Atlas of the normal intracranial electroencephalogram. Brain , 141(4), 1130-1144

Parameter	Description
Data repository	https://mni-open-ieegatlas.research.mcgill.ca/
Subjects	106 patients with therapy-refractory focal epilepsy
Demographics	54 males; mean age 33.1 ± 10.8 years
Electrode types	SEEG (stereo-EEG): 89 patients (84%), 1,520 channels; ECoG (grids/strips): 17 patients (16%), 265 channels
Total channels	1,785 bipolar channels (left hemisphere: 1,066; right hemisphere: 719)
Channel definition	Bipolar derivation
Recording condition	Resting wakefulness, eyes closed
Epoch duration	60-second artifact-free sections
Sampling rate	Original varied (up to 2000 Hz); downsampled to 200 Hz
Bandpass filter	0.5-80 Hz
Channel inclusion criteria	(1) Located in normal tissue per MRI; (2) Outside seizure onset zone; (3) No interictal epileptic discharges; (4) No overt slow-wave anomaly; (5) Recording obtained ≥ 72 h after electrode insertion
Channel exclusions	White matter contacts; lesional tissue; large cortical malformations
Our subset	1,771 bipolar channels from 105 subjects (excluded 1 subject with single channel)

Table S2: Scalp EEG Dataset Description (Li et al., 2022)

Parameter	Description
Dataset name	HarMNqEEG (Harmonized Multinational qEEG Norms)
Reference	Li, M., et al. (2022). Harmonized-Multinational qEEG Norms. NeuroImage , 256, 119190
Data repository	https://www.synapse.org/HarMNqEEG (Synapse ID: syn26712693)
Subjects	1,564 healthy participants (783 females, 781 males)
Age range	5-95 years (lifespan dataset)
Countries	9 countries: Barbados, China, Colombia, Cuba, Germany, Malaysia, Russia, Switzerland, USA
Recording devices	12 different EEG devices across sites

Parameter	Description
Channels	19 channels (10-20 International System: Fp1, Fp2, F3, F4, C3, C4, P3, P4, O1, O2, F7, F8, T3/T7, T4/T8, T5/P7, T6/P8, Fz, Cz, Pz)
Reference	Average reference; Pz electrode removed to eliminate redundancy (19 → 18 channels)
Recording condition	Resting-state, eyes closed, quasi-stationary
Epoch duration	>= 1 minute artifact-free; segmented into 2.56 s epochs (frequency resolution: 0.39 Hz)
Frequency range	Amplifiers: 0.5-35 Hz; analysis restricted to 1.17-19.14 Hz
Preprocessing	(1) Within-site artifact rejection; (2) Average re-referencing; (3) Maximum likelihood shrinkage for positive-definite cross-spectra; (4) Global scale factor (GSF) correction
Quality control	Three-stage: local site filtering, neurophysiologist visual inspection, machine-learning outlier detection (t-SNE + robust Mahalanobis distance)
Outlier exclusion	191 subjects identified as outliers and excluded from normative analysis
Our subset	960 subjects retained (excluded subjects with <\$140 usable segments to match iEEG segment count of 157)

Table S3: Our Analysis Parameters (Both Datasets)

Parameter	Value
Sampling rate	200 Hz
Window length	1.5 s (300 samples)
FFT length	300
Overlap	75%
Frequency range	0-50 Hz
Spectral method	Multitaper (pmtm) with scaled sine tapers
Time-bandwidth product	NW = 1.5 (K = 2NW-1 = 2 tapers)
Bicoherence normalization	Haubrich (1965)
Segments per channel	157 (matched across datasets)
iEEG montage	Bipolar (original atlas format)
Scalp EEG montage	Average reference (18 channels after Pz removal)

Response: We clarified this point by using “bipolar channels” consistently and by adding an explicit

reconciliation of the iEEG counts (1785 -> 1772 -> 1771). In short, “1771 intracranial channels” is the final analysis subset of bipolar channel time series, not a count of recording sessions.

We also added the requested dataset and processing details for both iEEG and scalp EEG in Supplementary Tables S1-S3 (dataset composition, preprocessing, sampling settings, and analysis parameters), with a concise pointer retained in Methods Section 4.1.

1.2 R1-2: “Nonlinear Resonance” Terminology

Reviewer’s Comment: The authors talk frequently about nonlinear resonance detected via bicoherence. It would be helpful if they explain what they mean by that term. What is resonating and in which manner? The term is quite clear for a linear resonance, meaning that the magnitude of the response solely depends on the frequency but not on the strength of an external stimulus. But the concept is quite unclear for a nonlinear resonance.

Changes Made: Introduction (after the first mention of “nonlinear resonance”): > “*We use the term ‘nonlinear resonance’ to describe the phenomenon where quadratic frequency mixing generates combination frequencies that match natural frequencies of the system, producing large-amplitude oscillations in those modes (Kartashova, 2010; Rajasekar & Sanjuan, 2016). This phenomenon is quantified by the bispectrum $B(f_1, f_2) = \mathbb{E}[X(f_1)X(f_2)X^*(f_1 + f_2)]$, which captures frequency mixing effects such as harmonic generation and intermodulation (Brillinger, 2001; Nichols et al., 2009).*”

Response: We added an explicit definition in the Introduction. In our usage, nonlinear resonance refers to quadratic frequency mixing that generates combination frequencies $(f_1 + f_2, |f_1 - f_2|)$, quantified by the bispectrum.

1.3 R1-3: Method Only Detects Quadratic Nonlinearity

Reviewer’s Comment: Their method is sensitive to quadratic nonlinearity, a particular feature, but is not capable to detect any kind of nonlinear autocorrelations, like e.g. mutual information, which in principle can detect (auto)correlations of any order. Thus, they cannot claim that the so-called aperiodic background does not contain nonlinear features or is not driven by nonlinear processes. They can only claim that they are not able to detect any signature of nonlinearity with the method used.

Changes Made: - Manuscript-wide: Updated wording “no detectable nonlinearity” to “no detectable quadratic nonlinearity” where appropriate. - Manuscript-wide: Applied the same qualifier consistently in all similar statements. - Discussion section: Added a detailed note acknowledging that formal testing of the independence assumption between Xi and Rho components is planned for future work: > “*Formal testing of the statistical independence assumption between Xi and*

Rho components—for example, via mutual information applied to laminar recordings, where distinct oscillatory generators occupy different cortical layers (Silva et al., 1991; van Kerkoerle et al., 2014; Mendoza-Halliday et al., 2024) and the aperiodic trend arises partly from subcritical network dynamics (Brake et al., 2024)—is planned for future work.”

Response: We agree. We now state explicitly that our method tests quadratic (bispectral) nonlinearity only, so a negative finding does not exclude other nonlinear dependencies. Complementary independence validation is retained as future work.

1.4 R1-4: “Conventional spectral analysis ignores non-sinusoidal structure”

Reviewer’s Comment: 4) On line 64 the authors claim that “Conventional spectral analysis ignores non-sinusoidal structure”. What is meant by that? Fourier analysis works very well even for signals with non-sinusoidal wave forms.

Changes Made: Introduction (Section 1): > “*While the Fourier transform faithfully captures non-sinusoidal waveforms through harmonic decomposition, the conventional power spectrum discards the inter-harmonic phase relationships (Brillinger, 1965, 2001). Two signals with identical power spectra can exhibit fundamentally different phase structures; the bispectrum preserves this phase coupling information that the power spectrum discards. We illustrate this in Fig. S2, where simulated time series with identical power spectra (same AR model) but different innovations or system nonlinearities produce distinct bicoherence signatures.*”

Response: We corrected the wording: Fourier decomposition captures non-sinusoidal waveforms, while the power spectrum discards inter-harmonic phase relationships. The revised sentence and Fig. S2 now make this distinction explicit.

1.5 R1-5: “Multiple Lags” and Terminology

Reviewer’s Comment: 5) What is meant by “multiple lags” on line 116? And probably you mean “Nonlinear analysis of time series” instead of “Analysis of nonlinear time series”.

Changes Made: Introduction (Section 1): > “*Nonlinear analysis of time series is therefore needed. The power spectrum is the Fourier transform of the autocorrelation function $R(\tau) = E[x(t)x(t+\tau)]$, which involves a single time lag τ . In contrast, the bispectrum is the Fourier transform of the third-order cumulant $c_3(\tau_1, \tau_2) = E[x(t)x(t+\tau_1)x(t+\tau_2)] - (\text{Gaussian terms})$, > > which involves two time lags (τ_1, τ_2) and captures phase relationships invisible to second-order analysis.*”

Response: We adopted the reviewer’s suggested phrasing (“Nonlinear analysis of time series” rather than “Analysis of nonlinear time series”). We also made explicit that bispectral quantities

correspond to third-order dependence across two time lags (τ_1 , τ_2), in contrast to the power spectrum which is based on a single-lag autocorrelation.

1.6 R1-6: Wiener-Khinchin Theorem

Reviewer’s Comment: 6) The fact that sequence of Fourier amplitudes does not contain any information about determinism or nonlinear behavior is also evident via the Wiener-Khinchin theorem, which states that the Fourier transform of the linear autocorrelation function is the power spectral density of the signal. Hence, signatures of determinism are encoded in the sequence of Fourier phases (which are usually ignored) or in a possible correlation between Fourier phases and amplitudes. This fact constitutes also the theoretical background for the method of Fourier transform surrogates. Maybe this comment is helpful for the authors.

Changes Made: Introduction (Section 1), following the revised text from R1-4 and R1-5: > *“This limitation is formalized by the Wiener-Khinchin theorem: the power spectrum, being the Fourier transform of the autocorrelation function, is a purely second-order statistic that captures only linear correlations (Brillinger, 1965); in contrast, the bispectrum explicitly quantifies inter-frequency phase coupling.”*

Response: We added this Wiener-Khinchin clarification to state explicitly that the power spectrum is second-order and does not encode inter-frequency phase coupling, which motivates the bispectral analysis. We did not add Fourier-transform surrogates in this revision because significance is already tested with the bispectral null framework (Methods Section 4.2.3).

1.7 R1-7: Fig. 1 Colors

Reviewer’s Comment: 7) Fig. 1: orange and brown colors are hard to distinguish.

Changes Made: Fig. 1: - Adopted a colorblind-safe palette for all Fig. 1 lines: - Non-peak lines (spectrum plot only): Orange (ξ process: Bluish Green, fitted spectrum: IBM Blue) - Peak lines (appear on both spectrum and bicoherence plots): Pure Red, Bright Violet, Bright Yellow - Peak line colors specifically chosen to contrast with the inferno/viridis colormaps used in bicoherence surface plots. - Warm and cool colors are balanced across the six lines; all are distinguishable for viewers with red-green color vision deficiency.

Response: We agree. We updated the palette to a colorblind-safe scheme that improves discriminability while preserving consistent semantic mapping across panels.

Updated Fig. 1

1.8 R1-8: Supplementary notation tables incomplete

Reviewer’s Comment: 8) The table with mathematical notations provided in the supplementary material is very helpful, but unfortunately not complete. Many notations are not listed in this table, like e.g. $b^{(L)}$, $b^{(G)}$, $c_{\alpha}^{(L)}$, Please complete table S1.

Changes Made: Supplementary Section S.1, Tables S4-S5: - Added missing definitions for: F_s , U_2 , U_3 , k_i , $b^{(G)}$, $b^{(L)}$, $c_{\alpha}^{(G)}$, $c_{\alpha}^{(L)}$, N_s^{eff} to Table S5 “Symbol of quantities and operators”. - Definitions for the statistical test symbols ($b^{(G)}$, $b^{(L)}$, $c_{\alpha}^{(G)}$, $c_{\alpha}^{(L)}$, N_s^{eff}) are provided in R1-10.

Response: We expanded the S.1 symbol tables, especially Table S5, to define all symbols and statistics used in the manuscript and figures, including test statistics and critical values.

1.9 R1-9: Fig. 2 PC1, PC2, PC3

Reviewer’s Comment: 9) Fig. 2: PC1, PC2 and PC3 denote principal components?

Changes Made: - Added to the Fig. 2 caption a clarifying sentence: - “PC1, PC2, and PC3 denote the first three principal components from PCA applied to Takens’ delay-embedded state vectors (Hankel-matrix time-delay embedding with lag order p).”

Response: Yes. The 3D phase portraits show the attractor geometry projected onto the first three principal components (PC1/PC2/PC3) of the Takens-embedded state space, constructed via Hankel-matrix time-delay embedding with lag order p .

1.10 R1-10: Statistical Significance Threshold

Reviewer’s Comment: Line 192: you state that the value 0.161 is a statistically negligible bicoherence value. What is the threshold for statistical relevant values? Which tet statistics has been used?

Changes Made: - Methods Section 4.2.3: The statistical framework (Eqs. 10-18), including the asymptotic $\chi^2(2)$ null distribution, the median-based Gaussianity test, and the maximum-based linearity test with Gumbel extreme-value calibration, was already described in the original submission. - Results Section 2.1 (near Fig. 2F caption): Added a clarifying sentence stating that $b^{(L)} = 0.0161$ is a measured bicoherence value (not a significance threshold) and referencing the threshold derivation in Methods Section 4.2.3. - Added explicit quantitative clarification:

Quantity	Value	Interpretation
$b^{(L)}$	0.0161	Measured max bicoherence
$c_{\alpha}^{(L)}$	≈ 0.31	Significance threshold ($\alpha = 0.001$)

Quantity	Value	Interpretation
----------	-------	----------------

Response: The test statistic is the maximum bicoherence $b^{(L)} = \max_{f_1, f_2} |\hat{b}(f_1, f_2)|$, evaluated with the $\chi^2(2)$ -based framework and Gumbel calibration described in Methods Section 4.2.3 (Eqs. 10–18). In Fig. 2F, $b^{(L)} = 0.0161$ is well below the significance threshold $c_\alpha^{(L)} \approx 0.31$, so the result is statistically negligible.

1.11 R1-11: EEG vs iEEG Nonlinearity Percentage

Reviewer’s Comment: Table 1 on page 10: If the argument is true that nonlinear features are averaged out on larger scales, I find it surprising that the percentage of data segments with nonlinear signatures in EEG recordings (81.6%) is notably higher than in iEEG signals (67.9%). I think this finding is worth discussing in more detail. Furthermore, I find it surprising that iEEG shows a predominance of GL signals in the frontal regions, which is responsible for higher cognitive information processing. Does this finding imply that this activity is controlled by simpler linear mechanism? And what is the physiological interpretation that NGL iEEG signals are more frequent in the occipital area?

Changes Made:

1. New Supplementary Section S.3.3 (forward-model simulation): Injected a focal nonlinear mu source plus single-tone alpha sources through realistic iEEG super-leadfields and EEG head-models on the same 2k cortical mesh. Key result (real-leadfield configuration, significance level $\alpha=0.001$): fraction significant in iEEG = 30.9%, in EEG = 94.7%. The complete framework and figures are now provided in Supplementary Section S.3.3.
2. Discussion Section 3.1 (“Linear vs. Nonlinear Modeling Debate”) was revised to incorporate the new forward-model evidence from Section S.3.3, supporting the interpretation that the higher scalp-EEG nonlinearity prevalence can reflect volume-conduction spread and motivating our focus on iEEG for regional analyses.
3. Discussion Section 3.2 (“Interpretation of Oscillatory vs. Aperiodic Components”) was revised to briefly interpret the frontal GL and occipital NGL patterns, emphasizing weaker frontal alpha-band power and the possibility that occipital NGL reflects marginal non-Gaussianity together with a relatively symmetric waveform.
4. Abstract revised to include the quantitative 81.6%/67.9% finding and volume-conduction explanation, replacing the original vague statement (“Both recordings show significant departures from linear Gaussian behavior”). Revised abstract:

“Whether macroscale brain signals reflect linear or nonlinear organization remains poorly characterized. This distinction matters for modeling neural dynamics and interpreting

oscillatory biomarkers of cognition and disease. Spectral analysis reveals aperiodic broadband and rhythmic narrowband components but does not capture nonlinear resonance, such as quadratic phase coupling among oscillations, which requires higher-order spectral analysis. We introduce BiSpectral EEG Component Analysis (BiSCA), combining spectral and bispectral analysis to separate aperiodic (Ξ) from rhythmic (Rho) components, localize nonlinear signatures, and distinguish nonlinearity from non-Gaussianity; simulations confirm this separation. Applying BiSCA to two large datasets (1,771 intracranial channels; 960 scalp EEG subjects), we detect significant nonlinear or non-Gaussian structure in 81.6% of scalp EEG and 67.9% of iEEG channels; forward modeling indicates the higher scalp prevalence reflects volume-conduction spread of focal nonlinear sources. In spatially focal iEEG, aperiodic Ξ shows no detectable quadratic nonlinearity or non-Gaussianity, whereas Rho components, including Alpha and Mu, carry the dominant cortical quadratic coupling. Despite higher occipital Alpha power, the strongest nonlinear signatures arise from parietal Mu. Nonlinear resonance is thus expressed primarily through oscillatory rather than aperiodic dynamics.”

5. Added concise quantitative support in the revised manuscript text for category-level interpretation (GL/NGL/GNL/NGNL thresholds and proportions), and moved extended tabular detail to Supplementary Section S.3.3.

Response: The higher scalp EEG percentage is now supported by direct forward-model evidence (Section S.3.3: 94.7% EEG vs 30.9% iEEG for identical sources), consistent with volume-conduction spread rather than stronger source-level nonlinearity. We also added a concise frontal GL / occipital NGL interpretation in the Discussion, with extended detail in Supplementary Section S.3.3.

1.12 R1-12: Fig. 3 Color Scales

Reviewer’s Comment: 12) Fig. 3 on page 11: color scales are quite disparate for the different panels, but it is not commented on by the authors. Possibly it might be helpful to present results for the frontal, central and occipital regions in form of a histogram, maybe drawn on a logarithmic scale to make small values visible and to present the results using the same scales. Same could be useful for the results shown in Fig. 4 panels G) to L).

Changes Made: Fig. 3 (GL/NGL/GNL/NGNL panels): - Unified color scales across panels within each dataset (rsEEG and iEEG independently, as the two datasets have different dynamic ranges). - Added \log_{10} -scaled bifrequency panels alongside linear-scale panels. Colorbar labels display original probability values (e.g., 0.001, 0.01, 0.1); the \log_{10} mapping makes small probability differences visible. - Zero-probability bins mapped to the color-scale floor (black in the *inferno* colormap); invalid frequency-mask regions remain blank (white).

Fig. 4 (panels G–L): - Tested unifying color scales across panel pairs sharing the same physical quantity (G/H: energy; I/J: $b^{(G)}$; K/L: $b^{(L)}$) with \log_{10} mapping; retained independent scales per

panel in the submitted version (see both versions below).

Response: We unified color scales within each dataset and added \log_{10} panels to improve small-value visibility and cross-panel comparability in Fig. 3. For Fig. 4 G–L, we retained independent panel scales in the submitted version because unified scaling compresses the ξ panels due to the large ξ – ρ magnitude gap (Supplementary Section S.3.3).

1.13 R1-13: Frequency Doubling and Intermodulation

Reviewer’s Comment: 13) Line 251: what do the authors mean by “frequency doubling and intermodulation effects”?

Changes Made: Results Section 2.3: Added bispectral notation and back-reference to the Introduction: > “...suggest frequency doubling ($f + f = 2f$; diagonal ridge) and intermodulation ($f_1 + f_2$, $f_1 \neq f_2$; off-diagonal bands) effects, following the nonlinear resonance framework introduced in the Introduction.”

Response: Defined. In Results Section 2.3, we now specify frequency doubling ($f + f = 2f$) and intermodulation ($f_1 + f_2$, $f_1 \neq f_2$), with a back-reference to the Introduction.

1.14 R1-14: iEEG Bifrequency Distribution More Diffuse

Reviewer’s Comment: 14) Isn’t it counterintuitive that for the iEEG bifrequency distribution is more diffuse? I would expect a more accentuated distribution. Please comment on that.

Changes Made: Discussion (near description of bifrequency distributions, referencing Figures 3B/D): - Added the following text: *“The more diffuse bifrequency distribution in iEEG reflects greater spatial specificity: iEEG electrodes distributed across different cortical regions each detect local nonlinear sources at distinct frequencies, producing significance at diverse bifrequency locations. Conversely, volume conduction causes each EEG channel to mix multiple sources, concentrating significance at fewer, more stereotyped bifrequency locations.”*

Response: We clarified this point in the revised Discussion: the diffuse iEEG map arises because Figures 3B/D pool significance locations across many electrodes, each sampling region-specific nonlinear frequencies, whereas scalp EEG source mixing concentrates significance at more stereotyped bifrequency bins.

Updated Fig. 3

Fig. 4 with unified color scales per quantity pair (\log_{10} mapping). The ξ panels (G, I, K) lose spatial contrast because their values occupy only the low end of the shared scale.

Fig. 4 with independent color scales (submitted version). Each panel uses its own range, maximizing spatial readability.

1.15 R1-15: Fig. 4 Issues

Reviewer’s Comment: 15) Fig. 4: I guess labels of the x-axis in panel B) should be $b^{(L)}$. Violine plots for the full signal are above those for the Rho processes. Given that for the aperiodic part almost no significant nonlinear signatures are found, isn’t that a counterintuitive result? The mean value for the Rho processes is just upon the significance value, while that of the full signal lies clearly above. How does that come about? What means different colors in panels C) to F), Please provide a color scale. Panel I) and L) are both denoted as I).

Changes Made: Fig. 4: - Fixed Panel B x-axis label to $b^{(L)}$. - Added colorbar to Panels C–F; caption updated to describe point density encoding. - Changed panel label font from Arial to Aptos so that lowercase “l” and uppercase “I” are visually distinct. - Discussion: Added explanation of full-signal vs. Rho bicoherence difference. - Added explicit model statement in Section 4.2.4 (Eq. 15) that component bicoherences are normalized by full-signal power and combined as

$$\tilde{b}_\xi(f_1, f_2) = \frac{\tilde{B}_\xi(f_1, f_2)}{\sqrt{S(f_1)S(f_2)S(f_1+f_2)}}, \quad \tilde{b}_\rho(f_1, f_2) = \frac{\tilde{B}_\rho(f_1, f_2)}{\sqrt{S(f_1)S(f_2)S(f_1+f_2)}}, \quad \hat{b} = \tilde{b}_\xi + \tilde{b}_\rho + \epsilon_b.$$

Response: 1. The reviewer is correct; the Panel B x-axis label was fixed to $b^{(L)}$. 2. The full-signal violin can lie above the Rho violin because \hat{b} includes the fitting residual ϵ_b , while \tilde{b}_ρ does not. 3. Colors in Panels C–F encode point density from 2D histograms; a colorbar and caption explanation were added. 4. Panel I/L was a font-readability issue (Arial), not a labeling error; this was resolved by switching to Aptos.

1.16 R1-16: Correlation Quantification

Reviewer’s Comment: 16) Fig. 4: Linearity statistics of the Xi processes are somewhat correlated with those of the full signals. What is the quantitative value of this correlation? Is it above what one would expect for random correlations? Please provide also the values of the Pearson coefficients for panels C) to F). What are the corresponding results for the EEG?

Changes Made:

1. Results Section 2.4: Revised the statement “No such relationship exists for the Xi process” to “A substantially weaker relationship exists for the Xi process ($r = 0.40$) compared to the Rho process ($r = 0.90$),” because the quantitative analysis shows a moderate (not absent) Xi–Full linearity correlation.
2. Fig. 4 caption: Added Pearson r values for all four scatter/density panels (C–F) as requested by the reviewer.
3. Added explicit quantitative summary block (aligned with the revised manuscript values):

Test statistic	Xi vs Full	Rho vs Full
Gaussianity ($b^{(G)}$)	$r = -0.08$	$r = 0.75$
Linearity ($b^{(L)}$)	$r = 0.40$	$r = 0.90$

4. Moved interpretation detail out of Response and into this section: effect-size pattern is consistently stronger for Rho than Xi.

Response: The requested Pearson coefficients are now reported for Panels C–F (Xi vs Full: $r = -0.08$ for $b^{(G)}$, $r = 0.40$ for $b^{(L)}$; Rho vs Full: $r = 0.75$ for $b^{(G)}$, $r = 0.90$ for $b^{(L)}$).

These values are above random-correlation expectation in magnitude and show a consistent effect-size pattern: correlations with Full are much stronger for Rho than for Xi. The decomposition analysis was performed on iEEG; corresponding scalp EEG context is discussed in R1-11.

1.17 R1-17: Overclaiming (part 1)

Reviewer’s Comment: 17) Line 317/318: As already mentioned above. The fact that the authors do not detect signatures of nonlinearity does not imply that signals do not contain information about nonlinearity. The method they are using is sensitive for a particular non-linear feature. Though, they should slow down when they are writing “The broadband Xi process (aperiodic 1/f component) behaves largely as a linear, Gaussian process, whereas the narrowband Rho components (oscillatory peaks, e.g. the alpha rhythm and Mu rhythm) are the primary sources of nonlinearity in the EEG”. None of the two declarations of this sentence is necessarily true. At least, this study does not provide evidence for them. Note, similar declarations are throughout the whole text and should be corrected.

Changes Made: - Revised the Discussion claim (Section 3.1) to scope the statement explicitly to quadratic phase coupling, removing the prior general overclaim about Xi. - Added the qualifier “at the bispectral (quadratic) level” consistently in Section 2.4 title/body, Discussion, and Abstract (see also R1-11). - Added an explicit forward-looking sentence that broader Xi linear/Gaussian status remains a hypothesis requiring higher-order tests (e.g., trispectrum).

Response: Our bispectral test is sensitive only to quadratic (second-order) nonlinearity (as acknowledged in R1-3), so the original phrasing overstated what the evidence supports. We added the “quadratic” qualifier and reframed the Xi characterization as a hypothesis to be confirmed rather than a definitive finding.

1.18 R1-18: Overclaiming (part 2)

Reviewer’s Comment: 18) Similarly in line 333/334: ”The question of whether large-scale brain signals exhibit nonlinear instead of purely linear behavior is directly answered by our results.” This statement is simply not true, because their method does not search for determinism and nonlinearity in general!

Changes Made: - Original (Discussion Section 3.1): “The question of whether large-scale brain signals exhibit nonlinear instead of purely linear behavior is directly answered by our results.” - Revised: “The question of whether large-scale brain signals exhibit nonlinear instead of purely linear behavior is directly addressed by our bicoherence results, which detect the presence of quadratic phase coupling at the macroscopic signal level. The bicoherence test detects whether quadratic nonlinearity is present (i.e., not fully averaged out at macroscopic scale); it does not classify the mechanistic type of the detected nonlinearity (deterministic vs. stochastic). Nonlinear structure is robustly detected in the oscillatory Rho-related component (81.6% EEG channels, 67.9% iEEG channels; Table 1; Fig. 3; Section S.3.3), whereas the broadband Xi process showed no detected quadratic nonlinearity or non-Gaussianity in this dataset.” - Added explicit scope qualifier in the manuscript and reply: the test targets quadratic nonlinearity in stochastic processes, not deterministic chaos. - Added concise method-positioning pointer to Supplementary Section S.7 (comparison with complementary time-domain and information-theoretic approaches).

Response: Our method does not identify deterministic versus stochastic mechanism, and we revised the manuscript to state this limit explicitly. Within that scope, we report detected macroscopic quadratic phase coupling in the oscillatory component (Table 1; Fig. 3; Section S.3.3), while Xi is now described conservatively for this dataset.

1.19 R1-19: Overclaiming (part 3)

Reviewer’s Comment: 19) Line 336/337: ”...our bicoherence tests revealed that the most EEG channels exhibited higher-order statistical structure”. The manuscript does not contain any statistics about the number of data channels showing nonlinear behavior.

Changes Made: - Original (Discussion Section 3.1): “our bicoherence tests revealed that the most EEG channels exhibited higher-order statistical structure” - Revised (Discussion, same sentence): “our bicoherence tests revealed that, as shown in Table 1 and Fig. 3, 81.6% of EEG channels and 67.9% of iEEG channels exhibited significant higher-order statistical structure” - Added explicit evidence pointer: “as shown in Table 1 and Fig. 3” (previously missing in Discussion). - Added channel-level percentages (81.6% EEG, 67.9% iEEG) as requested by the reviewer. - Regional breakdowns of the GL/NGL/GNL/NGNL classification proportions are provided in Supplementary Section S.3.3.3 (grouped bar charts by brain region for both iEEG and EEG).

Response: We agree and revised the Discussion sentence to report channel-level statistics explicitly

with evidence pointers (81.6% EEG, 67.9% iEEG; Table 1 and Fig. 3). Per-region classification proportions are further detailed in Supplementary Section S.3.3.3.

1.20 R1-20: Overclaiming (part 4)

Reviewer’s Comment: 20) Line 340/341: "...volume conduction primarily spreads this activity across the scalp..." is just a hypothesis or a proposal for a possible mechanism, but no evidence for this is provided in the manuscript. Authors should be more careful with their declarations.

Changes Made: - Discussion Section 3.1 was revised to replace the earlier mechanistic overstatement with a more cautious formulation that cites the new forward-model evidence from Section S.3.3 (94.7% significant EEG vs. 30.9% iEEG from identical sources), so that volume conduction is presented as an evidence-supported interpretation rather than proof.

Response: The reviewer is correct; the original wording over-asserted mechanism. We revised the text to cite the forward-model evidence (Section S.3.3; 94.7% EEG vs 30.9% iEEG) and now present volume conduction as an evidence-supported interpretation rather than proof.

1.21 R1-21: 1/f Spectrum and Xi Linearity

Reviewer’s Comment: 21) Lines 362 to 370: As outlined above mentioning the Wiener Khinchin theorem. The power spectral density, which determines the $f^{-\beta}$ characteristics, has nothing to do with determinism, nonlinearity or chaos. The sequence of Fourier amplitudes carry exclusively information about linear autocorrelations. Thus, the observation that the aperiodic part does not contain quadratic nonlinearity does certainly not challenge the established opinion that the aperiodic processes have a 1/f spectrum. That is just a matter of fact. It is proven and there is a whole bunch of literature sustaining it. Finally - again - the fact that the authors do not find nonlinear signatures with their method, does not imply that they are not present.

Changes Made:

Discussion Section 3.2 (“Interpretation of Oscillatory vs. Aperiodic Components”):

- Qualified all Xi linearity claims with “with respect to quadratic phase coupling” (see also R1-17).
- Removed the assertion “We have not been able to find convincing reports in the literature that the aperiodic component is a 1/f process.” Replaced with a neutral framing: while “1/f” is widely used, empirical exponents range from 0 to 4 (Donoghue et al., 2022) and vary

with frequency band (Boncompte et al., 2025); the precise functional form remains an open model-selection question.

- Reframed the concluding sentence: “Whether the Xi process is predominantly linear and Gaussian beyond quadratic phase coupling remains a hypothesis to be confirmed with higher-order statistical tests (e.g., the trispectrum).”
- Added concise literature synthesis in the revised manuscript (Donoghue et al., 2022; Boncompte et al., 2025; Kramer & Chu, 2024; Valdes et al., 1992).
- Added that even parametric frameworks require a knee parameter for low-frequency deviations from a pure power law (Donoghue et al., 2020). Additionally, nonparametric spectral decomposition reveals that the aperiodic exponent is frequency-dependent and the spectral shape can be irregular across conditions and brain regions (Hu et al., 2024), reinforcing the “open model-selection question” framing.
- Cited Hu et al. (2024) at the nonparametric modeling discussion: their Xi-Pi decomposition empirically shows frequency-dependent aperiodic exponents and irregular spectral shapes, in line with our treatment of Xi without assuming a fixed parametric law.

Response: PSD is a second-order statistic and does not by itself constrain nonlinear dynamics; our bispectral test addresses a complementary quantity. We revised the text accordingly to frame the functional form of the aperiodic spectrum as an open model-selection question and to qualify Xi linearity claims to the quadratic scope of our test (R1-17).

1.22 R1-22: Typo

Reviewer’s Comment: 22) Line 473/474: ”The study focused on 60-second eyes, in order to be consistent with the EEG data set.” I think it should be ”The study focused on 60-second CLOSED eyes, in order to be consistent with the EEG data set.” There are more typos throughout the text.

Changes Made: - Original (Methods Section 4.1): “The study focused on 60-second eyes, in order to be consistent with the EEG data set.” - Revised: “The study focused on 60-second eyes-closed recordings, in order to be consistent with the EEG data set.” - Performed a full proofreading pass on the manuscript to correct additional typographical errors.

Response: We corrected the typo.

1.23 R1-23: Symbols Unexplained

Reviewer’s Comment: 23) Page: U_2 , U_3 , NW , k_i etc. are not explained.

Changes Made: - Consolidated with R1-8: added missing symbols to Table S4. - Currently

defined in Table S4 of the Supplementary Information: $x, x, X, i, c, e, f, t, k, NW, N, W$ - Missing symbols added: - F_s = Sampling frequency (Hz) - U_2, U_3 = Second and third-order cumulant statistics - k_i = Taper index (already partially covered by k , clarified notation) - $b^{(L)}$ = Maximum bicoherence magnitude (linearity test statistic) - $b^{(G)}$ = Median bicoherence magnitude (Gaussianity test statistic) - $c_\alpha^{(L)}$ = Critical value for linearity test at significance level α - $c_\alpha^{(G)}$ = Critical value for Gaussianity test at significance level α - N_s^{eff} = Effective number of independent segments

Response: We ensured all symbols are defined in the Supplementary table, including statistical thresholds and normalization constants.

1.24 R1-24: Supplementary Issues

Reviewer’s Comment: 24) Page 45 (supplementary): what is the MNI atlas, what is meant by F_s ? Fig. S2: 1) insets are not readable because of the small size of the letters. 2) one should put labels a), b) c) etc. at each panel, 3) the figure holds only for iEEG, how does it look for extracranial EEG?, 4) blue line for chi-square is almost not visible, please use other color.

Changes Made: - Supplementary Section S.1 tables: clarified the dataset label as “MNI Open iEEG Atlas” in the dataset table and added “ F_s = Sampling frequency (Hz)” in the symbol table. - Fig. S3: (1) increased inset font sizes; (2) added panel labels (a)–(e); (3) changed χ^2 line color from blue to magenta; (4) added shared legend; updated simulation parameters so all five cases match manuscript expectations. Caption updated accordingly. - Added explicit cross-reference for reviewer point (3) (extracranial EEG counterpart): Fig. S3 is kept as iEEG validation with reduced spatial-mixing confounds, while the scalp-EEG counterpart is provided in main Fig. 3 and interpreted with Supplementary Section S.3.3. - Added concise rationale sentence in the revised supplementary explanation for point (3): sensor-level scalp EEG bicoherence reflects spatial source mixing, so Fig. S3 is used as low-mixing iEEG validation and the scalp counterpart is reported in Fig. 3 with supporting evidence in S.3.3. - Added a summary table mapping all four Fig. S3 requests to implemented revisions:

Reviewer request (Fig. S3)	Implemented change
(1) Insets unreadable	Increased inset font sizes
(2) Missing panel labels	Added labels (a)–(e)
(3) Need extracranial EEG counterpart	Added explicit pointer to main Fig. 3 + S.3.3
(4) Blue χ^2 line low visibility	Changed to magenta

Response: We addressed all four Fig. S3 requests and clarified the terminology in Supplementary Section S.1 tables, including the dataset label and the definition of F_s . The extracranial EEG

Updated Fig. S3

counterpart is now explicitly linked to main Fig. 3, with supporting evidence in Supplementary Section S.3.3.

2 Supplementary Section S.3.3: Forward-Model Simulation — Volume Conduction and Nonlinearity Prevalence

The higher prevalence of significant nonlinearity in scalp EEG (81.6%) compared to iEEG (67.9%) motivates mechanistic interpretation. Two candidate mechanisms — waveform-shape asymmetry and spatial mixing via volume conduction — are reviewed below, followed by a forward-model simulation that quantifies their combined effect and a regional classification analysis.

2.1 S.3.3.1 Waveform-Shape Mechanism

Non-sinusoidal alpha and mu waveforms generate bicoherence signatures consistent with physiological waveform asymmetry rather than necessarily stronger nonlinear dynamics. The bispectrum is sensitive to waveshape properties such as rise-decay and peak-trough asymmetry (Bartz et al., 2019), and non-sinusoidal waveforms can produce harmonic structure linked to underlying physiology (Cole & Voytek, 2017). Mu rhythms have been reported to exhibit stronger peak-trough asymmetry than alpha rhythms (Bender et al., 2023), and non-sinusoidal alpha can generate narrow-band harmonic peaks in bicoherence (Schaworonkow, 2023).

2.2 S.3.3.2 Spatial-Mixing Mechanism (Volume Conduction)

Volume conduction in scalp EEG spreads focal nonlinear activity across many sensor channels, increasing the fraction of channels that exceed significance thresholds. Scalp-level bicoherence is strongly affected by volume conduction, with substantial differences between source and sensor estimates (Shahbazi Avarvand et al., 2018); for instance, frontal sensors can reflect up to 75% contribution from occipital and central sources (Schaworonkow & Nikulin, 2022). Univariate normalization can mitigate coupling-dependent artifacts in bicoherence estimation (Shahbazi et al., 2014).

The forward-model simulation below supports this mechanism quantitatively: identical cortical sources yield 94.7% significant channels in EEG vs. 30.9% in iEEG.

2.3 S.3.3.3 Regional GL/NGL/GNL/NGNL Distribution

To complement the topographic and scatter-map visualizations in Fig. 3, all contacts (iEEG) or channel-subject pairs (EEG) are pooled by brain region, and GL/NGL/GNL/NGNL classification proportions are displayed as grouped bar charts with a shared y-axis.

For iEEG, the 39 fine-grained anatomical regions from the MNI atlas are merged into 8 lobe-level groups (Frontal, Temporal, Parietal, Occipital, Insular, Cingulate, Limbic/MTL, Central Opercu-

lum). For EEG, the 18 standard 10-20 channels are grouped into 5 scalp regions (Frontal: 7 channels; Central: 3; Temporal: 4; Parietal: 2; Occipital: 2).

Per-region GL/NGL/GNL/NGNL classification proportions for iEEG (a) and EEG (b).

Figure S.4: (a) iEEG: NGNL dominates all four major lobes (49–70%), peaking in occipital cortex; insular and cingulate regions show higher GL proportions (approximately 42%). (b) EEG: GNL is dominant in frontal and central regions (approximately 43–45%), while NGNL increases posteriorly (56–60% in parietal/occipital). Both modalities show an anterior-to-posterior NGNL gradient. The higher overall GNL proportion in EEG is consistent with volume conduction spreading focal nonlinear sources across multiple scalp channels (see simulation below).

2.4 S.3.3.4 Simulation Design

We constructed a forward-model simulation to test whether volume conduction can account for the higher prevalence of significant nonlinearity in scalp EEG (81.6%) compared to iEEG (67.9%). The simulation uses realistic leadfield matrices so that identical cortical sources are projected through both EEG and iEEG forward models, and channel-wise bicoherence significance is evaluated with the same statistical framework as the main analysis.

All simulations used a fixed random seed for exact reproducibility.

2.5 S.3.3.5 Source Model

Sources are defined on a 2003-vertex cortical mesh (Desikan-Killiany atlas, 2k resolution). For each vertex $j = 1, \dots, N_{\text{src}}$ ($N_{\text{src}} = 2003$), a background aperiodic process is generated as an AR(1) process:

$$E_j(t) \sim \mathcal{N}(0, \sigma_{\text{bg}}^2), \quad S_j^{\text{bg}}(t) = a S_j^{\text{bg}}(t-1) + E_j(t)$$

where $a = 0.98$ is the AR coefficient and $\sigma_{\text{bg}} = 1.0$ the driving noise amplitude. Each background source is then standardized to zero mean and unit variance.

Two types of oscillatory sources are injected into specific cortical regions of interest (ROIs):

Nonlinear mu source (parietal). A quadratic nonlinearity generates waveform asymmetry and harmonic content:

$$S_\mu(t) = \sin(2\pi f_\mu t) + \kappa \cdot [\sin(2\pi f_\mu t)]^2 + \sigma_\mu \eta_\mu(t)$$

where $f_\mu = 11$ Hz, $\kappa = 0.6$ controls the quadratic coupling strength, $\sigma_\mu = 0.5$ is the source noise amplitude, and $\eta_\mu(t) \sim \mathcal{N}(0, 1)$. The signal is standardized to zero mean and unit variance after generation.

Single-tone alpha source (occipital and frontal). No quadratic nonlinearity:

$$S_\alpha(t) = \sin(2\pi f_\alpha t) + \sigma_\alpha \eta_\alpha(t)$$

where $f_\alpha = 10$ Hz and $\sigma_\alpha = 0.05$. The signal is likewise standardized.

We note that modeling the alpha source as a single-tone sinusoid does not imply alpha oscillations are linear in general—symmetric (odd-order) nonlinear systems (e.g., Stuart-Landau limit cycles with cubic dynamics) can generate near-sinusoidal waveforms with negligible bicoherence. The simulation isolates the effect of quadratic nonlinearity on bicoherence detection.

The injected sources are additively superimposed on the background at ROI vertices defined by the Desikan-Killiany atlas:

ROI	Atlas Labels	Injected Signal	Vertices
Parietal (mu)	Inferior parietal L/R, Superior parietal L/R, Postcentral L/R	$S_\mu(t)$ (nonlinear)	~300
Occipital (alpha)	Pericalcarine L/R, Cuneus L/R, Lingual L/R	$S_\alpha(t)$ (single-tone)	~200
Frontal (alpha)	Superior frontal L/R	$S_\alpha(t)$ (single-tone)	~150

For each ROI vertex j :

$$S_j(t) = S_j^{\text{bg}}(t) + S_{\text{injected}}(t)$$

Vertices outside all ROIs retain only the background AR(1) process.

2.6 S.3.3.6 Forward Model

Sensor-level signals are computed as linear mixtures of cortical sources via realistic leadfield matrices, without additional sensor noise:

$$\mathbf{X}_{\text{EEG}}(t) = L_{\text{EEG}} \mathbf{S}(t), \quad \mathbf{X}_{\text{iEEG}}(t) = L_{\text{iEEG}} \mathbf{S}(t)$$

$L_{\text{EEG}} \in \mathbb{R}^{19 \times 2003}$: OpenMEEG BEM headmodel on the same 2k cortical grid. The original gain matrix $G \in \mathbb{R}^{N_{\text{ch}} \times 3N_{\text{src}}}$ (three orientations per source) is collapsed to a scalar leadfield along the cortical surface normal:

$$L_{\text{EEG}}(i, j) = G_j^{(i)} \cdot \hat{n}_j$$

where $G_j^{(i)} \in \mathbb{R}^{1 \times 3}$ is the gain for channel i at source j and \hat{n}_j is the surface-normal orientation vector.

$L_{\text{iEEG}} \in \mathbb{R}^{1766 \times 2003}$: Super-subject leadfield constructed by vertically stacking per-subject leadfield matrices from all 106 subjects in the Frauscher et al. (2018) atlas. Channels with all-zero leadfield rows (no sensitivity to any cortical source) are excluded, yielding 1766 channels. The empirical analysis uses a different preprocessing pipeline and retains 1,771 channels; this small discrepancy (1,766 vs 1,771) reflects differences in channel exclusion criteria between the simulation and empirical analysis and does not affect the qualitative conclusions.

Both leadfield matrices use raw amplitudes (no row-normalization). The observed signal at each channel is the linear combination of all cortical source signals weighted by the leadfield row.

2.7 S.3.3.7 Signal Parameters

Parameter	Symbol	Value
Sampling rate	F_s	200 Hz
Signal length	N	11,600 samples (58 s)
Random seed	—	1 (deterministic)
AR(1) coefficient	a	0.98
Background noise amplitude	σ_{bg}	1.0
Mu fundamental frequency	f_μ	11 Hz
Alpha fundamental frequency	f_α	10 Hz
Quadratic coupling strength	κ	0.6
Mu source noise	σ_μ	0.5
Alpha source noise	σ_α	0.05
Source grid vertices	N_{src}	2003
EEG channels	$N_{\text{ch}}^{\text{EEG}}$	19
iEEG channels (super-subject)	$N_{\text{ch}}^{\text{iEEG}}$	1766

2.8 S.3.3.8 Bicoherence Analysis

For each channel, the bispectrum is estimated using multitaper spectral analysis with parameters matched to the main empirical analysis:

Parameter	Value
Window length	300 samples (1.5 s)
FFT length (N_{fft})	300
Overlap	75%
Frequency range	0–45 Hz
Spectral method	PMTM (multitaper)
Time-bandwidth product (NW)	1.5 ($K = 2NW - 1 = 2$ tapers)
Bicoherence normalization	Haubrich (1965)
Segment sampling	Random, 157 segments per channel

The bispectrum and normalized bispectrum are defined as:

$$B(f_1, f_2) = \mathbb{E}[X(f_1) X(f_2) X^*(f_1 + f_2)]$$

$$b(f_1, f_2) = \frac{B(f_1, f_2)}{\sqrt{S(f_1) S(f_2) S(f_1 + f_2)}}$$

$$|b(f_1, f_2)| = \frac{|B(f_1, f_2)|}{\sqrt{S(f_1) S(f_2) S(f_1 + f_2)}}$$

where $S(f)$ denotes the power spectrum (Haubrich normalization).

Channel-wise significance is assessed using two complementary test statistics (see R1-10 for full derivation):

Nonlinearity (max statistic):

$$t^{(L)} = 2N_s^{\text{eff}} \cdot \max_{f_1, f_2} |b(f_1, f_2)|^2$$

Non-Gaussianity (median statistic):

$$t^{(G)} = 2N_s^{\text{eff}} \cdot \text{median}_{f_1, f_2} |b(f_1, f_2)|^2$$

The effective number of independent segments N_s^{eff} is estimated per channel via an AR-surrogate bootstrap procedure ($M = 100$ iterations, KDE-based density estimation; cache disabled to ensure

per-channel estimation). The significance level is $\alpha = 0.001$, with the critical value $c_\alpha^{(L)}$ derived from extreme value theory (Gumbel distribution; see R1-10).

2.9 S.3.3.9 Results

Metric	iEEG	EEG
Number of channels	1,766	19
Fraction significant (max statistic)	30.9%	94.7%

Figure S.5: Violin plot of channel-wise bicoherence statistics for iEEG and EEG. Left pair: nonlinearity ($b^{(L)}$, max statistic); right pair: Gaussianity ($b^{(G)}$, median statistic). Dashed lines mark significance thresholds ($\alpha = 0.001$). EEG shows 94.7% channels significant for nonlinearity vs. 30.9% in iEEG; neither modality shows significant non-Gaussianity (0.0%).

Even with identical underlying cortical sources, spatial mixing inherent to scalp EEG produces a substantially larger proportion of channels labeled as nonlinear (94.7% vs. 30.9%). The channel count asymmetry (19 EEG vs. 1,766 iEEG) mirrors the inherent difference between these recording modalities and is consistent with the empirical data (18 scalp EEG channels vs. 1,771 iEEG channels); importantly, each channel's bicoherence significance is tested independently, so the per-channel test is unaffected by the total number of channels. This supports the interpretation that the higher prevalence of significant nonlinearity in EEG compared to iEEG (81.6% vs. 67.9% in the empirical data) is consistent with volume conduction amplifying the detectability of quadratic phase coupling at the sensor level.

3 Supplementary Section S.7: Complementary Approaches to the Linear-Brain Question (R1-18)

Three recent methodologies address whether macroscopic brain signals are adequately described by linear Gaussian models. They operate in different statistical domains and should therefore be interpreted as complementary rather than mutually exclusive.

	Nozari et al. (2024)	Tani Raffaelli et al. (2024)	BiSCA (this work)
Domain	Time	Information-theoretic	Frequency
Statistic	Prediction R^2 (linear vs. nonlinear AR)	Relative Non-Linearity: $RNL = 1 - MI_{\text{Gauss}}/MI_{\text{total}}$	Bicoherence $\hat{b}(f_1, f_2)$ tested against $\chi^2(2)$ null
Null hypothesis	Linear AR model sufficient	Gaussian copula sufficient	Linear Gaussian process (no quadratic phase coupling)
Frequency resolution	None	None	Full (f_1, f_2) bifrequency map
Separates nonlinearity from non-Gaussianity	No	No	Yes, within the quadratic-bicoherence setting
Data modalities	fMRI, iEEG	Spikes, iEEG, EEG, fMRI	iEEG, EEG

Time-domain prediction frameworks such as Nozari et al. (2024) test whether nonlinear predictors improve forecast accuracy over linear baselines. They therefore address predictive sufficiency in the time domain, but they are not designed to localize phase-coupled interactions across bifrequency space.

Information-theoretic frameworks such as Tani Raffaelli et al. (2024) quantify dependence beyond a Gaussian-copula baseline. Their sensitivity is broader, but that broader sensitivity does not by itself separate nonlinear dynamics from non-Gaussian innovations or non-stationary contributions, nor does it localize the effect in frequency space.

BiSCA addresses a narrower but more specific target, namely quadratic phase coupling. It provides bifrequency-resolved diagnostics and distinguishes frequency-flat bicoherence, consistent with non-Gaussian input, from peaked bicoherence, consistent with quadratic coupling. Its scope is therefore complementary rather than universal, because it is designed for quadratic interactions and not for arbitrary higher-order nonlinear structure.

—# References

- Bartz, S., Avarvand, F. S., Leicht, G., & Nolte, G. (2019). Analyzing the waveshape of brain oscillations with bicoherence. *NeuroImage*, 188, 145-160.
- Bender, J. M., Voytek, B., & Schaworonkow, N. (2023). Resting-state is not enough: alpha and mu rhythms change shape across development and task. *Developmental Cognitive Neuroscience*, 64, 101340.
- Boncompte, G., Medel, V., Irani, M., Lachaux, J. P., & Ossandon, T. (2025). Aperiodic exponent of brain field potentials is dependent on the frequency range it is estimated. *IEEE J. Biomed. Health Inform.* DOI: 10.1109/JBHI.2025.3566118.
- Brillinger, D. R. (1965). An introduction to polyspectra. *The Annals of Mathematical Statistics*, 36(5), 1351-1374.
- Brillinger, D. R. (2001). *Time Series: Data Analysis and Theory*. SIAM.
- Chandran, V., Elgar, S., & Pezeshki, C. (1993). Bispectral and trispectral characterization of transition to chaos in the Duffing oscillator. *International Journal of Bifurcation and Chaos*, 3(3), 551-557.
- Clauset, A., Shalizi, C. R., & Newman, M. E. J. (2009). Power-law distributions in empirical data. *SIAM Review*, 51(4), 661-703.
- Cole, S. R., & Voytek, B. (2017). Brain oscillations and the importance of waveform shape. *Trends in Cognitive Sciences*, 21(2), 137-149.
- Donoghue, T., Haller, M., Peterson, E. J., et al. (2020). Parameterizing neural power spectra into periodic and aperiodic components. *Nature Neuroscience*, 23(12), 1655-1665.
- Donoghue, T., Schaworonkow, N., & Voytek, B. (2022). Methodological considerations for studying neural oscillations. *European Journal of Neuroscience*, 55, 3502-3527.
- Frauscher, B., von Ellenrieder, N., Zemann, R., et al. (2018). Atlas of the normal intracranial electroencephalogram: neurophysiological awake activity in different cortical areas. *Brain*, 141(4), 1130-1144.
- Hu, S., Zhang, Z., Zhang, X., Wu, X., & Valdes-Sosa, P. A. (2024). Xi-Pi: A nonparametric model for neural power spectra decomposition. *IEEE Journal of Biomedical and Health Informatics*, 28(5), 2624-2635. DOI: 10.1109/JBHI.2024.3364499.
- Kartashova, E. (2010). *Nonlinear Resonance Analysis: Theory, Computation, Applications*. Cambridge University Press.
- Kramer, M. A., & Chu, C. J. (2024). A general, noise-driven mechanism for the 1/f-like behavior of neural field potentials. *Cell Reports*, 43(3), 113878.
- Li, M., et al. (2022). Harmonized-Multinational qEEG norms (HarMNqEEG). *NeuroImage*, 256, 119190.
- Nichols, J. M., Olson, C. C., Michalowicz, J. V., & Bucholtz, F. (2009). The bispectrum and bicoherence for quadratically nonlinear systems subject to non-Gaussian inputs. *IEEE Trans. Signal Process.*, 57(10), 3879-3890.
- Nozari, E., Bertolero, M. A., Stiso, J., Caciagli, L., Cornblath, E. J., He, X., Mahadevan, A. S., Pappas, G. J., & Bassett, D. S. (2024). Macroscopic resting-state brain dynamics are best described by linear models. *Nature Biomedical Engineering*, 8, 68-84.
- Rajasekar, S., & Sanjuan, M. A. F. (2016). *Nonlinear Resonances*. Springer.
- Schaworonkow, N. (2023). Overcoming harmonic hurdles: genuine beta-band rhythms vs. har-

monic contamination. *Imaging Neuroscience*, 1, 1-16.

- Schaworonkow, N., & Nikulin, V. V. (2022). Is sensor space analysis good enough? Spatial patterns as a tool for assessing spatial mixing of EEG/MEG rhythms. *NeuroImage*, 253, 119093.
- Shahbazi Avarvand, F., Bartz, S., Engel, A. K., Leicht, G., Mulert, C., & Nolte, G. (2018). Localizing bicoherence from EEG and MEG signals. *NeuroImage*, 174, 352-363.
- Shahbazi, F., Ewald, A., Nolte, G. (2014). Univariate normalization of bispectrum using Holder's inequality. *Journal of Neuroscience Methods*, 233, 177-186.
- Tani Raffaelli, G., Jiricek, S., & Hlinka, J. (2024). Nonlinear brain connectivity from neurons to networks: quantification, sources and localization. *PNAS*, 121(51), e2411230121.
- Valdes, P., Biscay, R., Galan, L., Bosch, J., Szava, S., & Virues, T. (1992). High resolution spectral EEG norms topographic maps. *Brain Topography*, 4(4), 309-319.

Response to Reviewer #2 — Round 2

Step-by-Step Derivation: From Triple-Indexed Bispectrum to Pair-Indexed BiSCA Parametrization (Eq. 15)

Summary

We thank the reviewer for the continued engagement with the mathematical details of our method. The reviewer’s central concern is that the reduction from the K^3 -term triple sum (Eq. R.12) to the pair-indexed double sum of Eq. (15) was not justified by a step-by-step derivation in the manuscript. Specifically, the reviewer noted that for $K = 2$ one obtains 8 independent triple-sum terms and “no symmetries to reduce them to the $K(K+1)/2 = 3$ terms,” and that for $K = 6$ the three counts we cited (15 triads, 9 pairs, 21 formal pairs) appeared inconsistent.

In this revision we provide the “clear and detailed step-by-step derivation” requested by the reviewer, presented as a new Supplementary Section S.5.2.

Reviewer #2:

I am still concerned that their reasoning is deficient for how the triple sum of my equation (12) reduces to the double sum of their equation (15). [...] a simple calculation of the triple sum for $K = 2$ reveals 8 independent components, which as far as I can tell admits of no symmetries to reduce them to the $K(K+1)/2 = 3$ terms as implied by the double sum of their equation (15). [...] it behoves the authors to present a clear and detailed step-by-step derivation of their novel method in situ in the supplementary material.

Response. The reviewer looked for an algebraic symmetry of the third-order tensor $\mathbb{E}[X_m X_n X_k^*]$ that would reduce three free indices to two—and correctly found that no such symmetry exists. Our reduction takes a different path: it relies not on tensor symmetry but on the **narrowband structure of the spectral peaks**. The product of three narrowband kernels $t(f_1; \mu_m) t(f_2; \mu_n) t(f_1+f_2; \mu_k)$ is non-negligible only when $\mu_k \approx \mu_m + \mu_n$, which uniquely determines the third index k from the pair (m, n) . This is the closure condition inherited from the bispectral definition $B(f_1, f_2) = \mathbb{E}[X(f_1) X(f_2) X^*(f_1+f_2)]$ (Brillinger & Rosenblatt, 1967b), applied to discrete narrowband components.

We acknowledge that our previous response did not make this distinction sufficiently clear. The new Supplementary Section S.5.2 provides the complete derivation, enumerating every term for $K = 2$ and $K = 6$ and reconciling the three counts (15, 9, and 21) that appeared inconsistent.

This note provides a self-contained derivation showing how the triple-indexed bispectral expansion (Eq. R.12, containing K^3 terms) reduces to the pair-indexed BiSCA model (manuscript Eq. 15). All terms are enumerated for $K = 2$ and $K = 6$, reconciling the three counts from our previous response—15 admissible triads, 9 unique closure-admissible pairs, and 21 formal pairs in Eq. (15).

Response to Reviewer #2

The influence of nonlinear resonance on human cortical oscillations

Communications Biology

Contents

1	Detailed Response to Reviewer #2	3
2	Overall Response	3
3	Major Comments	4
3.1	Major 1: Independence and Additivity	4
3.2	Major 2: Missing Indices in Eq.(13)	5
3.3	Major 3: Triple Index vs Pair Index	7
3.4	Major 4: Frequency Dependence of h_B	8
3.5	Major 5: Flat-Pattern Interpretation (B, b, and Signed Panels)	9
4	Minor Comments	11
4.1	Minor 1: Data Preprocessing Parameters	11
4.2	Minor 2: Laplacian Re-referencing	13
4.3	Minor 3: Signed Panels and Negative Values in Fig.1	14
4.4	Minor 4: Triangular Domain	14
4.5	Minor 5: Parameter d Values	15
4.6	Minor 6: Eyes-Open vs Eyes-Closed Comparison	15
4.7	Minor 7: Number of Oscillatory Processes K	16
5	Supplementary Section S.5.2: Pair-Indexed Bispectrum Parametrization and Triad Sparsity	16
5.1	Triad sparsity under harmonic constraint	17
5.2	Equation mapping to reviewer’s derivation	17
6	Supplementary Section S.6.3: Independence Assumption ($\xi \perp\!\!\!\perp \rho$): Mathematical Basis, Physiological Motivation, and Numerical Validation	20
6.1	Mathematical basis	20
6.2	Neurophysiological motivation	20
6.3	Empirical dissociation	21
6.4	Numerical validation	21
6.5	Limitation	23
7	Supplementary Section S.4.2: Model-Order and Parameter Accounting for AIC	

Selection	23
8 References	24
8.1 Theoretical Framework	24
8.2 Methods	24
8.3 Neurophysiological Motivation	24
8.4 Empirical Dissociation	25
8.5 Limitations	25

Manuscript: The influence of nonlinear resonance on human cortical oscillations

Journal: Communications Biology

1 Detailed Response to Reviewer #2

2 Overall Response

Reviewer's Overall Assessment:

This is an interesting and potentially very important communication in which the authors aim to decompose resting electroencephalographic (EEG and ECoG) activity into aperiodic and oscillatory activity using a newly developed method that they refer to as BiSpectral EEG Component Analysis (BiSCA). This is of particular relevance given our incomplete understanding regarding the extent to which resting macroscopic cortical electrophysiological activity is composed of Gaussian random activity and non-linear dynamical activity, a resolution that has considerable importance in our ability to understand and model the dynamical genesis of macroscopic brain activity such that we can better detect its perturbation in health and disease.

However, as described their method is not sufficiently clear to enable independent replication, an obvious requirement for the communication of a novel and potentially important method. Further, corresponding ambiguities in their description potentially imply that their conclusions cannot be regarded as in any sense definitive, in particular that aperiodic background activity (assuming it meaningfully exists) exhibits no evidence of quadratic cross-frequency coupling.

My major concern centres around the unclear derivation of BiSCA in section 4.2.4. By attempting my own derivation I find multiple errors in the corresponding equations and some of the assumptions, particularly those regarding the independence of the various theoretically defined 'oscillatory' processes. In order to avoid any misunderstanding I outline, in some detail, my own derivation to highlight what I believe are significant problems with their formulation as presented.

Changes Made:

We revised Section 4.2.4 to explicitly state the $\xi \perp \rho$ independence assumption and its consequences for cumulant additivity (Major 1); corrected index notation in Eq.(13) (Major 2); clarified the pair-indexed closure-manifold parametrization vs. the reviewer's triple-index expansion (Major 3–4); and added an explicit distinction between bispectrum B and bicoherence $|b|$ (Major 5).

Response:

The reviewer's detailed re-derivation highlighted that our presentation of the independence assumption in Section 4.2.4 was unclear. We revised the manuscript accordingly. Previous peri-

odic/aperiodic decomposition methods (e.g., speccparam, IRASA) assumed additive power spectra, which required second-order uncorrelatedness between ξ and ρ . BiSCA extended this to full statistical independence ($\xi \perp\!\!\!\perp \rho$) because bispectral (third-order) additivity requires cumulant factorization at second and third order. This is a modeling assumption; see Major 1 for formal proofs and Supplementary Section S.6.3 for numerical validation.

3 Major Comments

3.1 Major 1: Independence and Additivity

Reviewer’s Comment:

It seems that their starting point is to assume that EEG/ECOG activity is the sum of two, not necessarily independent, random processes

$$x(t) = x_\xi(t) + x_\rho(t) \quad (\text{R.1})$$

$$= x_\xi(t) + \sum_k x_{\rho,k}(t) \quad (\text{R.2})$$

where $x_\xi(t)$ corresponds to the ‘aperiodic’ process and $x_{\rho,k}$ the k -th ‘oscillatory process’. Further, it seems to be assumed (though nowhere obviously stated) that the Fourier transform of each of these $k + 1$ processes is modelled as

$$x_\xi(t) \leftrightarrow X_\xi(f) \approx h_\xi^{1/2} t_\xi(f) \exp[i\phi_\xi(f)] \quad (\text{R.3})$$

$$x_{\rho,k}(t) \leftrightarrow X_{\rho,k}(f) \approx h_{\rho,k}^{1/2} t_{\rho,k}(f) \exp[i\phi_{\rho,k}(f)] \quad (\text{R.4})$$

where the respective amplitudes, $t_\xi(f), t_{\rho,k}(f)$ are defined parametrically according to their equation 14, while the parametric forms for $\phi_\xi(f), \phi_{\rho,k}(f)$ are left undefined.

From this basis the power spectrum will be

$$X(f)X^*(f) = \left[X_\xi(f) + \sum_k X_{\rho,k}(f) \right] \left[X_\xi^*(f) + \sum_k X_{\rho,k}^*(f) \right] \quad (\text{R.5})$$

which will only be equal to

$$X_\xi(f)X_\xi^*(f) + \sum_k X_{\rho,k}(f)X_{\rho,k}^*(f) \quad (\text{R.6})$$

iff all cross-spectral densities are zero. This does not seem to make sense as the inter-frequency phase couplings, to be evaluated using the bispectrum, necessarily assume that $X_{\rho,l}X_{\rho,m}^* \neq 0$.

Changes Made:

Section 4.2.4, paragraph following Eq.(15): > “The ξ and ρ processes are assumed statistically

independent ($\xi \perp \rho$), which is the standard assumption in the periodic/aperiodic decomposition literature. Under independence, cumulants of all orders decompose additively (Brillinger, 1965, Eq. 4.11): $S_x = S_\xi + S_\rho$ (2nd order) and $B_x = B_\xi + B_\rho$ (3rd order). The bispectrum model (Eq. 15) retains free complex amplitudes for both the aperiodic ($h_{B,\xi}$) and oscillatory ($h_{B,\rho,m,n}$) bispectral contributions; the Gaussian linear (GL) interpretation ($h_{B,\xi} = 0$) is tested by the fit rather than imposed as a structural constraint. Bispectral coupling—a 3rd-order statistic on the closure manifold $\{(f_1, f_2, f_1+f_2)\}$ —is mathematically independent of same-frequency cross-spectra between oscillatory components.”

Response:

The key distinction is statistical order: same-frequency cross-spectrum $E[X_{\rho,l}(f)X_{\rho,m}^*(f)]$ is 2nd order on the diagonal $\{(f, f)\}$, whereas the bispectrum $B(f_1, f_2) = E[X(f_1)X(f_2)X^*(f_1+f_2)]$ is 3rd order on the closure manifold $\{(f_1, f_2, f_1+f_2)\}$. Nonzero bispectral coupling does not require nonzero same-frequency cross-spectrum; a quadratic Volterra kernel with Gaussian input produces nonzero bispectrum on closure triads while the cross-spectrum remains zero (Marzocca et al., 2008, Eq. 24). The transition from (R.5) to (R.6) would additionally require mutual independence among all peak components $\{x_{\rho,k}\}$ —a stronger assumption we do not make; see Major 2–3. Importantly, the power spectrum model (Eq. 13) is a parametric basis expansion—a sum of kernel functions placed at detected spectral peaks—not a consequence of mutual independence among $\{x_{\rho,k}\}$. Because the peaks are narrow-band and spectrally separated by construction, cross-spectral terms between distinct ρ_k components are empirically negligible; moreover, even if such cross-terms were nonzero at 2nd order, this would not affect the 3rd-order cumulant decomposition $B_x = B_\xi + B_\rho$, which depends only on $\xi \perp \rho$ (Supplementary S.6.3, Theorem 1).

Mathematical proofs of cumulant additivity under $\xi \perp \rho$, physiological and empirical motivation for this assumption, and numerical validation (two simulation configurations yielding spectrum error $\leq 1.1\%$ and bispectrum error $\leq 10.3\%$) are provided in Supplementary Section S.6.3.

3.2 Major 2: Missing Indices in Eq.(13)

Reviewer’s Comment:

Nevertheless, assuming that Eqn.(R.6) holds, it can be rewritten in terms of their parametric form as

$$h_{S,\xi}t(f, 0, \sigma_\xi, \nu_\xi, d_\xi)^2 + \sum_k h_{S,\rho,k}t(f, \mu_{\rho,k}, \sigma_{\rho,k}, \nu_{\rho,k}, d_{\rho,k})^2 \quad (\text{R.7})$$

which is not their equation 13 as they have omitted the indices on the ‘oscillatory’ (ρ) parameters.

Changes Made:

1. Eq.(13): Added subscript k to center and amplitude parameters; retained shared shape: $>$
 $S_\rho(f) = \sum_{k=1}^K h_{S,\rho,k}t(f; \mu_{\rho,k}, \sigma_\rho, \nu_\rho, d_\rho)^2$

2. Following paragraph: Added clarification: \succ “Peak centers $\mu_{\rho,k}$ and amplitudes $h_{S,\rho,k}$ are peak-specific. Shape parameters $(\sigma_{\rho}, \nu_{\rho}, d_{\rho})$ are shared across all oscillatory peaks, reflecting a common kernel profile from the same underlying system and reducing the number of free parameters. An optional per-peak mode is available for applications where peak shapes differ across spectral neighborhoods. This parametric expansion does not assume independence between peak components within ρ .”

Response:

Indices should indeed be clarified. In BiSCA, peak centers and amplitudes are peak-specific (indexed by k), while shape parameters are shared:

- Per-peak: centers $\mu_{\rho,k}$, amplitudes $h_{S,\rho,k}$
- Shared: shape parameters $\sigma_{\rho}, \nu_{\rho}, d_{\rho}$ (common kernel profile)

The corrected equation is:

$$S_{\rho}(f) = \sum_{k=1}^K h_{S,\rho,k} t(f; \mu_{\rho,k}, \sigma_{\rho}, \nu_{\rho}, d_{\rho})^2$$

This is a parametric basis expansion, not a claim that each peak arises from an independent sub-process. The only independence assumption is $\xi \perp\!\!\!\perp \rho$. The shared shape parameters are consistent with a common kernel profile from a Wiener model structure (Marzocca et al., 2008, Eq. 16; Schetzen, 1980, Eq. 7.3-8).

3.3 Major 3: Triple Index vs Pair Index

Reviewer's Comment:

By assuming that the ' ξ ' and ' ρ ' processes are phase uncorrelated the theoretical bispectrum $B(f_1, f_2)$ is

$$B(f_1, f_2) = [X_\xi(f_1) + X_\rho(f_1)][X_\xi(f_2) + X_\rho(f_2)][X_\xi^*(f_1 + f_2) + X_\rho^*(f_1 + f_2)] \quad (\text{R.8})$$

$$= X_\xi(f_1)X_\xi(f_2)X_\xi^*(f_1 + f_2) + X_\rho(f_1)X_\rho(f_2)X_\rho^*(f_1 + f_2) \quad (\text{R.9})$$

$$= B_\xi(f_1, f_2) + B_\rho(f_1, f_2) \quad (\text{R.10})$$

Which, by using the parametric forms of Eqns. (R.3) and (R.4) can be written as

$$B(f_1, f_2) = h_\xi^{3/2} t_\xi(f_1) t_\xi(f_2) t_\xi(f_1 + f_2) \exp[i(\phi_\xi(f_1) + \phi_\xi(f_2) - \phi_\xi(f_1 + f_2))] \quad (\text{R.11})$$

$$+ \sum_{k,l,m} h_{\rho,k}^{1/2} h_{\rho,l}^{1/2} h_{\rho,m}^{1/2} t_{\rho,k}(f_1) t_{\rho,l}(f_2) t_{\rho,m}(f_1 + f_2) \exp[i(\phi_{\rho,k}(f_1) + \phi_{\rho,l}(f_2) - \phi_{\rho,m}(f_1 + f_2))]$$

which in terms of their parameterisations and symbols can be written as

$$B(f_1, f_2) = h_{B,\xi}(f_1, f_2) t_\xi^2(f_1, f_2, \sigma_\xi, \nu_\xi, d_\xi) \quad (\text{R.12})$$

$$+ \sum_{k,l,m} h_{B,\rho,lmn}(f_1, f_2) t_\rho^2(f_1, f_2, \mu_{\rho,k}, \mu_{\rho,l}, \mu_{\rho,m}, \dots)$$

where

$$t_\xi^2(f_1, f_2, \sigma_\xi, \nu_\xi, d_\xi) = t(f_1, 0, \sigma_\xi, \nu_\xi, d_\xi) t(f_2, 0, \sigma_\xi, \nu_\xi, d_\xi) t(f_1 + f_2, 0, \sigma_\xi, \nu_\xi, d_\xi) \quad (\text{R.13})$$

$$t_\rho^2(f_1, f_2, \mu_{\rho,k}, \mu_{\rho,l}, \mu_{\rho,m}, \sigma_{\rho,k}, \sigma_{\rho,l}, \sigma_{\rho,m}, \nu_{\rho,k}, \nu_{\rho,l}, \nu_{\rho,m}, d_{\rho,k}, d_{\rho,l}, d_{\rho,m}) =$$

$$t(f_1, \mu_{\rho,k}, \sigma_{\rho,k}, \nu_{\rho,k}, d_{\rho,k}) t(f_2, \mu_{\rho,l}, \sigma_{\rho,l}, \nu_{\rho,l}, d_{\rho,l}) t(f_1 + f_2, \mu_{\rho,m}, \sigma_{\rho,m}, \nu_{\rho,m}, d_{\rho,m}) \quad (\text{R.14})$$

and where the 'constants' $h_{B,\dots}$ will be complex and unless otherwise specified will depend on the frequencies f_1 and f_2 .

Changes Made:

1. Section 4.2.4: Added explanation: > “*BiSCA uses a pair-indexed parametrization on the closure manifold, following established precedent in higher-order spectral analysis (Brillinger & Rosenblatt, 1967b, pp. 193–195; Nikias & Petropulu, 1993, Ch. 2). Under the harmonic constraint $\mu_k = kf_0$, the closure condition $\mu_k + \mu_l = \mu_m$ reduces the triple index (k, l, m) to a sparse set of admissible pairs (e.g., $K=6$: 216 \rightarrow 15 triads \rightarrow 9 unique pairs).*”
2. Supplementary Section S.5.2: Added pair-indexed parametrization derivation, triad sparsity analysis, and equation mapping (Figs. R2-4, R2-5).

Response:

BiSCA uses a pair-indexed parametrization on the closure manifold rather than the reviewer’s triple-index expansion $\sum_{k,l,m}$. Under the harmonic constraint $\mu_k = kf_0$, the closure condition $\mu_k + \mu_l = \mu_m$ reduces K^3 candidate terms to a sparse set of admissible pairs—a direct consequence of the closure structure (Brillinger & Rosenblatt, 1967b, p. 193; Nikias & Petropulu, 1993, Section 2.3). This is a basis expansion on the closure manifold, not a generative independence claim about sub-processes within ρ . The detailed parametrization, triad sparsity analysis, numerical validation, and point-by-point equation mapping to the reviewer’s derivation are provided in Supplementary Section S.5.2.

3.4 Major 4: Frequency Dependence of h_B

Reviewer’s Comment:

In their formulation the $h_{B,\dots}$ appear to be independent of frequency, which makes me wonder if they have assumed that the $\phi_\xi(f)$ and the $\phi_{\rho,k}(f)$ are constant. If this is the case then for $\nu_\xi = \nu_{\rho,k} = 1$ and $d_\xi = d_{\rho,k} = 2$ (i.e., Lorentzian) we have

$$e^{-4\pi^2\sigma_\xi^2|\tau|}e^{i\phi_\xi} \leftrightarrow X_\xi(f, \sigma_\xi) \quad (\text{R.15})$$

$$e^{-4\pi^2\sigma_{\rho,k}^2|\tau|}e^{-i(2\pi\mu_{\rho,k}\tau - \phi_{\rho,k})} \leftrightarrow X_{\rho,k}(f, \mu_{\rho,k}, \sigma_{\rho,k}) \quad (\text{R.16})$$

Changes Made:

1. Section 4.2.4: Clarified that h_B is per-peak-pair: > “Each peak pair (m, n) has an independent complex amplitude $h_{B,\rho,m,n} \in \mathbb{C}$, allowing the model to capture heterogeneous coupling strengths across the bispectrum. The kernel basis functions t_2 capture the frequency-dependent shape on the closure manifold.”
2. Supplementary Section S.6: Added a section showing the relationship between our parametrization and the rigorous Volterra-series form (Marzocca et al., 2008; Nichols et al., 2009).
3. Supplementary Section S.6.4 (new): Added a dedicated subsection on the kernel function and its dynamical-system interpretation, formalizing the Lorentzian special case ($\nu = 1$, $d = 2$) as an Ornstein–Uhlenbeck process (ξ , $\mu = 0$) and a damped harmonic oscillator (ρ , $\mu > 0$), clarifying that $d > 2$ absorbs multitaper spectral smoothing rather than representing an independent dynamical parameter, and discussing the general kernel extension.

Response:

The reviewer is correct that each $h_{B,m,n}$ is a frequency-independent complex scalar. Our full bispectrum model (Eq.~15) includes both ξ and ρ terms:

$$B(f_1, f_2) \approx h_{B,\xi} t_2(f_1, f_2; \mu_\xi, \mu_\xi, \theta_\xi) + \sum_{m=1}^K \sum_{n=1}^m h_{B,\rho,m,n} t_2(f_1, f_2; \mu_m, \mu_n, \theta_\rho),$$

where $h_{B,\xi} \in \mathbb{C}$ is the free complex amplitude for the aperiodic bispectral contribution and $\theta_\rho = (\sigma_\rho, \nu_\rho, d_\rho)$ are the shape parameters inherited from the spectrum fit. Each peak pair (m, n) carries its own $h_{B,\rho,m,n} \in \mathbb{C}$, while frequency dependence comes from the kernel t_2 on the closure manifold. For any single kernel, $\arg(h_{B,m,n})$ is constant; however, since different peak pairs contribute at different bifrequency locations with different complex amplitudes, $\arg(\hat{B}(f_1, f_2))$ varies spatially. The ξ term enables the model to detect non-Gaussian linear (NGL) aperiodic activity ($h_{B,\xi} \neq 0$); the empirical finding $h_{B,\xi} \approx 0$ supports the Gaussian linear interpretation.

We do not parameterize individual Fourier phase functions $\phi_\xi(f)$ or $\phi_{\rho,k}(f)$; our model operates directly at the bispectrum level, where $h_{B,m,n}$ absorbs the net phase relationship per peak pair. Numerical validation: spectrum $R_S^2 > 99.9\%$; bispectrum $|B| R^2$: 96.1% (Config 1), 96.3% (Config 2); phase MAE ≤ 0.01 rad, coherence > 0.999 for both configurations (Fig. R2-7).

We also agree with the reviewer’s derivation (Eqs. R.15–R.16): in the Lorentzian special case ($\nu = 1$, $d = 2$), the ξ kernel corresponds to the power spectrum of an Ornstein–Uhlenbeck (OU) process—a continuous-time AR(1) with autocorrelation $R_\xi(\tau) \propto e^{-\lambda|\tau|}$, $\lambda = 2\pi\sigma_\xi$ —and each ρ kernel to a damped harmonic oscillator with autocorrelation $R_{\rho,k}(\tau) \propto e^{-\lambda_k|\tau|} \cos(2\pi\mu_{\rho,k}\tau)$. This connection anchors our parametrization to well-understood dynamical systems. For general (ν, d) , the kernel extends this: ν controls the tail decay rate (heavier tails for $\nu < 1$, sharper peaks for $\nu > 1$), while $d > 2$ absorbs the flat-top broadening introduced by multitaper spectral smoothing (bandwidth $2NW/T$) rather than representing an independent dynamical-system parameter. The complete derivation and discussion are provided in the new Supplementary Section S.6.4.

3.5 Major 5: Flat-Pattern Interpretation (B, b, and Signed Panels)

Reviewer’s Comment: i.e., the $x_\xi(t)$ process is effectively a noise driven decaying non-oscillatory exponential and the $x_{\rho,k}(t)$ processes are effectively noise driven damped oscillations of frequency $\mu_{\rho,k}$. On this basis the bispectrum for the $x_\xi(t)$ will be trivially flat, as I notice seems to be the case in their Fig.1F.

Comparison between my Eqn (R.12) and their equation (15) when the real and imaginary parts are taken, and comparison between their auxiliary function t^2 and my auxiliary functions t_ξ^2 and t_ρ^2 show that they are quite different, so much so that I cannot conclude that they have calculated a meaningful theoretical bispectrum to fit to empirical data.

Changes Made:

1. Fig.1 caption: Added unified quantity clarification: *> “Panels displaying $\text{Re}\{\hat{b}(f_1, f_2)\}$ are signed and can be negative. Magnitude bicoherence $|\hat{b}(f_1, f_2)|$ is nonnegative and bounded in $[0, 1]$. A flat pattern in $|\hat{b}|$ does not imply zero raw bispectrum $\hat{B}(f_1, f_2)$.”*
2. Section 4.2.3: Added explicit distinction: *> “We distinguish between the bispectrum $B(f_1, f_2)$ (unnormalized, complex), the normalized bispectrum $b(f_1, f_2) = B/\sqrt{S_1 S_2 S_3}$ (complex), and the bicoherence $|b(f_1, f_2)|$ (magnitude, real, $\in [0, 1]$). Flat bicoherence can indicate either a*

Gaussian linear process ($|b| = 0$) or a non-Gaussian linear process ($|b| = \text{const} > 0$); in both cases the bispectrum carries diagnostic information about the underlying system class.”

Response:

In our figures, signed panels (e.g., Fig.1D–G) display $\text{Re}\{b\}$, while bicoherence-magnitude panels display $|b|$. A “flat” pattern in $|b|$ does not imply that B is trivial or zero. Using the canonical taxonomy (Brillinger, 2001; Nichols et al., 2009; Berg et al., 2010):

Gaussian ($\gamma_3 = 0$)	Non-Gaussian ($\gamma_3 \neq 0$)
Linear: $ b = 0$	NGL: $ b = \gamma_3 /\sigma^3$ (constant)
Nonlinear: $ b = f(H_2)$ (peaks)	NGNL: constant background + peaks

A flat $|b|$ is a linear-class signature (GL or NGL), not evidence that B is trivial. For an NGL process, the bispectrum has structured frequency dependence shaped by the linear filter, while bicoherence is approximately constant, as illustrated by an NGL simulation (AR(1) $a=0.95$, Pearson-III innovation, $T=180$ s): both $|B|$ and denominator $\sqrt{S_1 S_2 S_3}$ exhibit a $1/f$ slope while their ratio $|b|$ remains approximately constant (Fig. R2-6).

Regarding the reviewer’s dynamical-system interpretation: we confirm that in the Lorentzian special case ($\nu = 1$, $d = 2$), the ξ kernel corresponds to an OU process and each ρ kernel to a damped harmonic oscillator, exactly as the reviewer derives in Eqs. (R.15)–(R.16). Importantly, the Lorentzian spectral shape characterizes the linear transfer function $|H_1(f)|$ regardless of whether the driving noise is Gaussian or non-Gaussian; under an NGL (non-Gaussian linear) input, the spectrum remains Lorentzian while the bispectrum becomes $B_\xi = H_1(f_1)H_1(f_2)H_1^*(f_1+f_2)\gamma_3 \neq 0$. Our bispectrum model (Eq.~15) retains a free complex amplitude $h_{B,\xi}$ for the ξ contribution, so the model can represent both GL ($h_{B,\xi} = 0$) and NGL ($h_{B,\xi} \neq 0$). The empirical finding $h_{B,\xi} \approx 0$ —consistent with the flat near-zero pattern in Fig.1F—provides data-driven support for the GL interpretation rather than imposing it as a structural constraint. The full derivation and the general-kernel extension are given in the new Supplementary Section S.6.4.

Regarding the structural mismatch between Eq.(R.12) and our Eq.(15): our Eq.(15) is obtained from the reviewer’s general triple-index expansion (Eq. R.12) by restricting to the harmonic closure manifold $\{(k, l, m) : \mu_k + \mu_l = \mu_m\}$, which reduces the triple index to a sparse set of admissible pairs. The complete term-by-term mapping is provided in Supplementary Section S.5.2 (Table S.5). The empirical evidence supports practical adequacy of this constrained parametrization (see Major 4 for detailed fit statistics). Population-level: median $R_B^2 \approx 0.55$, IQR [0.44, 0.66] (manuscript Fig. S6). An additional visual comparison between empirical and fitted bispectra for the two validation configurations is provided in Supplementary Figure S.14.

Fig. R2-6. Bicoherence decomposition for NGL process

4 Minor Comments

4.1 Minor 1: Data Preprocessing Parameters

Reviewer's Comment:

The parameters around the resting EEG data used are not clear. It appears that resting data was recorded using multiple different systems, but there is no specification of sampling frequencies, resampling, band pass filtering, artefact rejection or any other form of preprocessing. The only information we can infer is that recordings were sampled/resampled at/to 200 Hz.

Changes Made:

1. Methods Section 4.1: Added preprocessing details table (Table S1)
2. Supplementary Table S1: Included full acquisition parameters from source publications
3. Supplementary table renumbering: the S.1 table package now occupies Tables S1-S5, with Table S1 for iEEG dataset details, Table S2 for scalp EEG dataset details, Table S3 for shared analysis parameters, and Tables S4-S5 for symbols and operators

Response:

The preprocessing details are provided in Methods Section 4.1. We added a summary table (Table

BiSCA Fit: Cfg1 $|B|$ $R^2=0.962$, MAE=0.00, coh=1.000 | Cfg2 $|B|$ $R^2=0.961$, MAE=0.00, coh=1.000
 Top: Separated harmonics | Bottom: Overlapping harmonics

Fig. R2-7. BiSCA bispectrum model fit

S1):

Item	EEG (HarMNqEEG)	iEEG (Frauscher atlas)
Condition	Eyes-closed resting-state; at least 1 min artifact-free	Resting wakefulness eyes-closed; 60 s artifact-free
Sampling rate	Varies by site; min 0.5-35 Hz response	200/256/512/1000/1024/2000 Hz; downsample to 200 Hz
Filtering	Cross-spectra 1.17-19.14 Hz (0.39 Hz resolution)	Bandpass 0.5-80 Hz
Artifact handling	Within-site cleaning + centralized QC	Strict physiological-channel selection
Montage/reference	10-20 montage; average reference	Bipolar referencing

BiSCA pipeline: $F_s = 200$ Hz, multitaper bispectrum estimation (window 300 samples / 1.5 s, NFFT = 300, frequency range 0–50 Hz, 75% overlap, $NW = 1.5$, scaled sine tapers; Riedel & Sidorenko, 1995). No additional filtering, ICA, artifact rejection, or re-referencing was applied beyond what was present in the source datasets. For HarMNqEEG, approximately 960 out of ~1700 total records had time-series available.

4.2 Minor 2: Laplacian Re-referencing

Reviewer’s Comment:

Given that volume conduction will almost certainly add a spatial confound I am confused as to why they did not choose some form of Laplacian re-referencing, or at least some mention of it for future work.

Changes Made:

Discussion, “Limitations and future directions” paragraph: > “*We did not apply Laplacian re-referencing in this study. Given the substantial effect of volume conduction on bicoherence prevalence demonstrated by our forward-model simulation (Section S.3.3; 94.7% significant EEG channels vs. 30.9% iEEG from identical sources), we used iEEG for spatially resolved analysis to minimize volume conduction confounds. Laplacian re-referencing (Hjorth, 1975; Perrin et al., 1989; Kayser & Tenke, 2015) and source-space bispectral imaging (Shahbazi Avarvand et al., 2018) represent important future directions for extending bispectral analysis to scalp EEG.*”

Response:

We used iEEG for spatially resolved analyses because forward-model evidence (Supplementary Section S.3.3: 94.7% significant channels in scalp EEG vs. 30.9% in iEEG from identical sources) suggests substantial volume-conduction inflation at the scalp level. Scalp EEG was used only to

show sensor-level observability of nonlinear resonance; Laplacian re-referencing and source-space bispectral imaging remain future work.

4.3 Minor 3: Signed Panels and Negative Values in Fig.1

Reviewer’s Comment:

As bicoherence is defined on the range $[0, 1]$ why is it negative in the panels of Fig.1?

Changes Made:

See Major 5 Changes Made (Fig.1 caption clarification).

Response:

This concern is addressed by the quantity clarification in Major 5. Negative values occur because the shown quantity is the signed real part $\text{Re}\{\hat{b}(f_1, f_2)\}$, not the magnitude bicoherence $|\hat{b}(f_1, f_2)|$.

4.4 Minor 4: Triangular Domain

Reviewer’s Comment:

Is there any reason they did not choose to plot bicoherence over the triangular inner domain (e.g., $0 < f_2 < f_1$ and $f_1 + f_2 < f_s/2$), given its symmetry?

Changes Made:

Section 4.2.2, bispectrum display convention paragraph: > *“The principal non-redundant domain can be written as $0 \leq f_2 \leq f_1$ with $f_1 + f_2 \leq f_s/2$. For visualization, we display the closure-constrained region $f_1 \geq 0, f_2 \geq 0, f_1 + f_2 \leq f_{\max}$ (without symmetry reduction) so all closure-valid bins remain visible.”*

Response:

The triangular inner domain is indeed the minimal non-redundant region for a real-valued process, following established precedent (Brillinger & Rosenblatt, 1967b, pp. 193–195; Nikias & Petropulu, 1993, Fig. 2.11). We chose the full closure-constrained region ($f_1 \geq 0, f_2 \geq 0, f_1 + f_2 \leq f_{\max}$) without additionally collapsing by permutation symmetry, so all closure-valid bins remain visible for direct visual comparison.

4.5 Minor 5: Parameter d Values

Reviewer’s Comment:

The ‘free’ parametrisation of their equation (14) seems problematic, as the only values for the parameter d would be $2n, n \in \mathbb{N}$, otherwise t will be negative or complex.

Changes Made:

Eq.(14): Changed “ $((f - \mu)/\sigma)^d$ ” to “ $(|f - \mu|/\sigma)^d$ ”

Response:

We use $|f - \mu|^d$ in the implementation, ensuring non-negativity for any $d > 0$:

$$t(f; \mu, \sigma, \nu, d) = \left(1 + \left|\frac{f - \mu}{\sigma}\right|^d\right)^{-\nu}$$

4.6 Minor 6: Eyes-Open vs Eyes-Closed Comparison

Reviewer’s Comment:

Given that the EO-EC transition is arguably one of the most robust electroencephalographic phenomena I would have assumed that they would have compared these two states particularly as it would have provided useful information regarding the state dependency of any non-linear activity.

Changes Made:

Discussion, “Limitations and future directions” paragraph: > *“The current analysis focuses on eyes-closed resting state. Comparing eyes-open and eyes-closed conditions would provide insights into state-dependent nonlinearity and is an important direction for future work.”*

Response:

We agree that the EO-EC transition is one of the most robust EEG phenomena, and comparing BiSCA parameters across these states would be a compelling test of the method’s sensitivity to state-dependent nonlinearity. The current study focused exclusively on eyes-closed (EC) resting state because the HarMNqEEG normative database ($N = 1,966$) and the MNI iEEG dataset both consist of EC recordings; this design choice maximizes statistical power for atlas construction and enables direct scalp–intracranial comparison under matched conditions. EO-EC comparison would require additional data collection with matched protocols and is an important direction for future work.

4.7 Minor 7: Number of Oscillatory Processes K

Reviewer's Comment:

It is mentioned that the maximum number of oscillatory 'rho' processes was chosen by AIC (i.e., $2k - 2\ln[L]$), but as far as I could ascertain the k chosen (i.e., the number of modelled oscillatory processes) were not specified.

Changes Made:

- Methods Section 4.2.4: clarified notation in the AIC term, distinguishing total free-parameter count N_p from oscillatory-component count K .
- Methods Section 4.2.4: added explicit model-order search range and the selected range in this study ($K = 3-6$ in typical channels).
- Supplementary Section S.4.2: added full parameter-accounting details (dataset-level summary, worked example, and effective-sample-size context for AIC conditioning).

Response:

In the AIC expression $2k - 2\ln L$, the reviewer's k corresponds to our total free-parameter count N_p , whereas K denotes the number of oscillatory components. We now report the selected oscillatory-component range explicitly in the manuscript ($K = 3-6$ in typical channels).

Further parameter-accounting details and a concrete worked example are provided in Supplementary Section S.4.2.

5 Supplementary Section S.5.2: Pair-Indexed Bispectrum Parametrization and Triad Sparsity

BiSCA parametrizes the bispectrum (Eq.~15) as a pair-indexed weighted sum of kernel functions on the closure manifold, including both the aperiodic (ξ) and oscillatory (ρ) contributions:

$$\hat{B}(f_1, f_2) = h_{B,\xi} \cdot t_2(f_1, f_2; 0, 0) + \sum_{(m,n) \in \mathcal{P}} h_{B,m,n} \cdot t_2(f_1, f_2; \mu_m, \mu_n)$$

where $h_{B,\xi} \in \mathbb{C}$ is the aperiodic bispectral amplitude, \mathcal{P} is the set of closure-admissible oscillatory peak pairs, and $h_{B,m,n} \in \mathbb{C}$. The product kernel follows from the Volterra bispectrum structure (Marzocca et al., 2008, Eq. 24; Schetzen, 1980, Eq. 7.3-8; Nikias & Petropulu, 1993, Eq. 2.75):

$$t_2(f_1, f_2; \mu_m, \mu_n) = t(f_1; \mu_m) \cdot t(f_2; \mu_n) \cdot t(f_1 + f_2; \mu_m + \mu_n)$$

5.1 Triad sparsity under harmonic constraint

The bispectrum lives on the closure manifold $\{(f_1, f_2, f_1+f_2)\}$, so only triads satisfying $\mu_m \approx \mu_k + \mu_l$ produce non-negligible contributions. Under the harmonic constraint $\mu_k = kf_0$, the closure condition reduces to integer arithmetic $k + l = m$. For $K=6$ peaks: $K^3 = 216$ candidate terms collapse to 15 admissible triads and 9 unique pairs after symmetry—a $24\times$ parameter reduction (Fig. R2-5). This is not an ad hoc simplification but a direct consequence of the closure structure (Brillinger & Rosenblatt, 1967b, p. 193; Nikias & Petropulu, 1993, Section 2.3). Numerical validation: Config 1 ($K=6$) has 15 admissible triads (6.9% of K^3); Config 2 ($K=4$) has 6 admissible triads (9.4%).

5.2 Equation mapping to reviewer’s derivation

Reviewer Eq.	Topic	Our response
(R.5)→(R.6)	$S_x = S_\xi + S_\rho$	Proof (spectrum) in Supplementary Section S.6.3: ξ - ρ independence $\Rightarrow S_{\xi\rho} = 0$
(R.7)	Missing indices on $\mu_{\rho,k}$	Corrected: centers and amplitudes peak-specific; shape shared (Major 2)
(R.8)–(R.10)	$B_x = B_\xi + B_\rho$	Proof (bispectrum) in Supplementary Section S.6.3: mixed cumulants vanish under independence
(R.11)–(R.12)	Triple sum $\sum_{k,l,m}$	BiSCA uses double-sum on closure manifold
(R.13)–(R.14)	Auxiliary functions t^2	Our t_2 is the product kernel above; shape tied across axes
(R.15)–(R.16)	Constant phase	$h_{B,m,n} \in \mathbb{C}$ per-peak-pair; phase varies across pairs

Fig. R2-4. Bispectrum with admissible triad markers

Fig. R2-5. Parameter reduction under harmonic constraint

6 Supplementary Section S.6.3: Independence Assumption ($\xi \perp\!\!\!\perp \rho$): Mathematical Basis, Physiological Motivation, and Numerical Validation

We note that $\xi \perp\!\!\!\perp \rho$ is a commonly used modeling assumption in periodic/apperiodic decomposition, not unique to our method. We treat this independence primarily as a working assumption for cumulant additivity (Brillinger, 2001, Chapter 4), informed by mathematical proof, neurophysiological considerations, and numerical validation.

6.1 Mathematical basis

Under the independence assumption ($\xi \perp\!\!\!\perp \rho$), cumulants of all orders decompose additively (Brillinger, 1965, Eq. 4.11; Nikias & Petropulu, 1993, Properties 3–4, pp. 13–14):

$$\text{cum}_k(x) = \text{cum}_k(\xi) + \text{cum}_k(\rho), \quad \forall k$$

This yields $S_x = S_\xi + S_\rho$ ($k=2$) and $B_x = B_\xi + B_\rho$ ($k=3$). The BiSCA bispectrum model (Eq.~15) retains both B_ξ and B_ρ as explicit terms with free complex amplitudes ($h_{B,\xi}$ and $h_{B,\rho,m,n}$). If ξ is Gaussian linear (GL), $B_\xi = 0$ and $h_{B,\xi} = 0$; if ξ is non-Gaussian linear (NGL), $h_{B,\xi} \neq 0$ and the model captures the flat bispectral background. The empirical finding $h_{B,\xi} \approx 0$ provides data-driven support for the GL interpretation.

Proof (spectrum). $S_x = E[(X_\xi + X_\rho)(X_\xi^* + X_\rho^*)] = S_\xi + S_{\xi\rho} + S_{\rho\xi} + S_\rho$. Independence $\Rightarrow E[x_\xi(t)x_\rho(s)] = 0$ for all t, s , so $S_{\xi\rho} = S_{\rho\xi} = 0$ and $S_x = S_\xi + S_\rho$. \square

Proof (bispectrum). Expanding $X = X_\xi + X_\rho$ in $B_x = E[X(f_1)X(f_2)X^*(f_1+f_2)]$ yields $2^3 = 8$ terms: two pure (B_ξ, B_ρ) and six mixed. Each mixed term is a mixed 3rd-order cumulant that vanishes under independence. Hence $B_x = B_\xi + B_\rho$. \square

Note that Eq.~(15) retains both B_ξ and B_ρ with free amplitudes; the GL conclusion ($h_{B,\xi} \approx 0$) is an empirical result from the fit, not a structural constraint of the model.

6.2 Neurophysiological motivation

Oscillatory generators can be lamina-specific (Silva et al., 1991), with distinct laminar directionality for low- vs. high-frequency rhythms (van Kerkoerle et al., 2014) and a reported spectrolaminar motif (Mendoza-Halliday et al., 2022). The aperiodic trend may arise partly from subcritical network dynamics whose changes affect peaks and trend relatively independently; however, shared biophysical parameters (synaptic kinetics, E/I ratio) multiplicatively affect both components, so full statistical independence is not guaranteed (Brake et al., 2024). Our framework adopts an additive periodic/apperiodic decomposition on the natural power scale. In this respect it is closer in spirit to ξ - π and to IRASA, which also separate periodic and aperiodic structure on the natural spectrum scale, than to log-spectrum parameterizations such as specparam. At the same time, these

approaches still differ in parameterization, estimation procedure and inferential scope. Definitive verification requires experimental designs that can isolate the two components at the source level.

6.3 Empirical dissociation

Periodic and aperiodic parameters exhibit differential sensitivity to disease (Alzheimer's: periodic-driven, Kópcanová et al., 2024), pharmacology (haloperidol: oscillatory-selective, Gallo et al., 2024), and developmental trajectories (distinct lifespan curves that may reflect different biological substrates, Li et al., 2025).

6.4 Numerical validation

Two simulation configurations (shared setup: $\xi = \text{AR}(1)$ $a=0.98$, quadratic nonlinearity $\alpha=0.8$, $T=60$ s, $F_s=200$ Hz)—Config 1 ($f_1=12$, $f_2=8$ Hz, separated harmonics) and Config 2 ($f_1=16$, $f_2=8$ Hz, overlapping harmonics)—support the following: (i) same-frequency cross-spectra are negligible within ρ (off/diag < -33 dB; Fig. R2-2), (ii) composite-level additivity approximately holds ($S_x \approx S_\xi + S_\rho$; spectrum error $\leq 1.1\%$, bispectrum error $\leq 10.3\%$; Fig. R2-3). The bispectral error is consistent with the higher estimation variance of 3rd-order statistics (Brillinger, 2001, Ch. 4), rather than clear evidence of ξ - ρ leakage.

Fig. R2-1. Higher-order spectral overview

Fig. R2-2. Cross-spectrum grid within Rho

Fig. R2-3. Additivity verification

6.5 Limitation

Periodic/apperiodic separation is a working assumption; perfect separation is not achievable given the infinite solutions to this inverse problem (Gerster et al., 2022). The independence assumption $\xi \perp \rho$ likewise warrants further empirical investigation across brain regions, states, and pathologies.

7 Supplementary Section S.4.2: Model-Order and Parameter Accounting for AIC Selection

To make the AIC notation explicit and reproducible, we report the parameter accounting used when selecting the number of oscillatory components.

Across $N = 1,771$ iEEG channels, AIC-selected models show a median of 5 fitted peaks above 5% spectral power (IQR: 3–6).

Worked example ($k_s = 0$, $k_h = 3$, $K = 3$): - Internal peak count: $K_{\text{int}} = 4$ - Bispectral grid dimension: $N_1 = 5$ - Spectrum parameters: $N_{p,S} = 14$ - Bispectrum parameters: $N_{p,B} = 20$ - Total free parameters: $N_p = 34$

Effective-sample-size context for AIC conditioning: - $N_e = N_f + (N_f^2 + N_f)/2 = 1952$ for $N_f = 61$ - $N_e/N_p \approx 57$

This clarifies that in $2k - 2 \ln L$, the k term corresponds to total free-parameter count N_p , whereas K denotes oscillatory-component count.

8 References

8.1 Theoretical Framework

- Brillinger, D. R. (1965). An introduction to polyspectra. *Ann. Math. Statist.*, 36(5), 1351-1374. DOI: 10.1214/aoms/1177699896.
- Brillinger, D. R., & Rosenblatt, M. (1967a). Asymptotic theory of estimates of k th-order spectra. In B. Harris (Ed.), *Spectral Analysis of Time Series* (pp. 153-188). Wiley.
- Brillinger, D. R., & Rosenblatt, M. (1967b). Computation and interpretation of k th-order spectra. In B. Harris (Ed.), *Spectral Analysis of Time Series* (pp. 189-232). Wiley.
- Brillinger, D. R. (2001). *Time Series: Data Analysis and Theory*. SIAM, Classics in Applied Mathematics 36, Chapter 4.
- Berg, A., Paparoditis, E., & Politis, D. N. (2010). A bootstrap test for time series linearity. *J. Stat. Plan. Inference*, 140(12), 3841-3857. DOI: 10.1016/j.jspi.2010.04.047.
- Nichols, J. M., Olson, C. C., Michalowicz, J. V., & Bucholtz, F. (2009). The bispectrum and bicoherence for quadratically nonlinear systems subject to non-Gaussian inputs. *IEEE Trans. Signal Process.*, 57(10), 3879-3890. DOI: 10.1109/TSP.2009.2024267.
- Marzocca, P., Nichols, J. M., Milanese, A., Seaver, M., & Trickey, S. T. (2008). Second-order spectra for quadratic nonlinear systems by Volterra functional series. *Mech. Syst. Signal Process.*, 22(8), 1882-1895. DOI: 10.1016/j.ymsp.2008.02.002.
- Schetzen, M. (1980). *The Volterra and Wiener Theories of Nonlinear Systems*. John Wiley & Sons. ISBN: 978-0-471-04455-0.
- Nikias, C. L., & Petropulu, A. P. (1993). *Higher-Order Spectra Analysis: A Nonlinear Signal Processing Framework*. Prentice Hall. ISBN: 978-0-13-678210-9.

8.2 Methods

- Hjorth, B. (1975). An on-line transformation of EEG scalp potentials into orthogonal source derivations. *Electroenceph. Clin. Neurophysiol.*, 39(5), 526-530. DOI: 10.1016/0013-4694(75)90056-5.
- Perrin, F., Pernier, J., Bertrand, O., & Echallier, J. F. (1989). Spherical splines for scalp potential and current density mapping. *Electroenceph. Clin. Neurophysiol.*, 72(2), 184-187. DOI: 10.1016/0013-4694(89)90180-6.
- Kayser, J., & Tenke, C. E. (2015). On the benefits of using surface Laplacian (current source density) methodology in electrophysiology. *Int. J. Psychophysiol.*, 97(3), 171-173. DOI: 10.1016/j.ijpsycho.2015.06.001.
- Riedel, K. S., & Sidorenko, A. (1995). Minimum bias multiple taper spectral estimation. *IEEE Trans. Signal Process.*, 43(1), 188-195. DOI: 10.1109/78.365298.

8.3 Neurophysiological Motivation

- Silva, L. R., Amitai, Y., & Connors, B. W. (1991). Intrinsic oscillations of neocortex generated by layer 5 pyramidal neurons. *Science*, 251, 432-435. DOI: 10.1126/science.1989498.

- van Kerkoerle, T., et al. (2014). Alpha and gamma oscillations characterize feedback and feedforward processing in monkey visual cortex. *PNAS*, 111(40), 14332-14341. DOI: 10.1073/pnas.1402773111.
- Mendoza-Halliday, D., et al. (2022). A ubiquitous spectrolaminar motif of local field potential power across the primate cortex. *bioRxiv*. DOI: 10.1101/2022.09.30.510398.
- Brake, N., et al. (2024). A neurophysiological basis for aperiodic EEG and the background spectral trend. *Nat. Commun.*, 15, 1514. DOI: 10.1038/s41467-024-45922-8.
- Shahbazi Avarvand, F., et al. (2018). Localizing bicoherence from EEG and MEG signals. *NeuroImage*, 174, 352-363. DOI: 10.1016/j.neuroimage.2018.03.040.

8.4 Empirical Dissociation

- Kopcanová, M., et al. (2024). Resting-state EEG signatures of Alzheimer's disease are driven by periodic but not aperiodic changes. *Neurobiol. Dis.*, 190, 106380. DOI: 10.1016/j.nbd.2023.106380.
- Gallo, D., et al. (2024). Differential effects of haloperidol on neural oscillations during wakefulness and sleep. *Neuroscience*, 560, 67-76. DOI: 10.1016/j.neuroscience.2024.09.020.
- Li, M., et al. (2025). Aperiodic and Periodic EEG Component Lifespan Trajectories. *bioRxiv*. DOI: 10.1101/2025.08.26.672407.

8.5 Limitations

- Gerster, M., et al. (2022). Separating neural oscillations from aperiodic 1/f activity: challenges and recommendations. *Neuroinformatics*, 20, 991-1012. DOI: 10.1007/s12021-022-09581-8.

Response to Reviewer #3

The influence of nonlinear resonance on human cortical oscillations

Communications Biology

Contents

1 Detailed Response to Reviewer #3	2
2 Major Comments	2
2.1 Major 1: Distinction between nonlinearity and non-Gaussianity	2
3 Minor Comments	3
3.1 Minor 1: Phase coupling terminology	3
3.2 Minor 2: Alpha peak in Fig. 1D	4
3.3 Minor 3: Statistics meaning in Fig. 4	4
3.4 Minor 4: Typo “recordings of thrytum”	4
3.5 Minor 5: Capitalization error	5
3.6 Minor 6: Source localization and mixing artifacts	5
3.7 Minor 7: Non-stationarity statement	5
3.8 Minor 8: Section 4.2.1 polish (NW calculation)	6
3.9 Minor 9: Bispectrum property for linear systems	6
3.10 Minor 10: Effective number of segments reference	6
3.11 Minor 11: “Warmup” clarification	7
4 Supplementary Section S.6: Mathematical Derivation	7
4.1 Conceptual distinction between system nonlinearity and input non-Gaussianity . . .	7
4.2 Definitions	8
4.3 Case 1: Linear + Gaussian i.i.d. (GL)	10
4.4 Case 2: Linear + Non-Gaussian i.i.d. (NGL)	10
4.5 Case 3: Nonlinear + Gaussian i.i.d. (GNL)	11
4.6 Case 4: Nonlinear + Non-Gaussian i.i.d. (NGNL)	12
4.7 Summary: The 2\$×\$2 Classification Matrix	12
4.8 Colored Noise and the Cascade Property	13
4.9 The Trivial Model $y(t) = x(t)$	13
5 References	13

Manuscript: The influence of nonlinear resonance on human cortical oscillations

Journal: Communications Biology

1 Detailed Response to Reviewer #3

2 Major Comments

2.1 Major 1: Distinction between nonlinearity and non-Gaussianity

Reviewer's Comment: The authors distinguish nonlinearity from non-Gaussianity by the spectral properties of bicoherence. Roughly speaking: a largely flat and significantly non-vanishing bicoherence spectrum corresponds to a linear system with a non-Gaussian input, and sharp peaks correspond to nonlinear dynamics with Gaussian input. First of all, I wonder whether sharp peaks could not also arise from non-Gaussian input in combination with nonlinear dynamics. More importantly, I wonder why one cannot explain any univariate data as a linear dynamical system with non-Gaussian input by considering the signal itself as input. Formally speaking: let $x(t)$ be the data, then we could write $x(t)=y(t)$ with $y(t)=x(t)$. This is a trivial linear AR-model with all AR-matrices set to zero and with non-Gaussian input $y(t)$. Apparently, the authors have a more restrictive understanding of what properties the non-Gaussian input may have. From personal experience I agree that bicoherence occurs as sharp peaks in EEG superimposed on a largely flat background. But in LFP data one can also observe Theta-Gamma phase-amplitude coupling resulting in basically a line in the bicoherence plot, i.e. a sharp peak in the low frequency and broad band in the high frequency. How would the authors interpret this in terms of non-Gaussianity and nonlinearity?

Changes Made:

1. Methods Section 4.2.3 was revised to add the explicit i.i.d. innovation assumption under the Brillinger-Hinich framework and its implication for constant bicoherence in linear systems.
2. Introduction Section 2.1 was revised to separate system nonlinearity (Volterra kernels) from input non-Gaussianity (third cumulant), and to state their distinct bicoherence signatures.
3. The Discussion (Limitations) was revised to note that wideband PAC patterns (e.g., theta-gamma ridges) require model extensions beyond the current discrete-harmonic parametrization.
4. Supplementary Section S.6 was added to provide the mathematical derivations for the four GL/NGL/GNL/NGNL cases, together with the colored-noise cascade extension.

Response: We revised the manuscript to make this distinction more explicit. In our framework, sharp peaks were not attributed to nonlinearity alone: they could also arise in the combined non-Gaussian + nonlinear case, whereas the constant-background case required a linear system driven by i.i.d. non-Gaussian innovations.

BiSCA decomposed a single observed univariate time series under the assumption of statistical independence: $x(t) = x_\xi(t) + x_\rho(t)$, with $x_\xi \perp\!\!\!\perp x_\rho$ assumed. Here x_ξ followed a linear system driven by i.i.d. innovation ϵ_t (Wold representation: $X_\xi = \sum_j a_j \epsilon_{t-j}$), and x_ρ followed a nonlinear system with Volterra kernel H_2 generating quadratic phase coupling. The innovations were assumed to be zero-mean i.i.d. processes, following the standard Brillinger-Hinich framework (Hinich, 1982; Brillinger, 2001; Berg et al., 2010).

- (1) Yes, sharp peaks could arise from non-Gaussian + nonlinear dynamics (the NGNL case), where bicoherence exhibited localized peaks superimposed on a non-zero constant background. Our two complementary tests (median-based for non-Gaussianity, maximum-based for nonlinearity) distinguished all four categories: GL, NGL, GNL, NGNL (Table 1, Fig. 1I; four-case simulations in Supplementary Fig. S2).
- (2) The trivial construction $x(t) = y(t)$ did not qualify as a linear model in this framework because “non-Gaussian input” specifically meant an i.i.d. innovation – independent, identically distributed, with no temporal dependence (Berg et al., 2010, p. 2). In the trivial construction $y(t) = x(t)$, the “input” carried temporal structure and was not i.i.d., so the constant-bicoherence theorem did not apply. For a true linear system with i.i.d. input, the transfer-function terms canceled in bicoherence normalization, yielding $|b| = |\gamma_3|/\sigma^3 = \text{const}$ (derivation in Supplementary Section S.6.2).
- (3) The theta-gamma PAC “line” pattern was interpreted as a bicoherence ridge arising from wideband cross-frequency coupling in nonlinear systems. For a quadratic Volterra system, the interaction between $H_1(\omega_\theta)$ (narrow resonance) and $H_1(\omega_\gamma)$ (broadband) produced a bicoherence ridge along $(f_\theta, f_\gamma - f_\theta)$ (Marzocca et al., 2008; Nichols et al., 2009). The current BiSCA parametric model focused on discrete harmonic peaks; extension to wideband PAC would require continuous Volterra kernel estimation.

See Supplementary Section S.6 for the complete GL/NGL/GNL/NGNL mathematical framework with derivations, including the colored-noise cascade property that extended the constant-bicoherence result to filtered innovations.

3 Minor Comments

3.1 Minor 1: Phase coupling terminology

Reviewer’s Comment: “phase coupling” is also within frequency.

Changes Made: Introduction was revised to “cross-frequency phase coupling”.

Response: Phase coupling can also occur within a single frequency. Because the bispectrum specifically quantifies interactions among distinct frequency components $(f_1, f_2, f_1 + f_2)$, we revised the text to “cross-frequency phase coupling” to reflect the scope of our analysis more precisely.

3.2 Minor 2: Alpha peak in Fig. 1D

Reviewer’s Comment: What is denoted as ”alpha peak” is already a coupling between (approximately) 10 Hz and 20 Hz, i.e. a coupling with the 2/1 harmonic.

Changes Made:

- Fig. 1D caption now states that the “alpha peak” at (10, 10) Hz denotes $\alpha + \alpha \rightarrow 2\alpha$ coupling.
- Fig. 1D legend now labels source lines for α , 2α , and 3α components.
- Added a clarifying note that each bifrequency point (f_1, f_2) represents coupling that generates $f_1 + f_2$.

Response: Correct. The “alpha peak” at coordinates (α, α) represents $\alpha + \alpha \rightarrow 2\alpha$ coupling, i.e., the 10 Hz and 10 Hz components phase-couple to generate the 20 Hz component. The colored lines in Fig. 1D are center lines from the BiSCA parametric model, showing where each harmonic participates as a coupling source.

3.3 Minor 3: Statistics meaning in Fig. 4

Reviewer’s Comment: The meaning of statistics is unclear at this point. Are the low or high values the statistically significant ones?

Changes Made: Added to Fig. 4 caption: “The dashed line indicates the significance threshold; values above this line indicate statistically significant rejection of Gaussianity (A) or linearity (B).” We also added text in the figure to show the region where the null hypothesis is rejected.

Response: In Fig. 4A-B, higher values indicate stronger deviation from Gaussianity (panel A, $b^{(G)}$) or linearity (panel B, $b^{(L)}$). Values above the dashed threshold reject the null hypothesis.

3.4 Minor 4: Typo “recordings of thrytum”

Reviewer’s Comment: I guess that should read ”recordings of the rhythm”.

Changes Made: Discussion Section 3.3: Changed “recordings of thrytum” to “recordings of the rhythm.”

Response: Corrected.

3.5 Minor 5: Capitalization error

Reviewer’s Comment: ”nonlinearity. a similar ...”. The ”a” should be capital.

Changes Made: Discussion Section 3.4: Changed “nonlinearity. a similar” to “nonlinearity. A similar.”

Response: Corrected.

3.6 Minor 6: Source localization and mixing artifacts

Reviewer’s Comment: ”Still, some spurious mixing in scalp recordings cannot entirely ruled out. To resolve these issues, source localization combined with BiSCA will help to confirm that the identified nonlinear interactions correspond to true neurophysiological coupling rather than mixing artifacts”. This confused me. All the analysis done here is univariate and coupling is found between signals at different frequencies and not between signals at different sensor locations. Mixing artifacts are problematic for the interpretation of the latter.

Changes Made: Discussion Section 3.3 was revised to clarify that the concern is univariate source superposition at a single sensor rather than multivariate inter-sensor mixing, and that source localization could help distinguish intrinsic generator dynamics from source overlap.

Response: Our concern was univariate-specific: a single electrode may record a superposition of multiple cortical sources. Source localization could help separate spatially overlapping sources and assess whether the observed nonlinearity reflected intrinsic dynamics of a specific generator or source overlap.

3.7 Minor 7: Non-stationarity statement

Reviewer’s Comment: ”Nevertheless, there is no question that brain activity is nonstationary.” I understand that this is a widespread view and the whole question is not so relevant for this paper. Therefore, I won’t ask for changing anything. Just for the record: I totally disagree with this and I would appreciate if the authors just drop the formulation ”there is no question”.

Changes Made: Discussion Section 3.4: “brain activity is widely considered to exhibit nonstationarity”

Response: Fair point. Stationarity (or quasi-stationarity) may hold over appropriately chosen time windows—which is our assumption. We softened the language.

3.8 Minor 8: Section 4.2.1 polish (NW calculation)

Reviewer’s Comment: Section 4.2.1 should be polished a little bit. In line 488, the NW is not part of any sentence. Also, when $NW=1.5$, then $K=2NW-1=2$ and not 4.

Changes Made:

- Section 4.2.1: Revised to form complete sentence: “In this paper, we use $NW = 1.5$, which yields $K = 2NW - 1 = 2$ tapers for the recordings.”
- Corrected $K = 4 \rightarrow K = 2$ throughout the relevant section

Response: Corrected. When $NW = 1.5$, the number of tapers is $K = 2 \times 1.5 - 1 = 2$, not 4.

3.9 Minor 9: Bispectrum property for linear systems

Reviewer’s Comment: “the bispectrum has the favorable properties of zeroing Gaussian processes, being constant for linear systems with non-Gaussian input, ...” This relates to my major objection. Why should it be zero for linear systems with non-Gaussian input?

Changes Made: Section 4.2.2 was revised to distinguish more explicitly among (i) zero bispectrum for Gaussian processes, (ii) constant but non-zero bicoherence for linear systems with non-Gaussian input, and (iii) frequency-specific peaks associated with quadratic nonlinear interactions.

Section 4.2.3 was revised to state explicitly that, for a linear system driven by non-Gaussian input, the transfer-function dependence cancels under bicoherence normalization, leaving a constant magnitude across frequency pairs.

Response: Agreed; the original text was imprecise. For linear systems with non-Gaussian input, the bispectrum is generally not zero; rather, the bicoherence magnitude is constant across frequency pairs. This constancy follows because the transfer-function dependence in $[B_X(\omega_1, \omega_2) = \beta H(\omega_1)H(\omega_2)H^*(\omega_1 + \omega_2)]$ was canceled by the corresponding power-spectrum factors in the bicoherence normalization, leaving a frequency-independent magnitude. The revised manuscript now distinguished explicitly between (i) zero bispectrum for Gaussian processes, (ii) constant but non-zero bicoherence for linear systems with non-Gaussian input, and (iii) frequency-specific bicoherence peaks associated with quadratic nonlinear interactions. Details were provided in Major 1 and Supplementary Section S.6.

3.10 Minor 10: Effective number of segments reference

Reviewer’s Comment: It would be helpful if the authors write that the estimation of the effective number of segments is explained below.

Changes Made: Section 4.2.3.1: “(see Section 4.2.3.3 below for estimation procedure)”

Response: Added forward reference.

3.11 Minor 11: “Warmup” clarification

Reviewer’s Comment: ”warmup” I guess this is the starting value for the LM algorithm, right?

Changes Made: Section 4.3: “as initial parameter values (starting points) for the spectrum and bispectrum joint fit”

Response: Yes. “Warmup” refers to using spectrum-only fit parameters as initial values for the joint spectrum and bispectrum fit (Levenberg-Marquardt optimization).

4 Supplementary Section S.6: Mathematical Derivation

4.1 Conceptual distinction between system nonlinearity and input non-Gaussianity

System linearity/nonlinearity and input Gaussianity/non-Gaussianity are orthogonal properties and should not be conflated.

- A linear system can produce non-Gaussian output when driven by non-Gaussian i.i.d. innovation.
- A nonlinear system can produce approximately Gaussian output under aggregation (central-limit effects).
- In the Brillinger-Hinich setting, input non-Gaussianity is characterized by the third cumulant γ_3 , while system nonlinearity is characterized by higher-order Volterra kernels (H_2, H_3, \dots) .
- These mechanisms produce distinct bicoherence signatures: constant bicoherence across frequencies for linear systems with non-Gaussian i.i.d. input, versus frequency-structured peaks for nonlinear phase coupling.
- Power-spectrum inspection alone cannot distinguish these generative mechanisms; higher-order spectral analysis is required.

4.2 Definitions

4.2.1 S.6.1 Linear Process: Wold Decomposition (Wold 1938; Brillinger 2001)

Wold's Theorem: Any stationary, zero-mean time series can be decomposed as:

$$X_t = \underbrace{\sum_{j=0}^{\infty} a_j \epsilon_{t-j}}_{\text{MA}(\infty) \text{ component}} + D_t \quad (1)$$

where ϵ_t is an i.i.d. innovation and D_t is a deterministic component (typically zero for stochastic processes).

Finite-order approximations:

Form	Time Domain	Transfer Function $H(\omega)$
AR(p)	$X_t = \sum_{k=1}^p \phi_k X_{t-k} + \epsilon_t$	$\frac{1}{1 - \sum_{k=1}^p \phi_k e^{-i\omega k}}$
MA(q)	$X_t = \sum_{j=0}^q \theta_j \epsilon_{t-j}$	$\frac{\sum_{j=0}^q \theta_j e^{-i\omega j}}{\sum_{j=0}^q \theta_j e^{-i\omega j}}$
ARMA(p, q)	$X_t - \sum_k \phi_k X_{t-k} = \sum_j \theta_j \epsilon_{t-j}$	$\frac{\sum_{j=0}^q \theta_j e^{-i\omega j}}{1 - \sum_{k=1}^p \phi_k e^{-i\omega k}}$

Key insight: AR, MA, and ARMA are equivalent representations of the same linear process class—they differ only in parameterization, not in the underlying stochastic structure.

i.i.d. Innovation requirements: - $E[\epsilon_t] = 0$ (zero mean) - $\text{Var}(\epsilon_t) = \sigma^2$ (constant variance) - $\epsilon_t \perp \epsilon_s$ for $t \neq s$ (independence) - $E[\epsilon_t^3] = \gamma_3$ (third cumulant; = 0 for Gaussian) - $\text{cum}(\epsilon_t, \epsilon_s, \epsilon_r) = \gamma_3 \cdot \delta_{t,s,r}$ (diagonal structure due to i.i.d.)

4.2.2 S.6.2 Nonlinear Process: Volterra Series (Wiener 1958; Schetzen 1980)

The Volterra functional series generalizes linear convolution to nonlinear systems. Any fading-memory nonlinear system can be expanded as:

$$X_t = \underbrace{\sum_j h_1(j) \epsilon_{t-j}}_{\text{Linear (1st order)}} + \underbrace{\sum_{j,k} h_2(j, k) \epsilon_{t-j} \epsilon_{t-k}}_{\text{Quadratic (2nd order)}} + \dots \quad (2)$$

where $h_p(\tau_1, \dots, \tau_p)$ is the p -th order Volterra kernel and ϵ_t is the i.i.d. innovation.

Truncated approximations (by kernel order):

Truncation	Retained Kernels	Spectral Consequence
Linear ($p=1$)	h_1 only	Reduces to Wold representation (Section S.6.1)
Quadratic ($p\leq 2$)	h_1, h_2	Bispectral peaks from quadratic phase coupling; BiSCA operates here
Cubic ($p\leq 3$)	h_1, h_2, h_3	Adds trispectral structure (not modeled in BiSCA)

Key insight: The linear case (Section S.6.1) is the special case $h_2 = h_3 = \dots = 0$. Bispectral analysis specifically targets the quadratic kernel h_2 .

In frequency domain:

$$X(\omega) = H_1(\omega)U(\omega) + \iint H_2(\omega_1, \omega_2)U(\omega_1)U(\omega_2)\delta(\omega_1+\omega_2-\omega) d\omega_1 d\omega_2 + \dots$$

Special structures of H_2 : - Wiener model (linear filter \rightarrow static nonlinearity): $H_2(\omega_1, \omega_2) = \alpha \cdot H_1(\omega_1)H_1(\omega_2)$ – separable; used in BiSCA (Marzocca et al., 2008) - General Volterra: $H_2(\omega_1, \omega_2)$ non-separable – more expressive but harder to identify from data

Spectral consequences: - H_1 : determines the power spectrum shape ($S_X = |H_1|^2\sigma^2$ at leading order) - H_2 : introduces quadratic phase coupling (QPC), creating bispectral peaks at triads (f_1, f_2, f_1+f_2) - Higher kernels H_3, \dots : contribute to trispectrum and beyond (not modeled in BiSCA)

4.2.3 S.6.3 Power Spectrum (Wiener-Khinchin)

$$S_X(\omega) = \sum_{\tau=-\infty}^{\infty} R_X(\tau)e^{-i\omega\tau} \quad \text{where } R_X(\tau) = E[X_t X_{t+\tau}] \quad (3)$$

For linear systems with i.i.d. input: $S_X(\omega) = |H(\omega)|^2 \cdot \sigma^2$

4.2.4 S.6.4 Bispectrum and Bicoherence

Bispectrum:

$$B_X(\omega_1, \omega_2) = E[X(\omega_1)X(\omega_2)X^*(\omega_1 + \omega_2)] \quad (4)$$

Power spectrum:

$$f_X(\omega) = E[|X(\omega)|^2] \quad (5)$$

Squared bicoherence:

$$|b(\omega_1, \omega_2)|^2 = \frac{|B_X(\omega_1, \omega_2)|^2}{f_X(\omega_1)f_X(\omega_2)f_X(\omega_1 + \omega_2)} \quad (6)$$

4.3 Case 1: Linear + Gaussian i.i.d. (GL)

4.3.1 Conditions:

- System: Linear (AR, MA, or ARMA — all equivalent by Wold)
- Innovation: $\epsilon_t \sim N(0, \sigma^2)$, i.i.d.

4.3.2 Derivation:

Step 1 — Gaussian third cumulant: For Gaussian random variables, all cumulants of order > 2 are zero:

$$\gamma_3 = E[\epsilon_t^3] = 0 \quad (7)$$

Step 2 — Bispectrum of linear process: For any linear process $X(\omega) = H(\omega) \cdot U(\omega)$ with i.i.d. input (Brillinger 2001, Examples 2.9.1-2.9.2):

$$B_X(\omega_1, \omega_2) = \gamma_3 \cdot H(\omega_1) \cdot H(\omega_2) \cdot H^*(\omega_1 + \omega_2) \quad (8)$$

Step 3 — Result: Substituting $\gamma_3 = 0$:

$$B_X(\omega_1, \omega_2) = 0 \quad (9)$$

Therefore, taking the square root:

$$\boxed{|b(\omega_1, \omega_2)| = 0} \quad (\text{GL})$$

4.4 Case 2: Linear + Non-Gaussian i.i.d. (NGL)

4.4.1 Conditions:

- System: Linear (AR, MA, or ARMA — all equivalent by Wold)
- Innovation: ϵ_t is i.i.d. with non-Gaussian distribution (e.g., Pearson Type III)
- $\gamma_3 = E[\epsilon_t^3] \neq 0$

4.4.2 Derivation:

Step 1 — Bispectrum:

$$B_X(\omega_1, \omega_2) = \gamma_3 \cdot H(\omega_1) \cdot H(\omega_2) \cdot H^*(\omega_1 + \omega_2) \quad (10)$$

Step 2 — Power spectrum:

$$S_X(\omega) = \sigma^2 |H(\omega)|^2 \quad (11)$$

Step 3 — Bicoherence (key cancellation):

$$\begin{aligned} |b|^2 &= \frac{|B_X|^2}{S_X(\omega_1) \cdot S_X(\omega_2) \cdot S_X(\omega_1 + \omega_2)} \\ &= \frac{|\gamma_3|^2 |H(\omega_1)|^2 |H(\omega_2)|^2 |H(\omega_1 + \omega_2)|^2}{\sigma^6 |H(\omega_1)|^2 |H(\omega_2)|^2 |H(\omega_1 + \omega_2)|^2} = \frac{\gamma_3^2}{\sigma^6} \end{aligned} \quad (12)$$

The $|H(\omega)|^2$ terms cancel completely. Taking the square root:

$$\boxed{|b(\omega_1, \omega_2)| = |\gamma_3|/\sigma^3 = \text{CONSTANT}} \quad (\text{NGL})$$

This is Berg et al. (2010, p.5): “if a time series is linear, its normalized bispectrum is constant over all (λ_1, λ_2) ”

The transfer function $H(\omega)$ shapes the power spectrum (giving peaks at resonances), but this frequency dependence is normalized out in bicoherence. What remains is only the input’s skewness γ_3/σ^3 .

4.5 Case 3: Nonlinear + Gaussian i.i.d. (GNL)

4.5.1 Conditions:

- System: Nonlinear (Volterra with H_2 kernel, e.g., NAR: $X_t = \phi X_{t-1} + \beta X_{t-1}^2 + \epsilon_t$)
- Innovation: $\epsilon_t \sim N(0, \sigma^2)$, i.i.d.

4.5.2 Derivation:

For a Volterra system, the output bispectrum has contributions from both H_1 (linear) and H_2 (quadratic):

$$B_X(\omega_1, \omega_2) = \underbrace{\gamma_3 \cdot H_1(\omega_1) H_1(\omega_2) H_1^*(\omega_1 + \omega_2)}_{\text{Linear contribution (= 0 for Gaussian)}} + \underbrace{B_{H_2}(\omega_1, \omega_2)}_{\text{Nonlinear contribution}} \quad (13)$$

For Gaussian input, $\gamma_3 = 0$, so the linear contribution vanishes. But the nonlinear kernel H_2 creates quadratic phase coupling (QPC). The leading interaction term is (Nichols et al. 2009):

$$B_{H_2}(\omega_1, \omega_2) \approx 2\sigma^4 \cdot H_1(\omega_1) H_1(\omega_2) \cdot H_2^*(\omega_1, \omega_2) \quad (14)$$

Peaks arise because: - $H_2^*(\omega_1, \omega_2)$ does NOT factor as $H \cdot H \cdot H^*$ - Normalization does NOT cancel frequency dependence - Result: $|b(\omega_1, \omega_2)|$ has peaks at frequencies where $H_2(\omega_1, \omega_2)$ is large

$$\boxed{|b(\omega_1, \omega_2)| = \text{frequency-dependent (peaks)}} \quad (\text{GNL})$$

The nonlinear kernel creates energy transfer between frequencies (e.g., $\alpha + \alpha \rightarrow 2\alpha$ frequency doubling), which manifests as localized peaks in bicoherence at harmonic frequencies.

4.6 Case 4: Nonlinear + Non-Gaussian i.i.d. (NGNL)

4.6.1 Conditions:

- System: Nonlinear (Volterra with H_2 kernel)
- Innovation: ϵ_t is i.i.d. with non-Gaussian distribution, $\gamma_3 \neq 0$

4.6.2 Derivation:

The bispectrum now has both contributions:

$$B_X(\omega_1, \omega_2) = \underbrace{\gamma_3 \cdot H_1 \cdot H_1 \cdot H_1^*}_{\text{Non-Gaussian contribution}} + \underbrace{B_{H_2}(\omega_1, \omega_2)}_{\text{Nonlinear contribution}} \quad (15)$$

After normalization: - The first term gives a constant background (like NGL case): $|\gamma_3|/\sigma^3$ - The second term gives frequency-dependent peaks (like GNL case): $f(H_2)$

Result: To leading order, bicoherence has peaks superimposed on a non-zero background:

$$\boxed{|b(\omega_1, \omega_2)| \approx |\gamma_3|/\sigma^3 + f(H_2) = \text{constant background} + \text{peaks}} \quad (\text{NGNL})$$

4.7 Summary: The 2\$ \times \$2 Classification Matrix

Gaussian ($\gamma_3 = 0$)	Non-Gaussian ($\gamma_3 \neq 0$)
Linear: $ b = 0$	NGL: $ b = \gamma_3 /\sigma^3$ (constant)
Nonlinear: $ b = f(H_2)$ (frequency-dependent peaks)	NGNL: constant background + peaks

Here $\gamma_3 \equiv E[\epsilon_t^3]$ is the third cumulant of the driving innovation, and $|\gamma_3|/\sigma^3$ is its absolute skewness.

Key theoretical results: 1. Linearity = system property; Gaussianity = input property (orthogonal dimensions) 2. Linear system: $|H(\omega)|^2$ cancels in normalization, so $|b| = \text{constant}$ (or 0 if Gaussian) 3. Nonlinear system: H_2 doesn't factor, so $|b|$ has frequency-dependent peaks

4.8 Colored Noise and the Cascade Property

If $\epsilon_t = \sum_k b_k w_{t-k}$ (linear filter of i.i.d. w_t) and $X_t = \sum_j a_j \epsilon_{t-j}$ (second linear filter), then:

$$X_t = \sum_j a_j \sum_k b_k w_{t-j-k} = \sum_m c_m w_{t-m}, \quad c_m = (a * b)_m \quad (16)$$

In the frequency domain: $H_{\text{total}}(\omega) = G_1(\omega) \cdot G_2(\omega)$, where G_1 and G_2 are the transfer functions of the two linear stages. The cascade reduces to a single linear filter of the original i.i.d. source w_t , so the constant-bicoherence theorem applies regardless of how many linear filtering stages intervene.

4.9 The Trivial Model $y(t) = x(t)$

Setting $y(t) = x(t)$ with $A(\omega) = 1$ gives bispectrum equal to that of the structured signal itself:

$$B_X(\omega_1, \omega_2) = B_x(\omega_1, \omega_2) \quad (17)$$

$$|b(\omega_1, \omega_2)|^2 = \frac{|B_x(\omega_1, \omega_2)|^2}{f_x(\omega_1) f_x(\omega_2) f_x(\omega_1 + \omega_2)} \neq \text{constant} \quad (18)$$

Since $x(t)$ carries temporal structure and is not i.i.d., the factorization $B_X = \gamma_3 H H H^*$ (Eq. 8) does not hold, and the $|H|^2$ cancellation that produces constant bicoherence does not occur. This construction falls outside the definition of a linear process rather than contradicting the theorem.

5 References

Wold (1938). *A Study in the Analysis of Stationary Time Series*. Almqvist & Wiksell. Wiener, N. (1958). *Nonlinear Problems in Random Theory*. MIT Press. Brillinger (2001). *Time Series: Data Analysis and Theory*. SIAM. Examples 2.9.1-2.9.2. Berg et al. (2010). A bootstrap test for time series linearity. *J. Stat. Plan. Inference*, 140(12), 3841-3857. Hinich (1982). Testing for Gaussianity and linearity of a stationary time series. *J. Time Ser. Anal.*, 3(3), 169-176. Nichols et al. (2009). The bispectrum and bicoherence for quadratically nonlinear systems subject to non-Gaussian inputs. *IEEE Trans. Signal Process.*, 57(10), 3879-3890. Marzocca et al. (2008). Second-order spectra for quadratic nonlinear systems by Volterra functional series. *Mech. Syst. Signal Process.*, 22, 1882-1895. Schetzen, M. (1980). *The Volterra and Wiener Theories of Nonlinear Systems*. John Wiley & Sons.

Notation. Throughout this derivation, (m, n) denote the two free component indices, matching the pair indices of Eq. (15). In the triple sum, the third index is k , subject to the closure constraint $k = m + n$. The reviewer’s Eq. (R.11)–(R.12) use indices (k, l, m) ; these correspond to our (m, n, k) , respectively.

Two distinct mechanisms drive the reduction. They operate in sequence and must not be conflated:

1. **Closure condition** (physical): narrowband spectral localization forces $\mu_m + \mu_n \approx \mu_k$, eliminating the vast majority of K^3 terms. Under harmonic organization, this becomes $m + n = k$, which uniquely determines the third index k from the pair (m, n) , collapsing a triple sum to a pair sum. This is the primary reduction.
2. **Permutation symmetry** (algebraic): $B(f_1, f_2) = B(f_2, f_1)$ identifies pairs (m, n) and (n, m) , further halving the count when $m \neq n$.

Neither mechanism relies on independence among the ρ components. The closure condition is a consequence of the narrowband structure of the spectral peaks; the permutation symmetry is a general property of the bispectrum.

1 Bispectrum of a Univariate Time Series Has Two Free Frequencies

Before addressing the component-index reduction, we recall that the bispectrum of a univariate stationary time series is a function of **two** frequencies, not three. This follows from the third-order cumulant and its Fourier transform.

1.1 Third-order cumulant has two lags

For a zero-mean stationary process $x(t)$, the third-order cumulant (equivalent to the third-order moment for zero mean) is:

$$c_3(\tau_1, \tau_2) = \text{Cum}[x(t), x(t + \tau_1), x(t + \tau_2)] = \mathbb{E}[x(t)x(t + \tau_1)x(t + \tau_2)]. \quad (1)$$

Stationarity means the statistics depend only on the **lag differences** τ_1 and τ_2 , not on the absolute time t . Hence c_3 is a function of two variables, not three.

1.2 Bispectrum = 2D Fourier transform of c_3

The bispectrum is defined as the two-dimensional Fourier transform of the third-order cumulant (Nikias & Petropulu, 1993, Eq. 2.30; Brillinger, 1965):

$$C_3^x(\omega_1, \omega_2) = \sum_{\tau_1=-\infty}^{\infty} \sum_{\tau_2=-\infty}^{\infty} c_3(\tau_1, \tau_2) e^{-j(\omega_1\tau_1 + \omega_2\tau_2)}. \quad (2)$$

Since c_3 has two lag variables, its Fourier transform has **two frequency variables** (ω_1, ω_2) . There is no third independent frequency.

1.3 Connection to the triple product $E[X(f_1)X(f_2)X^*(f_3)]$

Alternatively, the bispectrum can be expressed as a frequency-domain expectation. For a stationary process, the Fourier coefficients satisfy:

$$\mathbb{E}[X(\omega_1)X(\omega_2)X^*(\omega_3)] = C_3^x(\omega_1, \omega_2) \cdot \delta(\omega_1 + \omega_2 - \omega_3). \quad (3)$$

The Kronecker/Dirac delta $\delta(\omega_1 + \omega_2 - \omega_3)$ enforces $\omega_3 = \omega_1 + \omega_2$.

To see why this delta appears, substitute the spectral representation $x(t) = \sum_{\omega} X(\omega) e^{j\omega t}$ into the third moment $\mathbb{E}[x(t) x(t + \tau_1) x(t + \tau_2)]$ and sum over t : the time summation produces $\delta(\omega_1 + \omega_2 - \omega_3)$, enforcing the closure $\omega_3 = \omega_1 + \omega_2$. In Brillinger & Rosenblatt's (1967b) spectral representation (their Eq. 2.14, p. 192), this delta function appears explicitly in the definition of the k -th order spectral density: $f(\lambda_1, \dots, \lambda_k) \delta(\sum_1^k \lambda_j) d\lambda_1 \cdots d\lambda_k = c(dZ(\lambda_1), \dots, dZ(\lambda_k))$, establishing $\sum \lambda_j \equiv 0$ as the mathematical expression of the closure manifold.

The bispectrum is *defined* as a function of two frequencies. Writing $B(f_1, f_2) = \mathbb{E}[X(f_1) X(f_2) X^*(f_1 + f_2)]$ is not an approximation—it is the definition, with $f_3 = f_1 + f_2$ enforced by stationarity. This is the frequency-domain closure manifold of Brillinger & Rosenblatt (1967b, pp. 193–195): $\lambda_1 + \lambda_2 + \lambda_3 = 0$, which reduces the three-frequency space to a two-frequency surface.

Contrast with spatial three-point correlations: In Cosmic Microwave Background (CMB) or turbulence studies, the three-point correlation $\langle \delta(\mathbf{k}_1) \delta(\mathbf{k}_2) \delta(\mathbf{k}_3) \rangle$ involves three *wavevectors* in 2D or 3D space, and the closure condition $\mathbf{k}_1 + \mathbf{k}_2 + \mathbf{k}_3 = 0$ reduces the dimensionality but does not collapse to a simple pair index. For a 2D field, each \mathbf{k}_i has 2 scalar components (6 total); the vector closure $\mathbf{k}_1 + \mathbf{k}_2 + \mathbf{k}_3 = 0$ imposes 2 scalar constraints, leaving $6 - 2 = 4$ free parameters (equivalently, a triangle in \mathbf{k} -space described by 2 side lengths and an angle, plus one rotational degree of freedom). For univariate time series, each frequency is a scalar, so $f_3 = f_1 + f_2$ fully determines the third quantity from the first two (Komatsu & Spergel, 2001).

2 The Oscillatory Bispectrum as a Triple Sum

Under the $\xi \perp \rho$ independence assumption (proved in Section S.6.3), the bispectrum decomposes as $B_x = B_\xi + B_\rho$. We focus on the oscillatory part.

The oscillatory signal is $x_\rho(t) = \sum_{m=1}^K x_{\rho,m}(t)$, where each $x_{\rho,m}$ is a narrowband process centered at frequency μ_m . Expanding $X_\rho = \sum_m X_{\rho,m}$ in the bispectrum definition:

$$\begin{aligned} B_\rho(f_1, f_2) &= \mathbb{E}[X_\rho(f_1) X_\rho(f_2) X_\rho^*(f_1 + f_2)] \\ &= \sum_{m=1}^K \sum_{n=1}^K \sum_{k=1}^K \mathbb{E}[X_{\rho,m}(f_1) X_{\rho,n}(f_2) X_{\rho,k}^*(f_1 + f_2)]. \end{aligned} \quad (4)$$

This is the reviewer's Eq. (R.12) (with shared shape parameters): K^3 terms in general.

3 Closure Condition: Why the Third Index Is Determined

The triple sum (4) has three apparently independent indices (m, n, k) . The third index k is in fact *determined* by (m, n) , so the triple sum collapses to a double sum.

3.1 Each component is narrowband

By construction (Eq. 13 in the manuscript), each oscillatory component $x_{\rho,m}$ is a narrowband process whose spectral energy is concentrated near frequency μ_m within a bandwidth of order $\sim 2\sigma_\rho$. The parametric kernel $t(f; \mu_m, \sigma_\rho, \nu_\rho, d_\rho)$ is sharply peaked at $f = \mu_m$ and decays rapidly away from it. In the Lorentzian case ($\nu = 1, d = 2$):

$$t(f; \mu_m, \sigma_\rho, 1, 2) = \frac{1}{1 + \left(\frac{f - \mu_m}{\sigma_\rho}\right)^2},$$

which falls to $\approx 1\%$ of its peak value when $|f - \mu_m| \approx 10\sigma_\rho$, i.e., well before reaching the next harmonic peak.

3.2 The kernel product selects the triad

The bispectrum $B_\rho(f_1, f_2)$ is evaluated at each point (f_1, f_2) in the bifrequency plane. For a given (f_1, f_2) , the (m, n, k) -th term in the triple sum involves the product:

$$\underbrace{t(f_1; \mu_m)}_{\text{large only if } f_1 \approx \mu_m} \cdot \underbrace{t(f_2; \mu_n)}_{\text{large only if } f_2 \approx \mu_n} \cdot \underbrace{t(f_1 + f_2; \mu_k)}_{\text{large only if } f_1 + f_2 \approx \mu_k}. \quad (5)$$

All three factors must be simultaneously non-negligible for the term to contribute. The first two factors constrain (f_1, f_2) to the neighborhood of (μ_m, μ_n) . Given this, $f_1 + f_2 \approx \mu_m + \mu_n$. The third factor then requires:

$$\mu_k \approx \mu_m + \mu_n. \quad (6)$$

This means that once (m, n) is chosen, k is **not free**—it is forced to be whichever component has center frequency closest to $\mu_m + \mu_n$. If no such component exists (i.e., $\mu_m + \mu_n$ does not match any μ_k), the third factor evaluates to approximately zero and the entire term is negligible.

3.3 From triple sum to double sum

We can now rewrite the triple sum by replacing the free index k with the constrained value $k^*(m, n)$ —the unique index satisfying (6):

$$\begin{aligned} B_\rho(f_1, f_2) &= \sum_{m=1}^K \sum_{n=1}^K \underbrace{\sum_{k=1}^K \mathbb{E}[X_{\rho,m}(f_1) X_{\rho,n}(f_2) X_{\rho,k}^*(f_1 + f_2)]}_{\text{only the } k=k^*(m,n) \text{ term survives}} \\ &\approx \sum_{m=1}^K \sum_{n=1}^K \mathbb{E}[X_{\rho,m}(f_1) X_{\rho,n}(f_2) X_{\rho,k^*}^*(f_1 + f_2)] \cdot \mathbf{1}[\mu_m + \mu_n \approx \mu_{k^*}]. \end{aligned} \quad (7)$$

The inner sum over k has collapsed: only one value of k contributes for each (m, n) .

Remark 1 (This is not a symmetry reduction). *The reviewer correctly noted that no algebraic symmetry of the tensor $\mathbb{E}[X_m X_n X_k^*]$ reduces 3 independent indices to 2. The reduction is a **physical** consequence: the narrowband kernel product acts as a selector that forces $k \approx m + n$, leaving only 2 free indices. Terms with $k \neq m + n$ are not zero by symmetry—they are negligible because the kernel product vanishes.*

3.4 Harmonic constraint makes the closure exact

The previous subsection established that k must satisfy $\mu_k \approx \mu_m + \mu_n$. Under the harmonic peak model $\mu_m = m f_0$ (where f_0 is the fundamental frequency), this approximate condition becomes exact:

$$m f_0 + n f_0 = k f_0 \quad \iff \quad \boxed{m + n = k}. \quad (8)$$

An admissible triad is therefore any (m, n, k) with $m, n \in \{1, \dots, K\}$ and $k = m + n \leq K$.

Summary: The triple sum $\sum_{m=1}^K \sum_{n=1}^K \sum_{k=1}^K$ collapses to a double sum $\sum_{m=1}^K \sum_{n=1}^K$ (with the constraint $m + n \leq K$) because:

- the narrowband kernel product forces $\mu_k = \mu_m + \mu_n$ (third index determined by first two);
- harmonicity converts this to $k = m + n$ (integer arithmetic).

The number of free summation indices drops from 3 to 2.

Remark 2 (Relationship to Brillinger's closure manifold). *As shown in Section 1, the closure $f_3 = f_1 + f_2$ follows from the definition of the bispectrum. Here we see how it manifests in the linear time-invariant (LTI) transfer function structure.*

Nikias & Petropulu (1993, Eq. 2.75) show that the n -th order output cumulant spectrum of a linear system is:

$$C_n^y(\omega_1, \dots, \omega_{n-1}) = H(\omega_1) \cdot H(\omega_2) \cdots H(\omega_{n-1}) \cdot H^*(\omega_1 + \cdots + \omega_{n-1}) \cdot C_n^x(\omega_1, \dots, \omega_{n-1}). \quad (9)$$

For the bispectrum ($n = 3$) with white noise input (Eq. 2.80):

$$C_3^y(\omega_1, \omega_2) = \gamma_3^x \cdot H(\omega_1) \cdot H(\omega_2) \cdot H^*(\omega_1 + \omega_2). \quad (10)$$

The closure is **structurally built into** these equations: the transfer function is evaluated at $n - 1$ free frequencies and at their sum. The sum frequency $\omega_1 + \omega_2$ is not a free variable—it is determined.

Our component-index reduction is the **discrete, narrowband specialization** of the same principle. When the signal consists of K discrete narrowband peaks rather than a continuum, the continuous closure manifold $f_1 + f_2 + f_3 = 0$ restricts to a finite set of points where $\mu_m + \mu_n = \mu_k$. In both cases, the underlying mechanism is identical: on the bispectral closure manifold, the third coordinate is determined by the first two, removing one degree of freedom.

	Continuous (Brillinger; Nikias)	Discrete (BiSCA)
Signal	Continuous spectrum	K narrowband peaks
Constraint	$\omega_3 = \omega_1 + \omega_2$	$\mu_k = \mu_m + \mu_n$
Free variables	(ω_1, ω_2) ; ω_3 determined	(m, n) ; k determined
Reduction	3D integral \rightarrow 2D	triple sum \rightarrow double sum
Key equation	Nikias Eq. 2.75 / 2.80	This derivation, Eq. (7)

The continuous case is exact (by definition of the bispectrum); the discrete case is a narrowband approximation whose accuracy depends on the ratio σ_ρ/f_0 (see Remark 3).

Remark 3 (Quantitative estimate of suppression). For typical BiSCA parameters ($\sigma_\rho/f_0 \approx 0.1$, $\nu = 1$, $d = 2$, i.e., Lorentzian kernel), a non-admissible triad whose sum frequency $\mu_m + \mu_n$ misses the nearest peak by $\Delta f = f_0$ experiences a kernel suppression factor of $t(f_0; 0, \sigma_\rho) = [1 + (f_0/\sigma_\rho)^2]^{-1} \approx 10^{-2}$ (for the Lorentzian case $\nu = 1$, $d = 2$). In the non-admissible triple product, only the third factor $t(f_1+f_2; \mu_k)$ is off-center; the first two factors $t(f_1; \mu_m) \approx 1$ and $t(f_2; \mu_n) \approx 1$ remain well-centered. The product kernel for non-admissible triads is thus suppressed by roughly two orders of magnitude relative to admissible triads. For the EEG data analyzed in this study, $\sigma_\rho/f_0 \approx 0.05$ – 0.15 , placing the analysis well within the narrowband regime.

4 Worked Example: $K = 2$

Peaks at $\mu_1 = f_0$ and $\mu_2 = 2f_0$. The triple sum has $K^3 = 8$ terms. The reviewer correctly observed that these are “8 independent components.” We now show that the closure condition—not a symmetry argument—eliminates 7 of the 8:

Result: Of the 8 triple-sum terms, exactly **1** satisfies the closure condition: $(m, n, k) = (1, 1, 2)$. This corresponds to the single pair $(m, n) = (1, 1)$.

Now consider the formal double sum in Eq. (15), which runs over $m = 1, \dots, K$ and $n = 1, \dots, m$, giving $K(K+1)/2 = 3$ formal pairs:

Reconciliation: Eq. (15) formally sums over 3 pairs, but the product kernel $t_2(f_1, f_2; \mu_m, \mu_n) = t(f_1; \mu_m) t(f_2; \mu_n) t(f_1+f_2; \mu_m+\mu_n)$ is negligible for pairs (2, 1) and (2, 2) because the third factor $t(f_1+f_2; 3f_0)$ or $t(f_1+f_2; 4f_0)$ finds no spectral peak to localize around—it evaluates to the far tail of the aperiodic kernel, contributing negligibly. In practice, the corresponding $h_{B,m,n}$ values are fitted to approximately zero.

Thus, for $K = 2$: 8 triple-sum terms \rightarrow 1 closure-admissible triad \rightarrow 1 active pair in the double sum (out of 3 formal pairs).

Table 1: All 8 triple-sum terms for $K = 2$. Only the term satisfying $m + n = k \leq K$ survives the closure condition.

m	n	k	$m+n$ vs k	Status
1	1	1	$2 \neq 1$	negligible
1	1	2	$2 = 2$	admissible ✓
1	2	1	$3 \neq 1$	negligible
1	2	2	$3 \neq 2$	negligible
2	1	1	$3 \neq 1$	negligible
2	1	2	$3 \neq 2$	negligible
2	2	1	$4 \neq 1$	negligible
2	2	2	$4 \neq 2$	negligible

Table 2: All formal pairs in Eq. (15) for $K = 2$ and their closure status.

Pair (m, n)	Sum freq $\mu_m + \mu_n$	Peak at sum freq?	Status
(1, 1)	$2f_0 = \mu_2$	Yes (peak at μ_2)	admissible ✓
(2, 1)	$3f_0$	No (exceeds $\mu_K = 2f_0$)	kernel ≈ 0
(2, 2)	$4f_0$	No	kernel ≈ 0

Remark 4 (Addressing the reviewer’s $K = 2$ concern). *The reviewer noted that $K = 2$ yields “8 independent components, which ... admits of no symmetries to reduce them to the $K(K+1)/2 = 3$ terms.” This is correct if one seeks a pure algebraic symmetry of the tensor $\mathbb{E}[X_m X_n X_k^*]$. The reduction does not come from tensor symmetry; it comes from the **narrowband closure condition**: 7 of the 8 terms are negligible because the product of three narrowband kernels centered at mismatched frequencies evaluates to approximately zero. The remaining 1 admissible term maps to 1 of the 3 formal pairs; the other 2 formal pairs have negligible kernel support and their fitted amplitudes converge to $h_B \approx 0$.*

5 Worked Example: $K = 6$

Peaks at $\mu_m = m f_0$ for $m = 1, \dots, 6$. The triple sum has $K^3 = 216$ terms.

5.1 Enumerating admissible triads

An admissible triad requires $m + n = k$ with $1 \leq m, n \leq 6$ and $k \leq 6$:

Table 3: All admissible triads for $K = 6$, grouped by sum index $k = m + n$.

k	Triads (m, n, k)	Count
2	(1, 1, 2)	1
3	(1, 2, 3), (2, 1, 3)	2
4	(1, 3, 4), (3, 1, 4), (2, 2, 4)	3
5	(1, 4, 5), (4, 1, 5), (2, 3, 5), (3, 2, 5)	4
6	(1, 5, 6), (5, 1, 6), (2, 4, 6), (4, 2, 6), (3, 3, 6)	5
Total admissible triads		15

5.2 From triads to pairs

Since $k = m + n$ is determined by (m, n) , each triad maps to a unique pair. Pairs (m, n) and (n, m) contribute to the same bifrequency region (by the permutation symmetry $B(f_1, f_2) = B(f_2, f_1)$), so we identify them. Restricting to $n \leq m$:

Table 4: Unique closure-admissible pairs for $K = 6$, with their implied sum peak.

Pair (m, n)	$n \leq m$	Sum $k = m+n$	Triads merged	Bifrequency center
(1, 1)	✓	2	(1, 1, 2)	(f_0, f_0)
(2, 1)	✓	3	(1, 2, 3) + (2, 1, 3)	$(2f_0, f_0)$
(3, 1)	✓	4	(1, 3, 4) + (3, 1, 4)	$(3f_0, f_0)$
(4, 1)	✓	5	(1, 4, 5) + (4, 1, 5)	$(4f_0, f_0)$
(5, 1)	✓	6	(1, 5, 6) + (5, 1, 6)	$(5f_0, f_0)$
(2, 2)	✓	4	(2, 2, 4)	$(2f_0, 2f_0)$
(3, 2)	✓	5	(2, 3, 5) + (3, 2, 5)	$(3f_0, 2f_0)$
(4, 2)	✓	6	(2, 4, 6) + (4, 2, 6)	$(4f_0, 2f_0)$
(3, 3)	✓	6	(3, 3, 6)	$(3f_0, 3f_0)$
Total unique pairs				9

5.3 Counting summary for $K = 6$

Remark 5 (Addressing the reviewer’s $K = 6$ concern). *The reviewer noted that “for $K = 6$ the $K^3 = 216$ reduces to 15 triads (or ‘9 unique pairs’ – which makes no sense in the calculation of the bispectrum), which is at odds with the number of terms implied by their equation (15) of 21.” We acknowledge that our previous response was ambiguous. The three counts are consistent but describe different stages of the reduction: $216 \rightarrow 15$ (closure), $15 \rightarrow 9$ (permutation symmetry), and 21 is the total formal pair count in Eq. (15), of which 9 are active and 12 are kernel-suppressed. The relationship is $21 = 9 + 12$, not $21 = 15$ or $21 = 9$. See Table 6 for a complete reconciliation.*

Table 5: Parameter count comparison for $K = 6$.

Quantity	Count	Explanation
Triple-sum terms (K^3)	216	General expansion
Closure-admissible triads	15	$m + n = k \leq K$, all orders of m, n
Unique pairs ($n \leq m$)	9	After permutation symmetry
Formal pairs in Eq. (15)	21	$K(K+1)/2$, all (m, n) with $n \leq m$
Active pairs in Eq. (15)	9	Pairs where $m + n \leq K$ (kernel non-negligible)
Inactive pairs in Eq. (15)	12	$m + n > K$; kernel ≈ 0 , fitted $h_B \approx 0$

Supplementary Figs. S8 and S9 visualize this sparsity: Fig. S8 overlays the admissible triads on the empirical bispectrum map, and Fig. S9 displays the $K^3 \rightarrow$ pairs \rightarrow pairs+sym parameter reduction bar chart.

5.4 Empirical verification in 3D frequency space

The preceding analysis relies on the algebraic closure condition $m + n = k$. To verify this independently of the standard FFT-based bispectrum estimator, we compute the **full three-dimensional** bispectral field $|E[X(f_1)X(f_2)X^*(f_3)]|$ on a (f_1, f_2, f_3) grid *without* constraining $f_3 = f_1 + f_2$.

For a stationary process, $E[X(f_1) X(f_2) X^*(f_3)] = C_3^x(f_1, f_2) \cdot \delta(f_1 + f_2 - f_3)$ (Eq. 3): off-manifold values are theoretically zero. With finite data, off-manifold contributions have random phase across segments and cancel through averaging, while on-manifold contributions maintain coherent phase (the biphasic) and accumulate.

Figure 1 shows the result for a simulation with $K = 6$ stochastic narrowband peaks at the harmonic frequencies from Table 3. Three features are visible:

1. The bispectral energy (red/orange scatter) concentrates on the closure plane $f_3 = f_1 + f_2$ (blue surface), confirming the dimensionality reduction from 3D to 2D.
2. Within the closure plane, energy localizes at the 15 admissible triad positions (green diamonds), confirming the further reduction from a continuous 2D surface to a discrete set of points determined by $m + n = k$.
3. The grey cube $[0, F_s/2]^3$ delineates the full Nyquist-limited frequency space; the vast majority of this volume is empty.

This provides an empirical confirmation of the closure-manifold argument that is independent of the conventional 2D bispectrum computation path.

Figure 1: Empirical 3D bispectral field $|E[X(f_1) X(f_2) X^*(f_3)]|$ computed on a full (f_1, f_2, f_3) grid without constraining $f_3 = f_1 + f_2$. Signal: $K = 6$ stochastic narrowband peaks at 4, 8, 12, 16, 20, 24 Hz with quadratic coupling ($\alpha = 0.8, F_s = 60$ Hz, > 7000 segments). Grey cube: Nyquist-limited space $[0, F_s/2]^3$. Blue surface: closure manifold $f_3 = f_1 + f_2$. Green diamonds: the 15 admissible triads from Table 3. Red/orange scatter: empirical bispectral energy above the 99.5-th percentile threshold. Each admissible triad coincides with an empirical peak.

6 Why Eq. (15) Sums Over $K(K+1)/2$ Pairs

6.1 Combinatorial origin of $K(K+1)/2$

The $K(K+1)/2$ formal pairs in Eq. (15) enumerate all unordered pairs drawn from K component indices *with replacement* (i.e., self-pairs like (m, m) are included). Combinatorially:

$$\binom{K+1}{2} = \underbrace{K}_{\text{self-pairs } (m,m)} + \underbrace{\binom{K}{2}}_{\text{cross-pairs } (m,n), m>n} = \frac{K(K+1)}{2}. \quad (11)$$

For $K = 6$: 6 self-pairs + 15 cross-pairs = 21 formal pairs. This is the number that the reviewer correctly computed from Eq. (15).

6.2 Reconciling 15, 9, and 21

Three numbers appeared in our previous response and caused confusion. We clarify each:

Table 6: The three counts and what each represents for $K = 6$.

Count	What it counts	Origin
15	Admissible triads (m, n, k) with $m+n = k \leq K$, both orders of (m, n)	Closure $m+n = k$, all orderings
9	Unique admissible pairs (m, n) with $n \leq m$ and $m+n \leq K$	Closure + permutation symmetry
21	Formal pairs (m, n) with $n \leq m$ in the double sum of Eq. (15)	$K(K+1)/2$: all pairs with replacement

The relationship: 21 = 9 (closure-admissible) + 12 (non-admissible, $m+n > K$). The 15 admissible triads reduce to 9 unique pairs after permutation symmetry. Thus $15 \rightarrow 9 \subset 21$.

6.3 Why Eq. (15) includes non-admissible pairs

The double sum in Eq. (15) runs over *all* $K(K+1)/2$ pairs (m, n) with $1 \leq n \leq m \leq K$, not just the 9 closure-admissible pairs. This is by design, for two reasons:

1. **Self-regularization by the kernel.** For non-admissible pairs (where $m + n > K$), the product kernel $t_2(f_1, f_2; \mu_m, \mu_n) = t(f_1; \mu_m) t(f_2; \mu_n) t(f_1+f_2; \mu_m+\mu_n)$ is negligible everywhere in the bifrequency plane, because the third factor $t(f_1+f_2; \mu_m+\mu_n)$ is centered at a frequency $\mu_m + \mu_n > K f_0$ where no spectral peak exists. The kernel evaluates to the far tail of the aperiodic background.
2. **Generality beyond strict harmonicity.** When peak centers are fitted freely (option 2 in the manuscript), the harmonic relationship $\mu_m = m f_0$ is only approximate. Retaining all formal pairs allows the model to capture near-harmonic interactions without hard-coding which pairs are ‘‘admissible.’’ The kernel naturally suppresses non-contributing terms.

In summary: the $K(K+1)/2$ formal pairs in Eq. (15) include both the closure-admissible pairs (which carry the bispectral signal) and the non-admissible pairs (which are automatically suppressed by the kernel). The fitted complex amplitudes $h_{B,m,n}$ for non-admissible pairs converge to approximately zero.

7 The Complete Reduction Chain

Assembling all steps, the derivation from the general triple sum to Eq. (15) proceeds as follows:

$$\underbrace{\sum_{m=1}^K \sum_{n=1}^K \sum_{k=1}^K \mathbb{E}[X_{\rho,m}(f_1) X_{\rho,n}(f_2) X_{\rho,k}^*(f_1+f_2)]}_{\text{Triple sum: } K^3 \text{ terms (reviewer's Eq. R.12)}} \quad (12)$$

Closure condition $\mu_m + \mu_n \approx \mu_k$ → Retain only triads where the three narrowband kernels overlap.

Harmonic constraint $m+n=k$ → The index k is determined by (m, n) ; triple index → pair index. K^3 → a sparse set of admissible triads.

Permutation symmetry $(m,n) \sim (n,m)$ → Restrict to $n \leq m$; merge permuted triads into single pairs.

Absorb phases and amplitudes into $h_{B,m,n} \in \mathbb{C}$ → For each admissible pair (m, n) with $k = m+n$, the amplitude and phase factors from the reviewer's Eq. (R.11) collapse to a single complex scalar:

$$h_{B,m,n} := h_{\rho,m}^{1/2} h_{\rho,n}^{1/2} h_{\rho,k}^{1/2} e^{i(\phi_m + \phi_n - \phi_k)},$$

where the constant biphas $\phi_m + \phi_n - \phi_k$ is absorbed into $\arg(h_{B,m,n})$ and the amplitude product into $|h_{B,m,n}|$. This is valid because the kernel product is sharply localized near (μ_m, μ_n) , so the phase varies negligibly within the support region (the standard narrowband approximation).

Parametrize with product kernel → Replace narrowband Fourier amplitudes with the parametric kernel t :

$$B_\rho(f_1, f_2) \approx \sum_{\substack{(m,n): n \leq m, \\ m+n \leq K}} h_{B,\rho,m,n} t(f_1; \mu_m) t(f_2; \mu_n) t(f_1+f_2; \mu_{m+n}).$$

Extend to all formal pairs → For implementation generality, let the sum run over all $K(K+1)/2$ pairs; the kernel automatically suppresses non-admissible terms:

$$\hat{B}_\rho(f_1, f_2) = \sum_{m=1}^K \sum_{n=1}^m h_{B,\rho,m,n} t_2(f_1, f_2; \mu_m, \mu_n, \sigma_\rho, \nu_\rho, d_\rho). \quad (13)$$

Adding the aperiodic term $h_{B,\xi,\xi} t_2(\dots; \mu_\xi, \mu_\xi, \sigma_\xi, \nu_\xi, d_\xi)$ completes Eq. (15).

8 General Formula for the Number of Admissible Pairs

For K harmonic peaks with $\mu_m = m f_0$, the number of admissible pairs (m, n) with $n \leq m$ and $m+n \leq K$ is:

$$N_{\text{pairs}} = \lfloor K^2/4 \rfloor. \quad (14)$$

Table 7: Admissible pair counts vs. formal pair counts.

K	K^3 (triple)	Admissible triads	Admissible pairs	$K(K+1)/2$ (formal)
2	8	1	1	3
3	27	3	2	6
4	64	6	4	10
5	125	10	6	15
6	216	15	9	21

9 Summary

The reduction from the triple-indexed bispectral expansion (reviewer’s Eq. R.12) to the pair-indexed BiSCA model (Eq. 15) proceeds through:

1. **Narrowband localization:** Each $x_{\rho,m}$ occupies a narrow frequency band around μ_m ; the triple product is non-negligible only when $\mu_m + \mu_n \approx \mu_k$.
2. **Closure constraint:** Under harmonic organization ($\mu_m = mf_0$), the condition becomes $m + n = k$, which uniquely determines k from (m, n) , collapsing the triple index to a pair index.
3. **Permutation symmetry:** $B(f_1, f_2) = B(f_2, f_1)$ allows restricting to $n \leq m$.
4. **Complex amplitude absorption:** Phase terms and Volterra kernel contributions are absorbed into per-pair complex scalars $h_{B,m,n} \in \mathbb{C}$.
5. **Formal sum over all pairs:** Eq. (15) sums over all $K(K+1)/2$ pairs for generality; the product kernel t_2 automatically suppresses non-admissible terms (those with $m + n > K$).

This is a function approximation exploiting the sparsity of the bispectral closure manifold, not a claim about independence of sub-processes within ρ .

Remark 6 (Limitation: broadband coupling). *The narrowband closure condition assumes each component occupies a narrow spectral band. Broadband cross-frequency interactions—such as theta–gamma phase-amplitude coupling, which produces extended ridges rather than localized peaks in the bicoherence—are not captured by the present pair-indexed model. Modeling such interactions would require frequency-dependent quadratic kernels $H_2(\omega_1, \omega_2)$ beyond the separable structure assumed here.*

References

- [1] Brillinger, D. R. (1965). An introduction to polyspectra. *Ann. Math. Statist.*, 36(5), 1351–1374.
- [2] Brillinger, D. R., & Rosenblatt, M. (1967). Computation and interpretation of k th-order spectra. In B. Harris (Ed.), *Spectral Analysis of Time Series* (pp. 189–232). Wiley.
- [3] Nikias, C. L., & Petropulu, A. P. (1993). *Higher-Order Spectra Analysis: A Nonlinear Signal Processing Framework*. Prentice Hall. [Eq. 2.75, 2.80]
- [4] Marzocca, P., et al. (2008). Second-order spectra for quadratic nonlinear systems by Volterra functional series. *Mech. Syst. Signal Process.*, 22(8), 1882–1895. [Eq. 24]
- [5] Schetzen, M. (1980). *The Volterra and Wiener Theories of Nonlinear Systems*. Wiley. [Eq. 7.3-8]
- [6] Kim, Y. C., & Powers, E. J. (1979). Digital bispectral analysis and its applications to nonlinear wave interactions. *IEEE Trans. Plasma Sci.*, 7(2), 120–131.
- [7] Schmidt, O. T. (2020). Bispectral mode decomposition of nonlinear flows. *Nonlinear Dynamics*, 102, 2479–2501.
- [8] Bedrosian, E., & Rice, S. O. (1971). The output properties of Volterra systems driven by harmonic and Gaussian inputs. *Proc. IEEE*, 59(12), 1688–1707.
- [9] Komatsu, E., & Spergel, D. N. (2001). Acoustic signatures in the primary microwave background bispectrum. *Phys. Rev. D*, 63, 063002.

Review: The influence of nonlinear resonance on human cortical oscillations

Wang et al 2025

COMMSBIO-25-11257-T

December 9, 2025

This is an interesting and potentially very important communication in which the authors aim to decompose resting electroencephalographic (EEG and ECoG) activity into aperiodic and oscillatory activity using a newly developed method that they refer to as BiSpectral EEG Component Analysis (BiSCA). This is of particular relevance given our incomplete understanding regarding the extent to which resting macroscopic cortical electrophysiological activity is composed of Gaussian random activity and non-linear dynamical activity, a resolution that has considerable importance in our ability to understand and model the dynamical genesis of macroscopic brain activity such that we can better detect its perturbation in health and disease.

However, as described their method is not sufficiently clear to enable independent replication, an obvious requirement for the communication of a novel and potentially important method. Further, corresponding ambiguities in their description potentially imply that their conclusions cannot be regarded as in any sense definitive, in particular that aperiodic background activity (assuming it meaningfully exists) exhibits no evidence of quadratic cross-frequency coupling.

1 Major Comments

My major concern centres around the unclear derivation of BiSCA in section 4.2.4. By attempting my own derivation I find multiple errors in the corresponding equations and some of the assumptions, particularly those regarding the independence of the various theoretically defined “oscillatory” processes. In order to avoid any misunderstanding I outline, in some detail, my own derivation to highlight what I believe are significant problems with their formulation as presented.

It seems that their starting point is to assume that EEG/ECoG activity is the sum of two, not necessarily independent, random processes

$$x(t) = x_\xi(t) + x_\rho(t) \quad (1)$$

$$= x_\xi(t) + \sum_k x_{\rho,k}(t) \quad (2)$$

where $x_\xi(t)$ corresponds to the “aperiodic” process and $x_{\rho,k}$ the k -th “oscillatory process”. Further, it seems to be assumed (though nowhere obviously stated) that the Fourier transform of each of these $k + 1$ processes is modelled as

$$x_\xi(t) \longleftrightarrow X_\xi(f) \equiv \sqrt{h_\xi} t_\xi(f) \exp[i\phi_\xi(f)] \quad (3)$$

$$x_{\rho,k}(t) \longleftrightarrow X_{\rho,k}(f) \equiv \sqrt{h_{\rho,k}} t_{\rho,k}(f) \exp[i\phi_{\rho,k}(f)] \quad (4)$$

where the respective amplitudes, $t_\xi(f), t_{\rho,k}(f)$ are defined parametrically according to their equation 14, while the parametric forms for $\phi_\xi(f), \phi_{\rho,k}(f)$ are left undefined.

From this basis the power spectrum will be

$$X(f)X^*(f) = \left[X_\xi(f) + \sum_k X_{\rho,k}(f) \right] \left[X_\xi^*(f) + \sum_k X_{\rho,k}^*(f) \right] \quad (5)$$

which will only be equal to

$$X_\xi(f)X_\xi^*(f) + \sum_k X_{\rho,k}(f)X_{\rho,k}^*(f) \quad (6)$$

iff all cross-spectral densities are zero. This does not seem to make sense as the inter-frequency phase couplings, to be evaluated using the bispectrum, necessarily assume that $X_{\rho,l}X_{\rho,m}^* \neq 0$.

Nevertheless, assuming that Eqn.(6) holds, it can be rewritten in terms of their parametric form as

$$h_{S,\xi} t(f, 0, \sigma_\xi, \nu_\xi, d_\xi)^2 + \sum_k h_{S,\rho,k} t(f, \mu_{\rho,k}, \sigma_{\rho,k}, \nu_{\rho,k}, d_{\rho,k})^2 \quad (7)$$

which is not their equation 13 as they have omitted the indices on the “oscillatory” (ρ) parameters.

By assuming that the “ ξ ” and “ ρ ” processes are phase uncorrelated the theoretical bispectrum $B(f_1, f_2)$ is

$$B(f_1, f_2) = [X_\xi(f_1) + X_\rho(f_1)] [X_\xi(f_2) + X_\rho(f_2)] [X_\xi^*(f_1 + f_2) + X_\rho^*(f_1 + f_2)] \quad (8)$$

$$= X_\xi(f_1)X_\xi(f_2)X_\xi^*(f_1 + f_2) + X_\rho(f_1)X_\rho(f_2)X_\rho^*(f_1 + f_2) \quad (9)$$

$$= B_\xi(f_1, f_2) + B_\rho(f_1, f_2) \quad (10)$$

Which, by using the parametric forms of Eqns. (3) and (4) can be written as

$$B(f_1, f_2) = h_\xi^{3/2} t_\xi(f_1) t_\xi(f_2) t_\xi(f_1 + f_2) \exp[i(\phi_\xi(f_1) + \phi_\xi(f_2) - \phi_\xi(f_1 + f_2))] + \quad (11)$$

$$\sum_{k,l,m} h_{\rho,k}^{1/2} h_{\rho,l}^{1/2} h_{\rho,m}^{1/2} t_{\rho,k}(f_1) t_{\rho,l}(f_2) t_{\rho,m}(f_1 + f_2) \exp[i(\phi_{\rho,k}(f_1) + \phi_{\rho,l}(f_2) - \phi_{\rho,m}(f_1 + f_2))]$$

which in terms of their parameterisations and symbols can be written as

$$B(f_1, f_2) = h_{B,\xi}(f_1, f_2) t_{2\xi}(f_1, f_2, \sigma_\xi, \nu_\xi, d_\xi) + \quad (12)$$

$$= \sum_{k,l,m} h_{B,\rho,lmn}(f_1, f_2) t_{2\rho}(f_1, f_2, \mu_{\rho,k}, \mu_{\rho,l}, \mu_{\rho,m}, \sigma_{\rho,k}, \sigma_{\rho,l}, \sigma_{\rho,m}, \nu_{\rho,k}, \nu_{\rho,l}, \nu_{\rho,m}, d_{\rho,k}, d_{\rho,l}, d_{\rho,m})$$

where

$$t_{2\xi}(f_1, f_2, \sigma_\xi, \nu_\xi, d_\xi) = t(f_1, 0, \sigma_\xi, \nu_\xi, d_\xi) t(f_2, 0, \sigma_\xi, \nu_\xi, d_\xi) t(f_1 + f_2, 0, \sigma_\xi, \nu_\xi, d_\xi) \quad (13)$$

$$t_{2\rho}(f_1, f_2, \mu_{\rho,k}, \mu_{\rho,l}, \mu_{\rho,m}, \sigma_{\rho,k}, \sigma_{\rho,l}, \sigma_{\rho,m}, \nu_{\rho,k}, \nu_{\rho,l}, \nu_{\rho,m}, d_{\rho,k}, d_{\rho,l}, d_{\rho,m}) =$$

$$t(f_1, \mu_{\rho,k}, \sigma_{\rho,k}, \nu_{\rho,k}, d_{\rho,k}) t(f_2, \mu_{\rho,l}, \sigma_{\rho,l}, \nu_{\rho,l}, d_{\rho,l}) t(f_1 + f_2, \mu_{\rho,m}, \sigma_{\rho,m}, \nu_{\rho,m}, d_{\rho,m}) \quad (14)$$

and where the ‘‘constants’’ $h_{B,\dots}$ will be complex and unless otherwise specified will depend on the frequencies f_1 and f_2 . In their formulation the $h_{B,\dots}$ appear to be independent of frequency, which makes me wonder if they have assumed that the $\phi_\xi(f)$ and the $\phi_{\rho,k}(f)$ are constant. If this is the case then for $\nu_\xi = \nu_{\rho,k} = 1$ and $d_\xi = d_{\rho,k} = 2$ (i.e., Lorentzian) we have

$$e^{-4\pi^2\sigma_\xi^2|t|} e^{i\phi_\xi} \longleftrightarrow X_\xi(f, \sigma_\xi) \quad (15)$$

$$e^{-4\pi^2\sigma_{\rho,k}^2|t|} e^{-i(2\pi\mu_{\rho,k}t - \phi_{\rho,k})} \longleftrightarrow X_{\rho,k}(f, \mu_{\rho,k}, \sigma_{\rho,k}) \quad (16)$$

i.e., the $x_\xi(t)$ process is effectively a noise driven decaying non-oscillatory exponential and the $x_{\rho,k}(t)$ processes are effectively noise driven damped oscillations of frequency $\mu_{\rho,k}$. On this basis the bispectrum for the $x_\xi(t)$ will be trivially flat, as I notice seems to be the case in their Figure 1F.

Comparison between my Eqn (12) and their equation (15) when the real and imaginary parts are taken, and comparison between their auxiliary function t_2 and my auxiliary functions $t_{2\xi}$ and $t_{2\rho}$ show that they are quite different, so much so that I cannot conclude that they have calculated a meaningful theoretical bispectrum to fit to empirical data.

2 Minor comments

- The parameters around the resting EEG data used are not clear. It appears that resting data was recorded using multiple different systems, but there is no specification of sampling frequencies, resampling, band pass filtering, artefact rejection or any other form of preprocessing. The only information we can infer is that recordings were sampled/resampled at/to 200 Hz.
- Given that volume conduction will almost certainly add a spatial confound I am confused as to why they did not choose some form of Laplacian re-referencing, or at least some mention of it for future work.
- As bicoherence is defined on the range $[0, 1]$ why is it negative in the panels of Figure 1?
- Is there any reason they did not choose to plot bicoherence over the triangular inner domain (e.g., $0 < f_2 < f_1$ and $f_1 + f_2 < f_s/2$), given its symmetry?
- The “free” parametrisation of their equation (14) seems problematic, as the only values for the parameter d would be $2n$, $n \in \mathbb{N}$, otherwise t will be negative or complex.
- Given that the EO-EC transition is arguably one of the most robust electroencephalographic phenomena I would have assumed that they would have compared these two states particularly as it would have provided useful information regarding the state dependency of any non-linear activity.
- It is mentioned that the maximum number of oscillatory “rho” processes was chosen by AIC (i.e., $2k - 2\ln[L]$), but as far as I could ascertain the k chosen (i.e., the number of modelled oscillatory processes) were not specified.